# NoisyGL: A Comprehensive Benchmark for Graph Neural Networks under Label Noise

**Zhonghao Wang**[1], **Danyu Sun**[1], **Sheng Zhou**[1*], **Haobo Wang**[1],
**Jiapei Fan**[2], **Longtao Huang**[2], **Jiajun Bu**[1]
[1]Zhejiang Key Laboratory of Accessible Perception and Intelligent Systems,
Collage of Computer Science, Zhejiang University     [2]Alibaba Group
{wangzhonghao, zhousheng_zju, wanghaobo, bjj}@zju.edu.cn,
danyu.21@intl.zju.edu.cn,
{jiapei.fjp, kaiyang.hlt}@alibaba-inc.com

## Abstract

Graph Neural Networks (GNNs) exhibit strong potential in node classification tasks through a message-passing mechanism. However, their performance often hinges on high-quality node labels, which are challenging to obtain in real-world scenarios due to unreliable sources or adversarial attacks. Consequently, *label noise* is common in real-world graph data, negatively impacting GNNs by propagating incorrect information during training. To address this issue, the study of Graph Neural Networks under Label Noise (GLN) has recently gained traction. However, due to variations in dataset selection, data splitting, and preprocessing techniques, the community currently lacks a comprehensive benchmark, which impedes deeper understanding and further development of GLN. To fill this gap, we introduce NoisyGL in this paper, the first comprehensive benchmark for graph neural networks under label noise. NoisyGL enables fair comparisons and detailed analyses of GLN methods on noisy labeled graph data across various datasets, with unified experimental settings and interface. Our benchmark has uncovered several important insights missed in previous research, and we believe these findings will be highly beneficial for future studies. We hope our open-source benchmark library will foster further advancements in this field. The code of the benchmark can be found in `https://github.com/eaglelab-zju/NoisyGL`.

## 1 Introduction

Many complex real-world systems can be represented as graph-structured data, including the citation network [19], biological networks [7], traffic networks [6], and social networks [9]. Graph Neural Networks (GNNs) have demonstrated substantial effectiveness in modeling graph data through a message-passing process that aggregates information from neighboring nodes [5]. Among the numerous applications of GNNs, node classification is the most thoroughly studied task, where GNNs are trained with the explicit assistance of semi-supervised node labels [1].

Although GNNs have achieved success, their performance in semi-supervised graph learning tasks is highly dependent on precise node labels, which are difficult to obtain in real-world scenarios [1]. For instance, in online social networks, the process of manually labeling millions of users is costly, and the labels often depend on unreliable user input [18]. Furthermore, graph data is vulnerable to adversarial label-flipping attacks [31]. Consequently, label noise is widespread in graph data. Research has demonstrated that label noise can significantly reduce the generalizability of machine learning models on computer vision and natural language processing scenarios [21]. In GNNs, the

---

*Sheng Zhou is the Corresponding Author

38th Conference on Neural Information Processing Systems (NeurIPS 2024) Track on Datasets and Benchmarks.

message-passing mechanism can further exacerbate this negative impact by propagating incorrect supervision from mislabeled nodes throughout the graph, leading to substantial results [18].

To address this challenge, an intuitive solution is to draw on the success of previous Learning with Label Noise (LLN) strategies and apply them to GNNs. However, these approaches are not always applicable to graph learning tasks due to the non-i.i.d nature, sparse labeling of graph data, and message-passing mechanism of GNNs[1]. All these factors make GNNs vulnerable to label noise and hinder traditional LLN methods from being directly applied to graph learning tasks.

In recent years, researchers have developed a series of Graph Neural Networks under Label Noise (GLN) methods to achieve robust graph learning in the presence of label noise. These methods succeeded greatly by adopting Loss regularization [14, 31, 11, 2], Robust training strategy [24], Graph structure augmentation [1, 18, 32], and contrastive learning [30, 10]. Despite the researcher's claim of the robustness of their proposed GLN methods, the comprehensive benchmark for evaluating these methods remains absent, bringing out the following problems: *1) Existing works utilize different datasets, noise types, rates, data splitting, and processing strategies, which makes it challenging to achieve a fair comparison. 2) Existing work lacks an empirical understanding regarding the impact of the graph structure itself on label noise—a critical distinction between LLN and GLN. 3) No existing work has thoroughly examined the applicability of traditional LLN methods to graph learning problems.* These problems hinder us from gaining a comprehensive understanding of the progress in this field.

In this research, we present NoisyGL — the first comprehensive benchmark for graph neural networks under label noise. Our benchmark includes seventeen representative methods: ten GLN methods to assess their effectiveness and robustness on graphs with noisy labels, and seven LLN methods to evaluate their applicability in graph learning tasks. We employ standardized backbones and APIs, consistent data splitting, and processing strategies to ensure a fair comparison and allow users to construct their models or datasets with minimal effort easily. Besides performance and robustness evaluations, our benchmark supports multidimensional analysis, enabling researchers to explore the time efficiency of different methods and understand the influence of graph structure on the handling of label noise.

Through extensive experiments, we have the following key findings: 1) Simply applying LLN methods can't significantly improve GNNs' robustness to label noise. 2) Existing GLN methods can alleviate label noise in their applicable scenarios. 3) Pair noise is the most harmful label noise due to its misleading impact. 4) Negative effects of label noise can spread through the graph structure, especially in sparse graphs. 5) GLN methods involving graph structure augmentation effectively mitigate the spread effect of label noise. Our contributions can be summarized as follows:

- **Perform an in-depth review of the current research challenge.** In our study, we revisited and scrutinized the entire progression of GLN. We discovered that the lack of a thorough benchmark in this domain significantly hinders a deeper understanding.

- **Provide a comprehensive and user-friendly benchmark.** We present NoisyGL, the first comprehensive benchmark for GLN. In this benchmark, we have selected and implemented a variety of LLN and GLN methods and evaluated them across eight commonly used datasets under uniform experimental settings. Our benchmark library is available to the public on GitHub, intending to aid future research efforts.

- **Highlight the key findings and future opportunities.** Our study has resulted in several crucial findings that have the potential to greatly advance this field.

## 2 Formulations and Background

**Notations.** Consider a graph denoted by $\mathcal{G} = \{\mathcal{V}, \mathcal{E}\}$, where $\mathcal{V}$ is the set of $N$ nodes and $\mathcal{E}$ is the set of edges. $\mathbf{A} \in \mathbb{R}^{N \times N}$ is the adjacency matrix and $\mathbf{X} = [\mathbf{x}_1, \mathbf{x}_2, \cdots \mathbf{x}_N] \in \mathbb{R}^{N \times d}$ denotes node features matrix with dimension $d$. Each node has a ground truth label, the set of which is denoted by $\mathcal{Y} = \{y_1^*, y_2^*, \cdots, y_N^*\}$. We focus on the semi-supervised node classification problem, where only a small set of nodes $\mathcal{V}_L$ has assigned labels for training procedure, denoted as $\mathcal{Y}_L = \{y_1^*, y_2^*, \cdots, y_l^*\}$, where $l$ is the number of labeled nodes. The rest of them are unlabeled nodes, denoted as $\mathcal{V}_U = \mathcal{V} - \mathcal{V}_L$. Given $\mathbf{X}$ and $\mathbf{A}$, the goal of node classification is to train a classifier $f_\theta : (\mathbf{X}, \mathbf{A}) \rightarrow \hat{\mathbf{Y}}^{N \times c} = \{\hat{\mathbf{y}}_1, \hat{\mathbf{y}}_2, \cdots, \hat{\mathbf{y}}_N\}$ by minimizing $\mathcal{L}(f_\theta(\mathbf{X}, \mathbf{A}), \mathcal{Y}_L)$, where $c$ is

the number of classes, $\mathcal{L}$ is a loss function that measures the difference between the predicted labels and the ground truth labels. Typically $f_\theta$ is a well-designed Graph Neural Network(GNN). In this way, according to the Empirical Risk Minimization (ERM) principle, the well-trained classifier $f_{\theta^*}$ can generalize on unseen data $\mathcal{V}_U$.

However, the accessible labels $\mathcal{Y}_L$ can be corrupted by label noise in the real world, reducing the generalization ability of $f_{\theta^*}$. We denote the observed noisy labels as $\mathcal{Y}_N = \{\widetilde{y}_1, \widetilde{y}_2, \cdots \widetilde{y}_l\}$ and $\mathcal{Y}_L$ is their corresponding true labels. Typically, we consider two types of label noise, and here are their definitions:

**Uniform noise [21]** or symmetric noise assumes that the true label has a probability of $\epsilon \in (0,1)$ to be uniformly flipped to another class. Formally, for $\forall_{j \neq i}$, we have $p(\widetilde{y} = j | y^* = i) = \frac{\epsilon}{c-1}$, where $c$ represents the number of classes.

**Pair noise [28]** or pair flipping. Assumes that the true label can only be flipped to its corresponding pair class with a probability $\epsilon$. Formally, we have $p(\widetilde{y} = y_p | y^* = y_c) = \epsilon$ and $\forall_{j \neq y_p, y_c} p(\widetilde{y} = j | y^* = y_c) = 0$, where $y_p$ is the corresponding pair class of $y_c$.

The transition patterns of pair noise and uniform noise are illustrated in the Appendix. B.1. It is important to note that these noise types assume that the transition probability depends only on the observed and true labels, and is independent of instances. In real-world scenarios, label noise can be much more complex. We focus on the most frequently used noise types, leaving the investigation of the other noise types for future studies.

## 3 Benchmark Design

### 3.1 Datasets and Implementations

**Datasets.** We selected 8 node classification datasets widely used among different studies on graph label noise. These selected datasets come from different domains and exhibit different characteristics, enabling us to evaluate the generalizability of existing methods across a range of scenarios. Specifically, we use three classic citation datasets [19], namely Cora, Citeseer, Pubmed, and one author collaboration network DBLP [15], as well as two representative product co-purchase network datasets Amazon-Computers and Amazon-Photo [20]. Additionally, to analyze the model performance on heterophilous graphs, we include two representative social media network datasets BlogCatalog and Flickr [26]. We present detailed introductions to these datasets in Appendix C.1.

The splitting methods for training, validation, and test sets of the same dataset in different tasks are not always consistent. This necessitates a unified dataset splitting in our work to achieve fair comparisons. For three citation datasets, i.e. Cora Citeseer and Pubmed, we follow the most commonly used split in [31, 24, 11, 8]. For the author collaboration network DBLP, we follow the split as [1, 10]. For two co-purchase datasets Amazon-Computers and Amazon-Photo, we follow the split as [24]. For the social network datasets BlogCatalog and Flickr, we use the same split as [18]. In this study, we assume that the labels of both the training set and validation set have been affected by label noise. A clean test set is used to evaluate the model's performance.

**Label Corruption.** In each experiment, we first generate a label transition probability matrix based on the given noise rate and the definition of noise. Then, for each clean label in the training and validation set, we draw a noisy label from a categorical distribution according to its corresponding transition probability. These noisy labels are used in the subsequent training procedure.

**Implementations** We consider a collection of state-of-the-art GLN algorithms, including NRGNN [1], RTGNN [18], CP [31], D-GNN [14], RCNGLN [32], CLNode [24], PIGNN [2], UnionNET [11], CGNN [30], and CRGNN [10]; and a set of well-designed LLN methods, including two loss correction methods Forward and Backward correction [16], two robust loss functions APL [13] and SCE [22], two multi-network learning methods Coteaching [4] and JoCoR [23], and one noise adaptation layer method S-model [3]. We have rigorously reproduced all methods according to their papers and source code. More details about these algorithms and implementations can be found in the Appendix C.2.

## 3.2 Research Questions

In this study, we aim to answer the following research questions:

**RQ1: Can LLN methods be applied directly to graph learning tasks?**

**Motivation.** While recent studies have suggested that applying traditional Learning with Label Noise (LLN) methods directly to graph learning tasks may not yield the best results [1], a comprehensive analysis of this issue is still lacking. We aim to investigate the suitability of existing LLN methods for graph learning and understand the underlying reasons. By tackling this question, we can gain a clearer insight into the unique challenges posed by graph label noise and identify which LLN techniques remain effective in graph learning contexts.

**Experiment Design.** To investigate this question, we select various LLN methods referenced in the Section 3.1 and implement them on the GCN[8] backbone using unified hyper-parameters. We then perform node classification experiments on the most frequently used datasets, evaluating their effectiveness under various types and levels of label noise. For each method and dataset, we record the mean accuracy metrics and standard deviations over 10 runs. Data splitting is performed randomly with a consistent ratio. By comparing the performance of these LLN methods with GCN, we determine whether they enhance the robustness of the backbone.

**RQ2: How much progress has been made by existing GLN methods?**

**Motivation.** While numerous GLN methods have been introduced in the literature, previous studies have used varied datasets, data splits, and preprocessing techniques, complicating the fair comparison of these methods' performance. Furthermore, we notice that the majority of existing approaches have been tested on homophily graphs, leading to concerns about their relevance to heterophily graphs, which are also commonly encountered in practice. By investigating this issue, we seek to determine if current GLN methods effectively address graph label noise and to identify their shortcomings.

**Experiment Design.** To address this question, we select and implement many advanced GLN methods as described in Section 3.1. We then assess the performance of these methods using uniform datasets and experimental settings. For each method and dataset, we record the mean test accuracy and the standard deviation across 10 runs. Since many of these GLN methods use GCN as their foundation, we compare their performance with GCN to evaluate their robustness to label noise.

**RQ3: Are existing GLN methods computationally efficient?**

**Motivation.** The efficiency of GNNs in terms of computation is crucial for their use in real-world applications, and considering label noise can lead to higher computational expenses. While previous research has deeply investigated the accuracy, generalization, and robustness of the GLN method, it has failed to address the computational efficiency of these approaches. Therefore, it is important to evaluate the computational efficiency of different methods.

**Experiment design.** To answer this question, we recorded the runtime and test accuracy of various methods on different datasets under $30\%$ uniform noise. Specifically, for each method, we conducted 10 experiments for each method on each dataset. In each experiment, we measured the time when the model achieved the best accuracy on the validation set, considering it as the total runtime for that method. Through these experiments, we can assess whether the GLN methods strike a balance between computational efficiency and test accuracy.

**RQ4: Are existing GLN methods sensitive to noise rate?**

**Motivation.** Previous studies utilize different noise rates, making it difficult to fairly compare the performance of various methods. Therefore, it is essential to assess different methods using a consistent set of noise rates and to verify if existing GLN methods maintain stable performance across different noise levels.

**Experiment Design.** To investigate this question, we assess the performance of several GLN methods over varying noise levels using the same datasets and noise types. Specifically, we introduce label contamination with pair noise and uniform noise at rates of $10\%$, $20\%$, $30\%$, $40\%$, and $50\%$, while using clean labels as a baseline. We then train the GLN methods on these datasets following the experimental settings described in RQ2 and record the mean test accuracy and standard deviation from 10 runs. This evaluation allows us to determine the robustness of each method.

**RQ5: Are existing GLN methods robust to different types of label noise?**

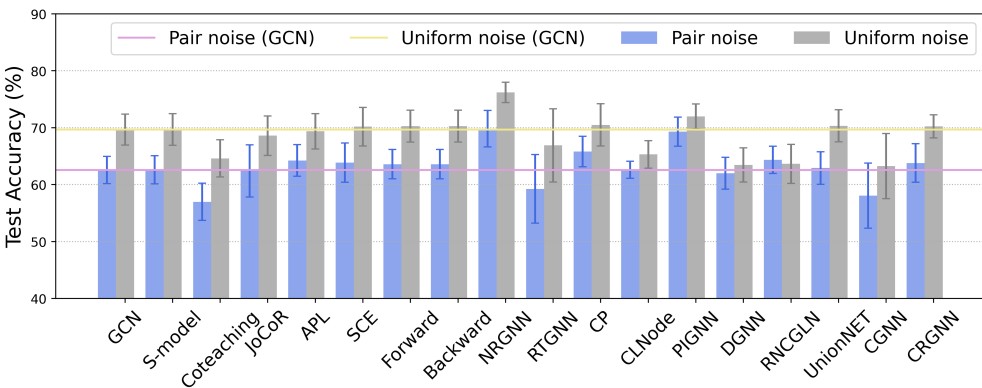

Figure 1: Test accuracy of LLN and GLN methods on DBLP dataset under 30% pair and uniform noise, respectively (10 Runs).

**Motivation.** Existing GLN methods have been developed with a variety of techniques and underlying assumptions, so they have unique strengths and weaknesses in managing different types of noise. It is crucial to identify which type of noise is most detrimental to graph learning and to understand the underlying reasons. Addressing this question will enhance our understanding of the specific scenarios in which each method excels and the distinct characteristics of different label noise types.

**Experiment design.** To tackle this question, we maintain a constant noise rate of 30% and apply both uniform and pair noise to the labels. Subsequently, we train GLN models on these noisy datasets following the experimental settings specified in RQ2 and record the mean test accuracy and standard deviation over 10 runs. This analysis enables us to determine the best method for each type of label noise and to comprehend the characteristics of various label noise types.

**RQ6: Good or bad? Revisiting the role of graph structure in label noise.**

**Motivation.** The graph structure plays a key role in distinguishing graph data from other types of data. The success of graph neural networks has largely relied on the neural message-passing mechanism, which aggregates information from neighboring nodes. However, in the presence of label noise, the messages propagated along the edges can have dual effects: on the one hand, label noise can negatively impact graph learning by spreading incorrect information; on the other hand, it can be alleviated by aligning with the majority label among the neighbors. Therefore, it is crucial to investigate whether the additional graph structure amplifies the effects of label noise and whether existing GLN methods can effectively address this challenge.

**Experiment Design.** To answer this question, we conducted comprehensive experiments on eighteen methods, including one GCN baseline, seven LLN methods, and ten GLN methods. Aiming to figure out how graph structure affects graph learning in the presence of label noise. Specifically, we recorded several metrics, including the Accuracy of Correctly Labeled Training nodes (ACLT), Accuracy of Incorrectly Labeled Training nodes (AILT), Accuracy of Unlabeled Correctly Supervised nodes (AUCS), Accuracy of Unlabeled Unsupervised nodes (AUU), and Accuracy of Unlabeled Incorrectly Supervised nodes (AUIS) under 30% uniform noise. Here, "correctly supervised," "incorrectly supervised," and "unsupervised" refer to unlabeled nodes that have a correctly labeled training node, an incorrectly labeled training node, and no labeled node in their neighborhood, respectively.

## 4  Experiment Results and Analyses

We present the performance of the eight methods, including vanilla GCN as a baseline, seven LLN methods with GCN backbone, and 10 GLN methods on eight datasets with different types and rates of label noise in Appendix A. Here are the key findings from the experimental results.

① **(RQ1) Most LLN methods do not significantly improve GNN robustness to label noise.** Table A1, A2, Figure 1 and Figure 2 reveal that most of the selected LLN methods do not substantially improve the performance of the GNN backbone when label noise is present. Mostly, the performance of these LLN methods remains statistically similar to the baseline. In some cases, the application of

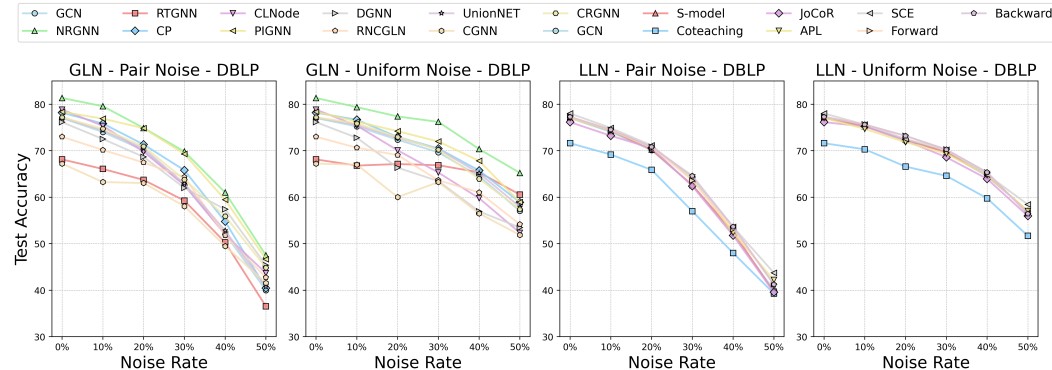

Figure 2: Test accuracy of GLN and LLN methods on DBLP dataset under different rate of pair and uniform noise (10 Runs).

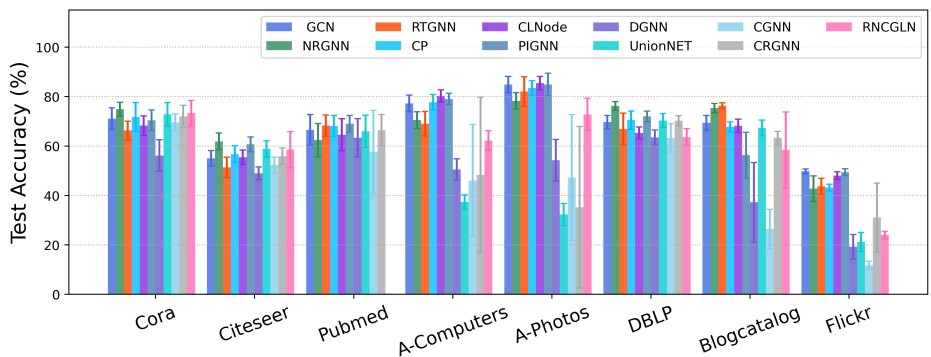

Figure 3: Test accuracy of GLN methods on all dataset under $30\%$ uniform noise (10 Runs).

additional LLN methods can lead to a worse result. Three LLN methods incorporate a noise transition matrix, i.e. S-model, Forward, and Backward correction, demonstrating performance similar to the baseline in most cases. Typically, these transition matrix-based methods learn a diagonal transition matrix, indicating their failure to learn the label transition pattern due to the scarcity of annotations. The multi-network learning methods Coteaching and JoCoR perform similarly to the baseline on sparse graphs but underperform on dense graphs. Notably, we find that two robust loss functions, Active Passive Loss (APL) and Symmetric Cross-Entropy (SCE), slightly enhance the robustness of the baseline model across most datasets. This improvement is likely due to their ability to reduce over-fitting on mislabeled samples, though it is limited by i.i.d. assumptions. Therefore, we conclude that merely applying LLN methods to GNNs does not achieve a label noise-robust graph learning solution. Detailed experimental results are available in Appendix A.

② (RQ2) Existing GLN methods can alleviate label noise in most cases, but this improvement is limited to specific applicable scenarios. As illustrated in Table A4, A5 and Figure 3, for each dataset, there is always at least one GLN method that consistently outperforms the baseline GCN across different types of label noise, indicating that these GLN methods are effective in mitigating the graph label noise problem. However, none of them consistently perform well across all datasets. For example, NRGNN significantly outperforms the baseline GCN in Cora, Citeseer, and DBLP, but not in other datasets. This observation suggests that existing GLN methods cannot generalize across different types of data. Additionally, we observed that on Flickr, all GLN methods fail to achieve better performance than the baseline, highlighting their deficiencies in dealing with highly heterophilous graphs. Detailed experimental results are available in Appendix A.

③ (RQ3) Some GLN methods are computationally inefficient. Table A6 demonstrates that multiple GLN methods, although effective at reducing label noise, often require substantial computational resources. Figure 4 indicates that some modern GLN techniques struggle to balance performance with computational efficiency. For instance, RNCGLN is the slowest, taking 66.8 times longer than

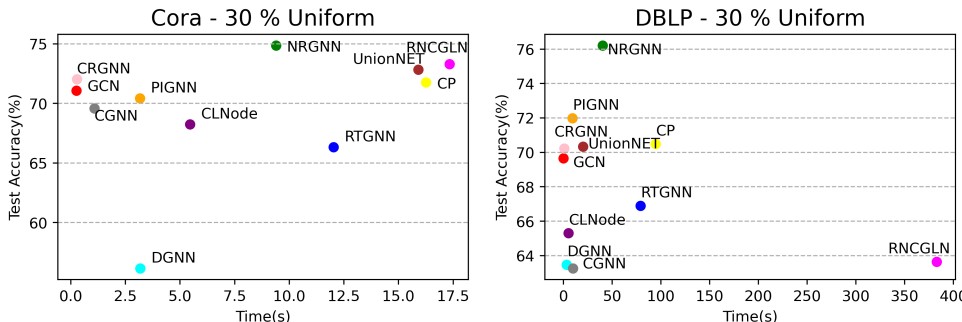

Figure 4: Time consumption and Test accuracy of different GLN methods on Cora and DBLP under 30% uniform noise (10 Runs).

Table 1: Misleading train accuracy of different methods under pair and uniform noise (10 Runs)

| Dataset (Avg. # Degree) | Noise type | GCN | JoCoR | APL | NRGNN | CLNode |
|---|---|---|---|---|---|---|
| **Cora (3.90)** | 50% **Uniform noise** | $78.49 \pm 9.87$ | $70.82 \pm 3.87$ | $77.29 \pm 12.35$ | $16.16 \pm 7.00$ | $68.46 \pm 17.64$ |
| | 50% **Pair Noise** | $94.53 \pm 5.51$ | $84.90 \pm 3.49$ | $92.93 \pm 6.46$ | $64.05 \pm 12.02$ | $88.03 \pm 6.22$ |
| **Citeseer (2.74)** | 50% **Uniform noise** | $98.78 \pm 1.15$ | $82.64 \pm 5.51$ | $93.21 \pm 5.13$ | $32.74 \pm 6.88$ | $79.54 \pm 15.19$ |
| | 50% **Pair Noise** | $98.54 \pm 2.46$ | $87.54 \pm 5.49$ | $96.19 \pm 3.85$ | $60.74 \pm 12.91$ | $82.23 \pm 8.78$ |
| **A-Computers (35.76)** | 50% **Uniform noise** | $19.68 \pm 6.59$ | $19.02 \pm 5.02$ | $23.49 \pm 6.44$ | $13.36 \pm 2.89$ | $15.27 \pm 5.00$ |
| | 50% **Pair Noise** | $75.30 \pm 7.91$ | $66.85 \pm 5.91$ | $72.42 \pm 9.90$ | $45.17 \pm 8.59$ | $64.34 \pm 11.66$ |
| **Blogcatalog (66.11)** | 50% **Uniform noise** | $29.13 \pm 9.95$ | $27.73 \pm 13.31$ | $42.23 \pm 16.14$ | $7.93 \pm 2.97$ | $31.67 \pm 9.33$ |
| | 50% **Pair Noise** | $72.50 \pm 15.54$ | $64.25 \pm 7.95$ | $73.39 \pm 13.65$ | $56.92 \pm 10.29$ | $64.70 \pm 11.64$ |

GCN on the Cora dataset and an astounding 2945.8 times longer on the DBLP dataset. Moreover, RNCGLN runs out of memory on the PubMed dataset, underscoring its inefficiency in memory usage. On the other hand, while the NRGNN method also consumes more time than GCN, it achieves a reasonable trade-off between performance and computational efficiency across both datasets. Detailed experimental results can be found in Appendix A.

④ **(RQ4) Most GLN methods can't ensure a high performance under severe noise.** Figure 2 depicts the performance of different GLN methods on the DBLP dataset under various types and levels of label noise. We observe that, in general, as the noise level increases, the test accuracy of each method decreases. This decrease is most pronounced for pair noise, where the test accuracy of all methods almost halves at 50% pair noise. Additionally, we noticed that RTGNN maintains relatively stable performance under uniform noise. Moreover, two methods, NRGNN and PIGNN, show better results than the baseline GNN over different noise levels and types on the DBLP dataset. Detailed experimental results are provided in Appendix A.

⑤ **(RQ5) Pair noise is more harmful to graph learning.** In our experiments, we consistently observed that pair noise poses the most significant threat to the generalization ability of models. We have an explanation for this finding: Recall the definition provided in Section 2. For uniform noise, the true label has a chance to flip to any other class, incorrect parameter updates caused by mislabeled instances can be partially compensated by other mislabeled instances. Pair noise, however, restricts the flipping class to a specific pair class. For classifiers, this type of pair flipping can be more misleading. After being fully trained, the classifier is more likely to over-fit the pair class. This becomes particularly harmful when node features propagate through message-passing mechanisms, which can lead to a more similar embedding within local neighbors and thus make them have a similar probability of being misclassified to their corresponding pair class. To validate our hypothesis, we conducted an empirical study. Specifically, we recorded the misleading train accuracy of five methods (including 1 GCN baseline, 2 LLN and 2 GLN) on four datasets under 50% pair and uniform noise. Here the misleading train accuracy represents the model's accuracy in making incorrect predictions to the misclassified classes. The experimental results (shown in Table 1) demonstrate that pair noise has the greatest impact, leading the model to overfit predict the mislabeled classes across different methods and datasets. Detailed experimental results are available in Appendix A.

Table 2: AUCS, AUU, AUIS of different methods on Cora and Amazon-Photos under 30% uniform noise (10 Runs)

| Dataset (Avg. # Degree) | Records | GCN | NRGNN | RTGNN | CP | CLNode | RNCGLN |
|---|---|---|---|---|---|---|---|
| **Cora (3.90)** | AUCS | $80.76 \pm 2.95$ | $83.11 \pm 3.16$ | $75.53 \pm 4.80$ | $80.85 \pm 3.46$ | $76.77 \pm 3.30$ | $77.06 \pm 3.30$ |
| | AUU | $71.99 \pm 3.44$ | $81.33 \pm 2.25$ | $73.44 \pm 4.95$ | $72.32 \pm 4.45$ | $67.11 \pm 5.11$ | $75.36 \pm 3.33$ |
| | AUIS | $51.55 \pm 6.53$ | $78.81 \pm 5.94$ | $69.50 \pm 8.06$ | $51.46 \pm 12.99$ | $43.86 \pm 7.48$ | $72.92 \pm 4.66$ |
| **A-Photos (31.13)** | AUCS | $92.21 \pm 2.44$ | $85.62 \pm 2.96$ | $89.98 \pm 2.10$ | $90.74 \pm 2.98$ | $93.08 \pm 1.97$ | $75.71 \pm 6.59$ |
| | AUU | $89.76 \pm 2.84$ | $84.29 \pm 2.79$ | $89.19 \pm 2.13$ | $88.85 \pm 3.20$ | $91.15 \pm 2.33$ | $74.40 \pm 6.73$ |
| | AUIS | $87.01 \pm 4.72$ | $82.46 \pm 5.86$ | $88.84 \pm 4.76$ | $86.34 \pm 3.99$ | $88.71 \pm 3.40$ | $71.08 \pm 7.89$ |

⑥ **(RQ6) Graph structure can amplify the negative effect of label noise.** From the experimental results in Table 2, we observed that in the sparse graph (Cora), AUIS and AUU exhibit a significant decrease compared to AUCS. Taking the performance of GCN on the Cora dataset as an example, this decrease is 36.17% and 10.85%, respectively. These results highlight the importance of proper supervision of neighboring nodes with correct annotations. Proper supervision of neighboring nodes with correct annotations significantly improves the classification accuracy of unlabeled nodes, while incorrect supervision of neighboring nodes severely reduces the classification accuracy of these nodes, even worse than when no neighborhood supervision is applied. Besides, our investigation also highlights the effectiveness of graph structure augmentation methods in mitigating the spread effect of label noise. According to Table 2, three methods, i.e. NRGNN, RTGNN, and RNCGLN, exhibit the smallest decrease in AUIS compared to AUCS and AUU among all methods. This indicates that they can effectively mitigate the spread effect of label noise. This phenomenon is even more pronounced in sparse graphs like Cora. One possible explanation can be easily drawn from the previous findings: The additional graph structure learning measures they adopted can lead to a denser graph structure used for predictions during the up-sampling process. Consequently, the classifier can rely on more references from the neighborhood, reducing its dependence on a small number of incorrectly labeled samples. Detailed experimental results are available in Appendix A.

⑦ **(RQ6) Sparse graphs are more vulnerable to the spread effect of label noise.** From Table 2 we see that the propagation effect of label noise can be very severe on sparse graphs with a relatively low average degree, like Cora, Citeseer, Pubmed, and DBLP, but not on dense graphs such as Amazon-Computers, Amazon-Photos, Blogcatalog and Flickr. The explanation for this observation is that unlabeled nodes on sparse graphs usually have only a limited number of annotated nodes in their neighborhood available for training. The prediction results of unlabeled nodes rely heavily on the annotated nodes in their neighborhood. However, if these nodes are incorrectly labeled, it will lead to erroneous learning of the embedding for the unlabeled nodes. In contrast, for dense graphs, the neighborhood of unlabeled nodes contains many annotated nodes that can serve as references. As a result, the classifier model is more likely to find correct supervision from these annotated nodes. This hypothesis is further supported by empirical evidence from Table 1, where we observe that compared to sparse graphs (such as Cora, Citeseer, and Pubmed), GCN is less susceptible to misleading on dense graphs like Blogcatalog and Amazon-Computers with a high average degree. Detailed experimental results are available in Appendix A.

## 5 Future directions

Based on the experimental results and analysis, we present several potential directions for the further development of the GLN.

**Designing widely applicable GLN approaches.** Our observations in finding ② reveal that most existing GLN methods cannot ensure consistently high performance across all scenarios. To address this problem, we need to explore three key questions: 1) What are the common properties of different graph datasets? 2) How can these common properties be utilized to enhance the robustness of GNNs against label noise? Our finding ⑥ indicated that enhancing graph structures can reduce the spread of label noise in graphs with varying densities, leading to the third question: 3) If identifying common properties is challenging, can we unify these features through data augmentation?

**Designing GLN approaches for various graph learning tasks.** Previous studies on GLN have predominantly focused on node classification tasks. However, the field of graph learning includes

other important tasks such as link prediction, edge property prediction, and graph classification. However, there is limited work on graph classification [27] and graph transfer learning [29] in the presence of label noise. Overall, research in other areas of graph learning, beyond node classification, is still in its early stages, and warrants further attention and exploration.

**Considering other types of label noise in graph learning.** Previous studies of GLN have mainly focused on pair noise and uniform noise. These noise types are instance-independent, assuming that the label corruption process is conditionally independent of node features when the true labels are given [21]. However, there exists another type of label noise—instance-dependent label noise—that is more realistic. In this case, the corruption probability depends on both the node features and the observed labels. However, none of the previous GLN studies have investigated this problem. Furthermore, unlike traditional machine learning tasks, graph learning involves additional graph structure, so the label noise model on graphs may also depend on graph topology. These issues are worth investigating, as they are more likely to occur in real-world scenarios.

## 6 Conclusions and Future work

In this research, we present NoisyGL, the first comprehensive benchmark designed for Graph Neural Networks under Label Noise (GLN) conditions. NoisyGL includes 7 prominent LLN and 10 GLN methods, allowing the community to fairly evaluate their effectiveness and robustness across various datasets. By using standardized backbones and APIs, consistent data splitting, and processing strategies, NoisyGL ensures a fair comparison and allows users to easily construct their own models or datasets with minimal effort. From this benchmark, we extract several key insights that are highly promising for the progression of this evolving field: Firstly, we point out that simply applying LLN methods cannot significantly improve the robustness of GNNs to label noise. Secondly, we found that existing GLN methods can alleviate label noise in their own applicable scenarios. In particular, pair noise emerges as the most harmful label noise due to its misleading effects. Finally, we discovered that negative effects of label noise can spread through the graph structure, especially in sparse graphs, and graph structure augmentation proves to be effective in mitigating the spread effect of label noise.

**Border Impacts and Limitations.** As NoisyGL provides a comprehensive benchmark for GNNs under label noise, we aim to attract more attention on the quality of graph data from the graph learning community, including the topology, node attributes and labels. However, NoisyGL also has some limitations that we aim to address in future work. Firstly, we aim to include a broader range of datasets to evaluate methods in different scenarios. While our current datasets are predominantly homogeneous graphs, we recognize that most GLN methods struggle with heterogeneous graphs, such as the Flickr network. Secondly, we hope to implement more GLN methods to gain a deeper understanding of the progress in the field. We will continuously update our repository to keep track of the latest advances in the field. We are also open to any suggestions and contributions that will improve the usability and effectiveness of our benchmark.

## 7 Acknowledgement

This work is supported by the National Natural Science Foundation of China (62106221, 62372408), Zhejiang Provincial Natural Science Foundation of China (Grant No: LTGG23F030005), Ningbo Natural Science Foundation (Grant No: 2022J183).

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

# A   Full experiment results

Table A1: Test accuracy of LLN methods (10 Runs)

| Dataset | Noise type | GCN | S-model | Coteaching | JoCoR | APL | SCE | Forward | Backward |
|---|---|---|---|---|---|---|---|---|---|
| **Cora** | 0% **clean** | $80.66 \pm 0.54$ | $80.74 \pm 0.58$ | $75.12 \pm 2.05$ | $80.51 \pm 0.61$ | $80.09 \pm 0.72$ | $82.08 \pm 0.80$ | $80.79 \pm 0.86$ | $80.78 \pm 0.84$ |
| | 10% **pair** | $76.44 \pm 2.48$ | $76.41 \pm 2.49$ | $69.69 \pm 2.01$ | $76.19 \pm 4.77$ | $76.70 \pm 2.61$ | $78.55 \pm 2.39$ | $76.91 \pm 2.94$ | $76.93 \pm 2.93$ |
| | 20% **pair** | $73.07 \pm 2.46$ | $73.04 \pm 2.32$ | $64.58 \pm 2.61$ | $71.53 \pm 6.82$ | $73.55 \pm 2.21$ | $73.68 \pm 1.94$ | $72.80 \pm 3.79$ | $73.12 \pm 3.24$ |
| | 30% **pair** | $65.36 \pm 5.54$ | $65.59 \pm 5.28$ | $56.87 \pm 4.40$ | $64.52 \pm 5.80$ | $65.97 \pm 5.13$ | $66.97 \pm 4.29$ | $65.96 \pm 6.17$ | $65.64 \pm 6.85$ |
| | 40% **pair** | $54.02 \pm 3.48$ | $54.03 \pm 3.81$ | $50.39 \pm 5.38$ | $55.26 \pm 5.43$ | $56.97 \pm 6.77$ | $54.56 \pm 4.44$ | $55.91 \pm 4.65$ | $55.57 \pm 4.29$ |
| | 50% **pair** | $44.15 \pm 8.52$ | $44.01 \pm 9.38$ | $41.15 \pm 6.03$ | $43.66 \pm 9.05$ | $42.84 \pm 8.07$ | $43.06 \pm 6.71$ | $43.32 \pm 8.77$ | $43.51 \pm 9.04$ |
| | 10% **uniform** | $78.58 \pm 2.04$ | $78.72 \pm 1.89$ | $71.06 \pm 2.75$ | $79.13 \pm 2.37$ | $79.10 \pm 1.25$ | $78.94 \pm 1.75$ | $78.29 \pm 1.97$ | $78.34 \pm 2.03$ |
| | 20% **uniform** | $75.92 \pm 1.49$ | $75.93 \pm 1.33$ | $66.68 \pm 4.16$ | $76.25 \pm 1.69$ | $76.00 \pm 1.60$ | $76.07 \pm 1.35$ | $75.91 \pm 1.56$ | $75.90 \pm 1.45$ |
| | 30% **uniform** | $71.06 \pm 4.39$ | $71.05 \pm 4.88$ | $61.22 \pm 5.88$ | $71.16 \pm 6.53$ | $72.37 \pm 4.00$ | $71.15 \pm 5.04$ | $72.47 \pm 4.81$ | $72.06 \pm 5.10$ |
| | 40% **uniform** | $67.88 \pm 3.73$ | $68.13 \pm 3.45$ | $59.19 \pm 2.97$ | $66.29 \pm 3.58$ | $68.24 \pm 3.58$ | $65.90 \pm 4.16$ | $68.01 \pm 3.96$ | $68.21 \pm 3.58$ |
| | 50% **uniform** | $54.42 \pm 4.72$ | $55.15 \pm 4.26$ | $47.28 \pm 6.24$ | $53.42 \pm 5.26$ | $55.13 \pm 4.96$ | $53.76 \pm 3.99$ | $53.82 \pm 5.50$ | $53.82 \pm 5.50$ |
| | 10% **random** | $78.19 \pm 1.98$ | $78.34 \pm 2.03$ | $71.34 \pm 2.99$ | $78.70 \pm 1.03$ | $78.50 \pm 1.65$ | $79.19 \pm 1.49$ | $78.75 \pm 1.36$ | $78.75 \pm 1.36$ |
| | 20% **random** | $74.27 \pm 3.42$ | $74.29 \pm 3.46$ | $67.14 \pm 2.80$ | $73.77 \pm 4.99$ | $74.76 \pm 3.90$ | $74.78 \pm 2.93$ | $74.57 \pm 3.40$ | $74.66 \pm 3.35$ |
| | 30% **random** | $69.72 \pm 3.49$ | $70.04 \pm 3.85$ | $62.09 \pm 4.06$ | $70.84 \pm 4.47$ | $70.67 \pm 4.17$ | $70.17 \pm 4.49$ | $70.01 \pm 5.24$ | $70.05 \pm 5.19$ |
| | 40% **random** | $62.62 \pm 3.25$ | $62.60 \pm 3.35$ | $54.50 \pm 3.64$ | $60.38 \pm 2.75$ | $62.51 \pm 2.75$ | $60.13 \pm 3.52$ | $61.39 \pm 5.90$ | $59.25 \pm 11.14$ |
| | 50% **random** | $55.59 \pm 6.92$ | $54.80 \pm 7.84$ | $46.47 \pm 5.68$ | $53.11 \pm 7.08$ | $55.05 \pm 7.28$ | $51.76 \pm 5.62$ | $54.36 \pm 8.65$ | $53.57 \pm 7.25$ |
| **Citeseer** | 0% **clean** | $69.01 \pm 0.72$ | $68.77 \pm 0.97$ | $58.46 \pm 3.95$ | $71.49 \pm 0.36$ | $69.36 \pm 0.91$ | $70.28 \pm 0.68$ | $69.55 \pm 0.86$ | $69.36 \pm 0.90$ |
| | 10% **pair** | $65.08 \pm 1.90$ | $64.74 \pm 2.15$ | $55.45 \pm 3.61$ | $68.24 \pm 2.03$ | $64.78 \pm 1.40$ | $65.43 \pm 2.07$ | $65.15 \pm 1.40$ | $65.06 \pm 1.50$ |
| | 20% **pair** | $58.22 \pm 3.29$ | $57.79 \pm 3.68$ | $50.69 \pm 4.65$ | $61.11 \pm 5.58$ | $56.65 \pm 4.35$ | $58.94 \pm 3.16$ | $58.30 \pm 3.44$ | $58.49 \pm 3.44$ |
| | 30% **pair** | $53.66 \pm 5.03$ | $53.41 \pm 5.26$ | $48.34 \pm 3.48$ | $56.30 \pm 5.14$ | $53.48 \pm 5.51$ | $54.61 \pm 4.53$ | $54.57 \pm 5.59$ | $54.62 \pm 5.19$ |
| | 40% **pair** | $43.47 \pm 4.89$ | $44.09 \pm 4.91$ | $42.05 \pm 3.73$ | $45.86 \pm 6.77$ | $43.85 \pm 4.31$ | $42.78 \pm 6.83$ | $44.26 \pm 5.46$ | $43.96 \pm 5.11$ |
| | 50% **pair** | $35.48 \pm 5.20$ | $36.03 \pm 5.01$ | $35.66 \pm 4.56$ | $36.50 \pm 5.87$ | $35.67 \pm 4.82$ | $35.82 \pm 5.46$ | $35.79 \pm 5.01$ | $35.87 \pm 4.80$ |
| | 10% **uniform** | $65.48 \pm 2.38$ | $65.46 \pm 2.13$ | $57.03 \pm 3.75$ | $69.06 \pm 2.33$ | $65.18 \pm 2.20$ | $66.47 \pm 2.27$ | $66.65 \pm 1.63$ | $66.36 \pm 1.78$ |
| | 20% **uniform** | $61.40 \pm 3.00$ | $61.10 \pm 3.11$ | $50.94 \pm 4.17$ | $65.92 \pm 2.52$ | $60.74 \pm 2.58$ | $62.53 \pm 2.87$ | $62.27 \pm 2.44$ | $62.35 \pm 2.60$ |
| | 30% **uniform** | $55.05 \pm 3.11$ | $54.68 \pm 3.26$ | $46.75 \pm 4.80$ | $57.14 \pm 3.22$ | $54.61 \pm 3.68$ | $57.00 \pm 2.87$ | $56.05 \pm 3.31$ | $55.81 \pm 3.59$ |
| | 40% **uniform** | $48.89 \pm 4.56$ | $49.30 \pm 4.34$ | $43.54 \pm 4.05$ | $53.43 \pm 5.48$ | $49.18 \pm 3.99$ | $49.07 \pm 3.70$ | $50.82 \pm 4.35$ | $50.93 \pm 4.35$ |
| | 50% **uniform** | $43.51 \pm 5.26$ | $43.61 \pm 5.18$ | $39.57 \pm 4.52$ | $45.46 \pm 5.96$ | $42.98 \pm 5.72$ | $42.49 \pm 5.52$ | $44.69 \pm 5.95$ | $44.74 \pm 5.61$ |
| | 10% **random** | $66.01 \pm 2.17$ | $65.92 \pm 2.39$ | $57.92 \pm 3.80$ | $69.54 \pm 2.05$ | $66.01 \pm 2.18$ | $67.03 \pm 1.55$ | $66.92 \pm 2.06$ | $67.02 \pm 2.08$ |
| | 20% **random** | $61.11 \pm 4.66$ | $61.14 \pm 3.92$ | $50.87 \pm 5.92$ | $66.21 \pm 3.84$ | $61.10 \pm 5.42$ | $62.76 \pm 3.73$ | $62.74 \pm 3.16$ | $62.08 \pm 4.16$ |
| | 30% **random** | $56.47 \pm 4.93$ | $56.28 \pm 5.15$ | $47.47 \pm 5.07$ | $59.05 \pm 4.75$ | $56.02 \pm 4.65$ | $57.19 \pm 3.98$ | $56.94 \pm 3.47$ | $56.93 \pm 3.50$ |
| | 40% **random** | $47.80 \pm 5.64$ | $48.05 \pm 5.72$ | $43.82 \pm 6.04$ | $51.35 \pm 5.92$ | $47.01 \pm 5.51$ | $47.66 \pm 5.61$ | $48.49 \pm 6.32$ | $49.31 \pm 5.69$ |
| | 50% **random** | $41.76 \pm 6.67$ | $42.50 \pm 6.10$ | $38.62 \pm 5.01$ | $43.29 \pm 6.33$ | $42.30 \pm 5.90$ | $39.16 \pm 6.33$ | $43.35 \pm 6.65$ | $41.56 \pm 6.33$ |
| **Pubmed** | 0% **clean** | $78.68 \pm 0.49$ | $78.72 \pm 0.59$ | $74.48 \pm 1.33$ | $70.58 \pm 2.65$ | $78.11 \pm 0.36$ | $79.11 \pm 0.38$ | $79.09 \pm 0.58$ | $79.08 \pm 0.57$ |
| | 10% **pair** | $74.49 \pm 3.06$ | $74.64 \pm 3.03$ | $70.73 \pm 5.86$ | $66.12 \pm 7.92$ | $74.20 \pm 2.83$ | $75.63 \pm 2.57$ | $74.90 \pm 3.51$ | $74.95 \pm 3.45$ |
| | 20% **pair** | $70.61 \pm 6.79$ | $70.84 \pm 6.68$ | $68.55 \pm 3.81$ | $61.84 \pm 7.35$ | $71.10 \pm 5.95$ | $72.13 \pm 5.24$ | $70.76 \pm 6.57$ | $70.96 \pm 6.44$ |
| | 30% **pair** | $62.91 \pm 5.49$ | $63.16 \pm 5.46$ | $60.45 \pm 7.23$ | $55.88 \pm 7.20$ | $64.28 \pm 3.84$ | $63.78 \pm 4.86$ | $62.68 \pm 5.53$ | $62.35 \pm 5.39$ |
| | 40% **pair** | $55.67 \pm 9.59$ | $55.61 \pm 9.77$ | $55.93 \pm 8.67$ | $49.76 \pm 7.44$ | $56.19 \pm 10.55$ | $51.44 \pm 6.89$ | $55.69 \pm 9.15$ | $55.73 \pm 9.13$ |
| | 50% **pair** | $42.99 \pm 9.12$ | $43.07 \pm 9.09$ | $43.16 \pm 7.39$ | $42.55 \pm 8.55$ | $42.88 \pm 8.34$ | $46.24 \pm 8.61$ | $43.49 \pm 8.03$ | $43.50 \pm 7.98$ |
| | 10% **uniform** | $74.61 \pm 2.04$ | $74.53 \pm 1.90$ | $71.36 \pm 2.58$ | $65.59 \pm 8.41$ | $74.16 \pm 2.27$ | $75.81 \pm 1.59$ | $75.57 \pm 1.85$ | $75.58 \pm 1.83$ |
| | 20% **uniform** | $70.26 \pm 3.66$ | $70.46 \pm 3.56$ | $68.90 \pm 2.89$ | $61.16 \pm 6.87$ | $70.94 \pm 3.40$ | $71.25 \pm 3.18$ | $69.75 \pm 3.90$ | $69.71 \pm 3.91$ |
| | 30% **uniform** | $66.53 \pm 6.23$ | $66.52 \pm 6.88$ | $64.38 \pm 7.27$ | $59.44 \pm 5.23$ | $67.56 \pm 4.96$ | $67.38 \pm 6.86$ | $65.62 \pm 6.41$ | $65.61 \pm 6.39$ |
| | 40% **uniform** | $57.86 \pm 4.98$ | $57.89 \pm 4.53$ | $57.50 \pm 5.89$ | $50.82 \pm 5.58$ | $58.59 \pm 5.47$ | $55.54 \pm 9.06$ | $56.52 \pm 5.21$ | $56.17 \pm 5.28$ |
| | 50% **uniform** | $52.73 \pm 6.42$ | $52.52 \pm 7.38$ | $50.54 \pm 6.79$ | $47.41 \pm 6.43$ | $50.46 \pm 8.97$ | $48.99 \pm 7.74$ | $52.38 \pm 7.22$ | $52.18 \pm 6.97$ |
| | 10% **random** | $73.79 \pm 2.37$ | $73.90 \pm 2.34$ | $69.83 \pm 3.08$ | $65.18 \pm 8.22$ | $73.74 \pm 2.42$ | $74.70 \pm 2.58$ | $73.96 \pm 2.65$ | $73.82 \pm 2.67$ |
| | 20% **random** | $72.49 \pm 1.69$ | $72.69 \pm 1.89$ | $69.13 \pm 3.04$ | $63.51 \pm 7.46$ | $72.46 \pm 1.98$ | $73.23 \pm 2.03$ | $71.61 \pm 2.16$ | $71.83 \pm 2.19$ |
| | 30% **random** | $66.53 \pm 2.29$ | $66.89 \pm 2.10$ | $61.76 \pm 3.43$ | $57.27 \pm 6.96$ | $67.27 \pm 2.52$ | $69.02 \pm 2.85$ | $65.97 \pm 2.92$ | $65.84 \pm 2.84$ |
| | 40% **random** | $56.98 \pm 8.35$ | $57.97 \pm 8.96$ | $56.89 \pm 10.81$ | $50.06 \pm 11.37$ | $57.57 \pm 7.85$ | $52.59 \pm 10.13$ | $55.98 \pm 8.76$ | $56.06 \pm 8.89$ |
| | 50% **random** | $46.24 \pm 9.08$ | $44.11 \pm 12.57$ | $47.90 \pm 8.05$ | $40.92 \pm 11.99$ | $43.91 \pm 11.86$ | $39.90 \pm 15.18$ | $46.05 \pm 8.32$ | $43.65 \pm 12.55$ |
| **A-Computers** | 0% **clean** | $84.73 \pm 0.82$ | $84.93 \pm 0.99$ | $80.95 \pm 3.52$ | $75.40 \pm 9.28$ | $84.97 \pm 0.93$ | $82.83 \pm 0.64$ | $75.95 \pm 15.16$ | $69.07 \pm 23.49$ |
| | 10% **pair** | $83.01 \pm 1.46$ | $83.55 \pm 1.95$ | $81.82 \pm 1.75$ | $79.03 \pm 4.64$ | $84.02 \pm 1.57$ | $81.80 \pm 1.07$ | $70.93 \pm 12.60$ | $60.89 \pm 26.81$ |
| | 20% **pair** | $77.62 \pm 4.47$ | $80.05 \pm 2.80$ | $77.21 \pm 5.79$ | $77.01 \pm 4.63$ | $79.38 \pm 4.41$ | $78.91 \pm 2.35$ | $74.34 \pm 9.41$ | $55.80 \pm 26.88$ |
| | 30% **pair** | $70.95 \pm 4.21$ | $73.81 \pm 5.91$ | $67.89 \pm 9.09$ | $66.82 \pm 4.70$ | $74.01 \pm 8.62$ | $72.85 \pm 2.19$ | $60.11 \pm 14.24$ | $32.31 \pm 24.74$ |
| | 40% **pair** | $61.26 \pm 9.47$ | $62.40 \pm 9.18$ | $51.04 \pm 15.23$ | $55.45 \pm 9.82$ | $64.82 \pm 11.10$ | $62.27 \pm 9.50$ | $43.61 \pm 15.94$ | $40.38 \pm 22.82$ |
| | 50% **pair** | $39.44 \pm 9.16$ | $43.41 \pm 8.77$ | $36.41 \pm 10.09$ | $41.91 \pm 11.75$ | $39.82 \pm 9.94$ | $42.20 \pm 8.67$ | $36.98 \pm 9.31$ | $21.85 \pm 11.29$ |
| | 10% **uniform** | $83.06 \pm 1.50$ | $84.10 \pm 1.14$ | $80.88 \pm 3.28$ | $78.41 \pm 5.14$ | $84.63 \pm 1.10$ | $80.64 \pm 2.14$ | $72.00 \pm 12.13$ | $53.29 \pm 33.87$ |
| | 20% **uniform** | $79.79 \pm 2.68$ | $81.24 \pm 2.04$ | $80.29 \pm 2.56$ | $77.41 \pm 4.01$ | $82.38 \pm 1.73$ | $78.41 \pm 2.54$ | $64.01 \pm 16.21$ | $56.35 \pm 26.87$ |
| | 30% **uniform** | $77.26 \pm 3.34$ | $79.42 \pm 3.17$ | $76.25 \pm 5.65$ | $70.01 \pm 11.03$ | $79.27 \pm 3.49$ | $77.01 \pm 2.82$ | $48.98 \pm 15.91$ | $30.23 \pm 28.75$ |
| | 40% **uniform** | $73.35 \pm 3.37$ | $77.54 \pm 3.24$ | $72.53 \pm 4.52$ | $69.83 \pm 7.72$ | $78.41 \pm 2.35$ | $74.91 \pm 3.70$ | $56.24 \pm 14.91$ | $61.63 \pm 25.12$ |
| | 50% **uniform** | $68.31 \pm 5.59$ | $74.38 \pm 4.56$ | $66.69 \pm 10.34$ | $64.38 \pm 8.34$ | $65.62 \pm 18.69$ | $74.81 \pm 2.61$ | $49.00 \pm 21.72$ | $38.33 \pm 29.75$ |
| | 10% **random** | $82.02 \pm 1.82$ | $81.58 \pm 1.30$ | $80.82 \pm 1.41$ | $78.22 \pm 2.53$ | $82.43 \pm 1.06$ | $80.54 \pm 1.57$ | $66.21 \pm 15.28$ | $46.55 \pm 35.77$ |
| | 20% **random** | $78.39 \pm 2.62$ | $80.00 \pm 2.00$ | $79.34 \pm 2.11$ | $76.64 \pm 5.23$ | $81.08 \pm 1.62$ | $78.10 \pm 2.74$ | $60.95 \pm 17.20$ | $44.63 \pm 28.51$ |
| | 30% **random** | $75.47 \pm 3.88$ | $77.06 \pm 4.00$ | $67.65 \pm 12.86$ | $68.36 \pm 10.66$ | $75.86 \pm 9.33$ | $75.99 \pm 2.40$ | $55.91 \pm 11.57$ | $21.62 \pm 21.24$ |
| | 40% **random** | $73.19 \pm 4.83$ | $74.33 \pm 3.73$ | $72.56 \pm 3.47$ | $70.86 \pm 5.47$ | $72.52 \pm 5.45$ | $74.34 \pm 3.03$ | $46.11 \pm 18.26$ | $43.28 \pm 32.59$ |
| | 50% **random** | $64.62 \pm 5.56$ | $66.63 \pm 4.56$ | $65.62 \pm 5.74$ | $57.20 \pm 11.97$ | $63.70 \pm 9.83$ | $65.49 \pm 7.97$ | $46.10 \pm 17.67$ | $43.24 \pm 28.48$ |

Table A2: Test accuracy of LLN methods (10 Runs)

| Dataset | Noise type | GCN | S-model | Coteaching | JoCoR | APL | SCE | Forward | Backward |
|---|---|---|---|---|---|---|---|---|---|
| **A-Photos** | 0% **clean** | 91.82 ± 0.69 | 92.05 ± 0.51 | 89.66 ± 2.21 | 77.14 ± 7.63 | 89.92 ± 1.74 | 91.04 ± 1.24 | 69.14 ± 12.01 | 68.34 ± 34.69 |
| | 10% **pair** | 89.83 ± 1.42 | 89.71 ± 1.50 | 89.47 ± 1.80 | 74.03 ± 7.45 | 88.08 ± 2.07 | 89.78 ± 1.41 | 68.64 ± 14.55 | 50.42 ± 39.364 |
| | 20% **pair** | 85.74 ± 2.86 | 85.47 ± 2.91 | 86.73 ± 3.25 | 76.04 ± 8.12 | 87.07 ± 2.30 | 86.74 ± 3.11 | 69.64 ± 16.08 | 61.26 ± 23.32 |
| | 30% **pair** | 79.26 ± 4.79 | 80.06 ± 5.25 | 74.56 ± 7.56 | 68.48 ± 8.62 | 82.39 ± 6.09 | 81.57 ± 4.96 | 67.29 ± 11.43 | 51.90 ± 25.85 |
| | 40% **pair** | 64.83 ± 6.28 | 63.05 ± 5.80 | 62.22 ± 11.82 | 57.33 ± 8.35 | 65.89 ± 11.32 | 64.30 ± 5.69 | 54.24 ± 10.74 | 44.17 ± 19.25 |
| | 50% **pair** | 44.87 ± 14.90 | 44.21 ± 14.45 | 41.36 ± 15.48 | 42.11 ± 13.98 | 45.28 ± 14.30 | 44.26 ± 13.12 | 42.07 ± 12.50 | 27.34 ± 18.84 |
| | 10% **uniform** | 89.42 ± 1.61 | 90.64 ± 1.48 | 89.91 ± 1.32 | 75.82 ± 7.95 | 89.52 ± 1.51 | 90.24 ± 0.91 | 77.44 ± 14.22 | 70.45 ± 34.88 |
| | 20% **uniform** | 88.02 ± 1.99 | 89.53 ± 2.39 | 88.61 ± 2.08 | 76.59 ± 9.10 | 88.40 ± 1.17 | 89.11 ± 1.64 | 64.41 ± 15.05 | 62.42 ± 31.57 |
| | 30% **uniform** | 84.86 ± 3.27 | 85.32 ± 4.17 | 82.78 ± 5.75 | 75.22 ± 8.24 | 86.17 ± 3.17 | 85.85 ± 4.42 | 65.13 ± 11.73 | 55.97 ± 32.95 |
| | 40% **uniform** | 80.02 ± 4.79 | 81.37 ± 6.43 | 80.47 ± 8.00 | 71.60 ± 8.55 | 80.38 ± 6.35 | 83.03 ± 5.76 | 65.63 ± 14.04 | 46.17 ± 26.38 |
| | 50% **uniform** | 75.18 ± 5.60 | 76.35 ± 5.90 | 73.10 ± 7.06 | 64.04 ± 10.51 | 76.17 ± 6.26 | 79.18 ± 4.67 | 53.02 ± 12.74 | 24.97 ± 15.83 |
| | 10% **random** | 88.05 ± 1.46 | 87.42 ± 3.45 | 88.20 ± 1.14 | 77.66 ± 4.26 | 88.88 ± 1.11 | 87.42 ± 2.07 | 72.67 ± 12.38 | 74.74 ± 22.32 |
| | 20% **random** | 86.82 ± 1.58 | 85.46 ± 5.68 | 86.72 ± 1.53 | 80.27 ± 7.36 | 88.24 ± 0.90 | 86.19 ± 2.79 | 61.02 ± 15.40 | 55.67 ± 33.49 |
| | 30% **random** | 82.23 ± 4.37 | 84.22 ± 3.38 | 77.84 ± 24.47 | 79.51 ± 7.69 | 85.27 ± 2.51 | 82.83 ± 5.31 | 63.60 ± 14.20 | 54.90 ± 33.14 |
| | 40% **random** | 76.32 ± 6.09 | 78.89 ± 3.95 | 76.05 ± 12.73 | 67.10 ± 10.96 | 80.74 ± 5.22 | 80.12 ± 4.00 | 61.81 ± 13.70 | 45.52 ± 25.57 |
| | 50% **random** | 70.69 ± 6.24 | 74.92 ± 6.32 | 66.85 ± 11.26 | 63.54 ± 8.04 | 69.74 ± 11.71 | 75.70 ± 5.23 | 59.77 ± 8.19 | 27.13 ± 15.82 |
| **DBLP** | 0% **clean** | 77.03 ± 0.35 | 77.11 ± 0.29 | 71.62 ± 2.47 | 76.15 ± 0.22 | 77.06 ± 0.44 | 78.03 ± 0.30 | 77.33 ± 0.21 | 77.30 ± 0.24 |
| | 10% **pair** | 74.04 ± 1.58 | 74.33 ± 1.48 | 69.18 ± 4.03 | 73.22 ± 1.84 | 74.40 ± 1.70 | 74.89 ± 1.43 | 74.45 ± 1.74 | 74.53 ± 1.66 |
| | 20% **pair** | 70.11 ± 1.40 | 70.25 ± 1.45 | 65.89 ± 2.42 | 70.57 ± 2.89 | 70.69 ± 1.84 | 71.15 ± 1.68 | 70.83 ± 1.61 | 70.84 ± 1.61 |
| | 30% **pair** | 62.56 ± 2.39 | 62.60 ± 2.48 | 56.98 ± 3.27 | 62.39 ± 4.59 | 64.25 ± 2.77 | 63.86 ± 3.44 | 63.59 ± 2.58 | 63.59 ± 2.58 |
| | 40% **pair** | 52.16 ± 7.86 | 52.50 ± 7.88 | 48.01 ± 6.58 | 51.78 ± 9.27 | 52.36 ± 7.61 | 53.61 ± 8.89 | 53.12 ± 7.25 | 53.65 ± 6.73 |
| | 50% **pair** | 39.99 ± 7.94 | 39.59 ± 7.79 | 39.28 ± 5.80 | 39.55 ± 9.60 | 42.19 ± 7.29 | 43.72 ± 7.67 | 41.30 ± 8.69 | 41.22 ± 8.73 |
| | 10% **uniform** | 75.24 ± 1.04 | 75.25 ± 1.01 | 70.31 ± 2.50 | 75.22 ± 1.38 | 74.69 ± 1.16 | 75.62 ± 1.18 | 75.65 ± 1.41 | 75.64 ± 1.44 |
| | 20% **uniform** | 72.37 ± 3.11 | 72.40 ± 3.09 | 66.59 ± 4.19 | 72.18 ± 4.09 | 71.84 ± 3.56 | 72.53 ± 2.48 | 73.20 ± 3.07 | 73.20 ± 3.07 |
| | 30% **uniform** | 69.66 ± 2.72 | 69.68 ± 2.79 | 64.60 ± 3.27 | 68.60 ± 3.46 | 69.37 ± 3.12 | 70.18 ± 3.39 | 70.27 ± 2.81 | 70.26 ± 2.80 |
| | 40% **uniform** | 64.53 ± 5.58 | 64.71 ± 5.56 | 59.76 ± 6.66 | 63.95 ± 5.66 | 64.76 ± 5.37 | 64.86 ± 5.02 | 65.33 ± 5.20 | 65.34 ± 5.19 |
| | 50% **uniform** | 57.05 ± 7.88 | 57.01 ± 7.65 | 51.69 ± 4.37 | 55.97 ± 9.33 | 57.04 ± 7.68 | 58.43 ± 8.09 | 56.50 ± 7.72 | 56.43 ± 7.77 |
| | 10% **random** | 75.40 ± 0.88 | 75.54 ± 0.86 | 69.46 ± 2.05 | 70.67 ± 1.83 | 75.95 ± 0.91 | 74.59 ± 1.34 | 75.33 ± 1.00 | 75.34 ± 1.00 |
| | 20% **random** | 72.50 ± 2.27 | 72.68 ± 2.23 | 67.23 ± 2.62 | 71.18 ± 3.03 | 72.92 ± 1.76 | 72.68 ± 2.55 | 73.07 ± 2.11 | 73.07 ± 2.11 |
| | 30% **random** | 66.60 ± 3.99 | 66.37 ± 4.40 | 61.99 ± 4.28 | 66.33 ± 4.99 | 66.81 ± 3.97 | 67.86 ± 4.05 | 67.03 ± 4.37 | 66.72 ± 5.02 |
| | 40% **random** | 62.76 ± 4.23 | 62.81 ± 4.27 | 55.37 ± 5.83 | 63.17 ± 2.90 | 63.61 ± 3.50 | 62.53 ± 4.66 | 63.82 ± 4.59 | 63.84 ± 4.61 |
| | 50% **random** | 54.26 ± 6.94 | 54.44 ± 7.00 | 50.77 ± 7.39 | 53.69 ± 6.44 | 54.87 ± 6.57 | 55.75 ± 7.07 | 55.69 ± 7.16 | 55.72 ± 7.18 |
| **Blogcatalog** | 0% **clean** | 76.52 ± 0.58 | 76.56 ± 0.73 | 61.89 ± 23.45 | 65.21 ± 0.81 | 75.89 ± 0.55 | 64.57 ± 7.43 | 75.58 ± 0.56 | 75.69 ± 0.57 |
| | 10% **pair** | 72.81 ± 1.53 | 74.08 ± 0.86 | 22.62 ± 17.94 | 65.08 ± 2.50 | 73.73 ± 1.04 | 61.31 ± 8.12 | 70.04 ± 5.72 | 62.96 ± 18.48 |
| | 20% **pair** | 67.09 ± 2.88 | 68.29 ± 3.32 | 27.43 ± 22.84 | 62.10 ± 3.31 | 71.92 ± 1.66 | 55.05 ± 5.40 | 65.06 ± 2.51 | 64.27 ± 6.36 |
| | 30% **pair** | 60.69 ± 1.76 | 60.51 ± 2.42 | 38.11 ± 22.91 | 57.13 ± 4.70 | 62.74 ± 4.94 | 48.91 ± 8.64 | 54.59 ± 9.00 | 47.02 ± 11.71 |
| | 40% **pair** | 46.74 ± 4.63 | 47.41 ± 5.01 | 30.88 ± 18.19 | 48.92 ± 3.37 | 46.75 ± 7.19 | 42.66 ± 4.86 | 40.25 ± 5.40 | 39.36 ± 5.21 |
| | 50% **pair** | 36.14 ± 6.74 | 35.42 ± 6.98 | 22.35 ± 8.35 | 35.96 ± 9.24 | 34.12 ± 5.82 | 33.99 ± 5.98 | 33.56 ± 7.55 | 33.77 ± 7.25 |
| | 10% **uniform** | 74.40 ± 1.03 | 75.22 ± 0.55 | 21.91 ± 17.32 | 65.69 ± 1.90 | 74.39 ± 0.60 | 62.28 ± 7.07 | 70.19 ± 5.59 | 71.09 ± 5.50 |
| | 20% **uniform** | 71.30 ± 1.23 | 71.69 ± 0.79 | 38.04 ± 27.78 | 63.16 ± 1.67 | 72.21 ± 1.46 | 58.10 ± 5.86 | 69.23 ± 4.88 | 67.95 ± 7.29 |
| | 30% **uniform** | 69.36 ± 2.99 | 70.06 ± 2.15 | 32.55 ± 23.26 | 63.16 ± 3.57 | 69.04 ± 4.20 | 54.59 ± 10.30 | 67.87 ± 3.95 | 68.17 ± 4.32 |
| | 40% **uniform** | 64.72 ± 2.36 | 65.72 ± 2.11 | 29.47 ± 20.66 | 59.76 ± 2.61 | 63.00 ± 2.48 | 55.87 ± 6.76 | 63.15 ± 2.43 | 61.57 ± 8.16 |
| | 50% **uniform** | 60.07 ± 3.59 | 60.83 ± 2.42 | 38.93 ± 19.36 | 56.83 ± 3.34 | 59.38 ± 4.03 | 53.67 ± 8.41 | 55.36 ± 5.79 | 52.94 ± 14.74 |
| | 10% **random** | 72.48 ± 1.44 | 73.53 ± 1.30 | 23.32 ± 18.40 | 66.04 ± 1.23 | 74.13 ± 0.76 | 63.80 ± 6.67 | 73.04 ± 0.92 | 68.40 ± 7.58 |
| | 20% **random** | 70.56 ± 1.26 | 71.83 ± 1.04 | 44.35 ± 28.47 | 64.90 ± 2.70 | 73.09 ± 1.08 | 61.47 ± 6.36 | 69.86 ± 5.03 | 64.19 ± 12.93 |
| | 30% **random** | 65.81 ± 2.07 | 66.02 ± 2.40 | 46.30 ± 25.23 | 61.22 ± 2.84 | 67.69 ± 3.15 | 56.70 ± 5.25 | 64.99 ± 4.10 | 62.65 ± 3.79 |
| | 40% **random** | 61.87 ± 3.95 | 61.92 ± 3.99 | 35.93 ± 24.68 | 56.84 ± 2.81 | 60.24 ± 7.00 | 51.42 ± 7.63 | 57.98 ± 7.47 | 50.33 ± 17.11 |
| | 50% **random** | 57.61 ± 3.48 | 56.82 ± 4.34 | 34.82 ± 17.99 | 52.75 ± 2.96 | 53.12 ± 8.59 | 51.51 ± 4.46 | 51.26 ± 7.86 | 46.39 ± 13.09 |
| **Flickr** | 0% **clean** | 58.02 ± 0.59 | 58.32 ± 0.59 | 49.29 ± 0.82 | 38.87 ± 18.39 | 58.52 ± 0.54 | 31.51 ± 2.21 | 28.76 ± 2.56 | 27.33 ± 6.41 |
| | 10% **pair** | 54.34 ± 1.21 | 54.48 ± 1.09 | 48.13 ± 1.54 | 50.95 ± 2.20 | 50.69 ± 1.24 | 24.84 ± 3.70 | 25.37 ± 2.84 | 26.79 ± 3.53 |
| | 20% **pair** | 51.50 ± 1.63 | 51.51 ± 1.43 | 45.71 ± 1.26 | 45.55 ± 1.26 | 46.97 ± 2.19 | 26.33 ± 4.68 | 23.88 ± 3.21 | 24.39 ± 4.15 |
| | 30% **pair** | 45.86 ± 3.45 | 45.47 ± 3.25 | 41.55 ± 2.67 | 41.20 ± 2.67 | 40.71 ± 2.69 | 23.36 ± 3.12 | 22.98 ± 1.52 | 25.40 ± 2.71 |
| | 40% **pair** | 38.95 ± 1.98 | 39.29 ± 2.00 | 35.64 ± 1.93 | 36.16 ± 2.20 | 36.39 ± 2.37 | 20.87 ± 4.13 | 20.75 ± 4.17 | 19.44 ± 4.38 |
| | 50% **pair** | 28.64 ± 3.09 | 28.77 ± 3.00 | 29.21 ± 2.62 | 27.91 ± 3.50 | 27.26 ± 2.55 | 19.75 ± 5.01 | 18.24 ± 3.94 | 17.17 ± 4.89 |
| | 10% **uniform** | 55.54 ± 0.48 | 55.95 ± 1.21 | 48.04 ± 1.36 | 50.96 ± 1.76 | 50.87 ± 2.77 | 25.65 ± 3.26 | 26.41 ± 1.93 | 28.18 ± 2.04 |
| | 20% **uniform** | 53.30 ± 1.98 | 53.15 ± 1.88 | 46.75 ± 2.65 | 47.44 ± 2.44 | 48.38 ± 2.64 | 24.10 ± 3.65 | 23.31 ± 3.42 | 21.64 ± 5.87 |
| | 30% **uniform** | 49.77 ± 0.94 | 49.42 ± 1.28 | 43.53 ± 2.32 | 43.61 ± 1.35 | 45.64 ± 2.09 | 22.69 ± 3.92 | 21.49 ± 3.20 | 23.93 ± 2.76 |
| | 40% **uniform** | 47.27 ± 2.44 | 47.09 ± 2.16 | 41.33 ± 1.23 | 41.39 ± 2.65 | 42.89 ± 4.28 | 22.11 ± 3.11 | 20.07 ± 1.70 | 21.68 ± 1.87 |
| | 50% **uniform** | 43.17 ± 2.01 | 43.08 ± 1.97 | 35.16 ± 8.56 | 38.39 ± 2.29 | 37.59 ± 3.63 | 20.30 ± 4.37 | 16.78 ± 4.43 | 18.37 ± 3.87 |
| | 10% **random** | 55.22 ± 1.06 | 55.46 ± 0.95 | 48.06 ± 1.57 | 50.85 ± 2.18 | 51.28 ± 2.34 | 25.27 ± 1.79 | 28.12 ± 2.00 | 29.64 ± 2.68 |
| | 20% **random** | 53.59 ± 0.94 | 53.26 ± 1.21 | 47.86 ± 0.68 | 48.02 ± 1.24 | 47.71 ± 1.88 | 29.14 ± 4.11 | 27.11 ± 3.00 | 25.10 ± 6.27 |
| | 30% **random** | 49.54 ± 1.28 | 49.27 ± 1.78 | 44.19 ± 2.55 | 44.17 ± 1.77 | 43.44 ± 2.55 | 24.32 ± 6.64 | 19.88 ± 5.00 | 22.74 ± 6.21 |
| | 40% **random** | 47.04 ± 1.62 | 46.53 ± 1.80 | 37.60 ± 9.78 | 41.68 ± 2.15 | 36.99 ± 5.80 | 19.59 ± 4.38 | 17.97 ± 3.25 | 18.86 ± 3.51 |
| | 50% **random** | 40.72 ± 2.87 | 40.40 ± 2.86 | 31.50 ± 8.09 | 36.17 ± 3.53 | 30.52 ± 4.94 | 19.01 ± 4.48 | 17.95 ± 2.61 | 19.00 ± 2.63 |

Table A3: Additional experiment results for LLN under 30% Uniform noise (10 Runs). ACLT denotes Accuracy of Correct Labeled Training nodes, AILT denotes Accuracy of Incorrect Labeled Training nodes, AUCS denotes Accuracy of Unlabeled Correct Supervised nodes, AUU denotes Accuracy of Unlabeled Unsupervised nodes, AUIS denotes Accuracy of Unlabeled Incorrect Supervised nodes.

| Dataset | Records | GCN | S-model | Coteaching | JoCoR | APL | SCE | Forward | Backward |
|---|---|---|---|---|---|---|---|---|---|
| **Cora** | ACLT | $98.53 \pm 0.93$ | $98.79 \pm 1.34$ | $98.81 \pm 1.13$ | $95.00 \pm 3.52$ | $97.50 \pm 2.09$ | $98.33 \pm 1.65$ | $98.70 \pm 0.96$ | $98.70 \pm 0.96$ |
| | AILT | $33.77 \pm 9.40$ | $30.92 \pm 6.95$ | $11.87 \pm 5.23$ | $41.59 \pm 10.40$ | $38.66 \pm 12.00$ | $22.90 \pm 13.41$ | $31.19 \pm 9.71$ | $31.19 \pm 9.71$ |
| | AUCS | $80.76 \pm 2.95$ | $81.12 \pm 3.10$ | $77.33 \pm 4.31$ | $80.02 \pm 3.30$ | $81.11 \pm 3.16$ | $79.75 \pm 3.35$ | $81.46 \pm 3.10$ | $81.46 \pm 3.10$ |
| | AUU | $71.99 \pm 3.44$ | $72.04 \pm 3.97$ | $64.88 \pm 6.03$ | $72.51 \pm 5.82$ | $73.41 \pm 3.98$ | $69.34 \pm 4.94$ | $72.89 \pm 3.95$ | $72.94 \pm 4.07$ |
| | AUIS | $51.55 \pm 6.53$ | $51.12 \pm 7.11$ | $34.72 \pm 8.61$ | $55.40 \pm 11.08$ | $56.25 \pm 7.93$ | $44.72 \pm 8.07$ | $53.56 \pm 6.87$ | $53.70 \pm 7.44$ |
| | Time | $0.18 \pm 0.29$ | $0.10 \pm 0.03$ | $2.58 \pm 1.02$ | $2.36 \pm 1.46$ | $0.13 \pm 0.03$ | $1.16 \pm 1.03$ | $1.75 \pm 0.16$ | $1.71 \pm 0.13$ |
| **Citeseer** | ACLT | $99.18 \pm 0.96$ | $99.42 \pm 0.83$ | $98.95 \pm 1.02$ | $90.11 \pm 2.50$ | $98.23 \pm 2.02$ | $99.88 \pm 0.37$ | $99.05 \pm 1.48$ | $98.93 \pm 1.44$ |
| | AILT | $1.39 \pm 2.65$ | $1.68 \pm 2.64$ | $1.98 \pm 1.87$ | $12.05 \pm 7.35$ | $3.10 \pm 3.39$ | $1.75 \pm 1.51$ | $2.48 \pm 2.70$ | $2.48 \pm 2.70$ |
| | AUCS | $74.70 \pm 3.49$ | $74.55 \pm 3.03$ | $72.39 \pm 3.22$ | $73.15 \pm 3.06$ | $75.44 \pm 3.33$ | $74.12 \pm 3.32$ | $75.00 \pm 2.56$ | $75.02 \pm 3.09$ |
| | AUU | $57.50 \pm 3.60$ | $57.19 \pm 3.42$ | $56.25 \pm 4.86$ | $57.08 \pm 2.90$ | $57.81 \pm 3.87$ | $57.19 \pm 3.49$ | $58.75 \pm 3.11$ | $58.96 \pm 3.19$ |
| | AUIS | $16.45 \pm 5.60$ | $15.37 \pm 5.67$ | $19.02 \pm 6.67$ | $18.53 \pm 7.35$ | $16.30 \pm 4.76$ | $16.80 \pm 5.65$ | $20.57 \pm 6.05$ | $20.91 \pm 5.57$ |
| | Time | $0.55 \pm 0.44$ | $0.74 \pm 0.69$ | $2.26 \pm 1.23$ | $1.54 \pm 1.25$ | $0.94 \pm 1.59$ | $3.79 \pm 1.47$ | $2.19 \pm 0.77$ | $2.40 \pm 0.86$ |
| **Pubmed** | ACLT | $97.08 \pm 2.61$ | $96.79 \pm 2.89$ | $97.67 \pm 2.50$ | $84.79 \pm 4.05$ | $96.41 \pm 2.41$ | $89.38 \pm 12.10$ | $97.55 \pm 3.24$ | $97.55 \pm 3.24$ |
| | AILT | $30.01 \pm 15.38$ | $25.21 \pm 18.01$ | $18.08 \pm 12.42$ | $30.72 \pm 10.11$ | $28.59 \pm 15.11$ | $35.51 \pm 15.07$ | $12.98 \pm 14.48$ | $12.46 \pm 13.81$ |
| | AUCS | $69.49 \pm 7.41$ | $70.11 \pm 7.61$ | $73.03 \pm 8.80$ | $60.31 \pm 11.59$ | $71.10 \pm 8.60$ | $68.16 \pm 5.19$ | $71.72 \pm 5.73$ | $71.72 \pm 5.73$ |
| | AUU | $59.47 \pm 7.87$ | $59.47 \pm 8.61$ | $60.53 \pm 9.37$ | $52.63 \pm 10.23$ | $61.05 \pm 7.53$ | $59.47 \pm 7.46$ | $59.47 \pm 7.87$ | $59.47 \pm 7.87$ |
| | AUIS | $27.06 \pm 21.05$ | $25.95 \pm 20.31$ | $34.80 \pm 28.88$ | $27.82 \pm 20.72$ | $28.97 \pm 18.29$ | $41.27 \pm 24.98$ | $22.86 \pm 15.84$ | $22.86 \pm 15.84$ |
| | Time | $0.21 \pm 0.06$ | $0.35 \pm 0.30$ | $2.25 \pm 1.33$ | $2.22 \pm 1.51$ | $0.35 \pm 0.13$ | $4.13 \pm 1.97$ | $2.12 \pm 0.74$ | $2.05 \pm 0.65$ |
| **A-Computers** | ACLT | $91.05 \pm 3.19$ | $92.33 \pm 2.22$ | $86.33 \pm 3.46$ | $84.85 \pm 3.60$ | $92.49 \pm 4.26$ | $91.04 \pm 4.03$ | $51.54 \pm 15.75$ | $37.62 \pm 39.19$ |
| | AILT | $68.90 \pm 9.98$ | $74.61 \pm 4.68$ | $67.96 \pm 6.57$ | $64.28 \pm 8.40$ | $72.38 \pm 9.26$ | $72.56 \pm 5.99$ | $45.95 \pm 14.45$ | $31.87 \pm 27.46$ |
| | AUCS | $83.43 \pm 2.44$ | $83.97 \pm 2.99$ | $80.30 \pm 4.47$ | $74.94 \pm 8.12$ | $84.59 \pm 2.65$ | $84.00 \pm 3.22$ | $47.10 \pm 11.71$ | $33.83 \pm 33.61$ |
| | AUU | $81.40 \pm 3.76$ | $82.77 \pm 3.19$ | $78.96 \pm 4.28$ | $72.77 \pm 8.48$ | $82.95 \pm 3.02$ | $82.37 \pm 3.29$ | $47.34 \pm 12.53$ | $33.87 \pm 32.51$ |
| | AUIS | $77.33 \pm 7.57$ | $81.24 \pm 5.33$ | $74.86 \pm 4.86$ | $69.32 \pm 9.02$ | $80.01 \pm 5.06$ | $79.58 \pm 5.87$ | $48.72 \pm 15.15$ | $31.86 \pm 29.47$ |
| | Time | $1.38 \pm 0.48$ | $1.25 \pm 0.40$ | $3.70 \pm 0.95$ | $3.47 \pm 1.04$ | $3.16 \pm 1.27$ | $3.12 \pm 1.09$ | $3.11 \pm 0.75$ | $4.97 \pm 2.37$ |
| **A-Photos** | ACLT | $91.74 \pm 2.96$ | $93.33 \pm 3.53$ | $89.70 \pm 4.48$ | $86.91 \pm 6.75$ | $92.50 \pm 7.11$ | $91.48 \pm 1.91$ | $71.55 \pm 15.92$ | $51.17 \pm 34.26$ |
| | AILT | $80.78 \pm 5.59$ | $81.99 \pm 7.92$ | $77.28 \pm 9.37$ | $72.73 \pm 9.64$ | $79.08 \pm 11.03$ | $83.32 \pm 6.83$ | $60.99 \pm 15.49$ | $47.84 \pm 29.61$ |
| | AUCS | $92.18 \pm 2.42$ | $91.67 \pm 3.15$ | $87.03 \pm 8.64$ | $83.16 \pm 7.40$ | $90.99 \pm 6.20$ | $92.92 \pm 1.09$ | $69.80 \pm 15.91$ | $50.49 \pm 36.54$ |
| | AUU | $89.73 \pm 2.81$ | $89.84 \pm 3.62$ | $85.44 \pm 7.99$ | $80.69 \pm 6.92$ | $88.91 \pm 6.24$ | $90.91 \pm 2.47$ | $67.41 \pm 15.81$ | $49.68 \pm 35.63$ |
| | AUIS | $87.01 \pm 4.72$ | $86.12 \pm 5.98$ | $82.50 \pm 7.82$ | $77.67 \pm 8.44$ | $85.82 \pm 8.87$ | $87.61 \pm 4.86$ | $63.56 \pm 15.53$ | $49.36 \pm 33.39$ |
| | Time | $0.90 \pm 0.32$ | $0.83 \pm 0.39$ | $3.40 \pm 0.81$ | $2.51 \pm 1.42$ | $2.91 \pm 1.18$ | $1.82 \pm 0.59$ | $2.68 \pm 0.46$ | $2.92 \pm 1.14$ |
| **DBLP** | ACLT | $98.34 \pm 1.71$ | $97.93 \pm 1.88$ | $98.60 \pm 2.26$ | $94.53 \pm 3.17$ | $97.59 \pm 3.28$ | $96.69 \pm 3.50$ | $97.58 \pm 1.24$ | $97.58 \pm 1.24$ |
| | AILT | $21.13 \pm 8.02$ | $22.55 \pm 8.58$ | $12.27 \pm 7.26$ | $35.82 \pm 7.49$ | $25.92 \pm 9.42$ | $28.77 \pm 8.09$ | $26.37 \pm 10.39$ | $26.37 \pm 10.39$ |
| | AUCS | $83.15 \pm 1.43$ | $83.28 \pm 1.30$ | $81.90 \pm 1.54$ | $82.52 \pm 2.03$ | $83.12 \pm 1.26$ | $83.11 \pm 1.51$ | $83.35 \pm 1.64$ | $83.38 \pm 1.61$ |
| | AUU | $74.10 \pm 2.64$ | $74.60 \pm 2.59$ | $70.47 \pm 5.51$ | $75.60 \pm 3.29$ | $74.85 \pm 2.70$ | $76.80 \pm 3.02$ | $75.58 \pm 2.40$ | $75.62 \pm 2.35$ |
| | AUIS | $55.65 \pm 9.47$ | $57.08 \pm 8.88$ | $48.87 \pm 11.69$ | $62.24 \pm 6.59$ | $57.99 \pm 9.69$ | $64.64 \pm 7.72$ | $59.80 \pm 8.13$ | $59.87 \pm 8.04$ |
| | Time | $0.13 \pm 0.03$ | $0.14 \pm 0.04$ | $2.41 \pm 1.32$ | $1.71 \pm 1.46$ | $0.19 \pm 0.07$ | $1.32 \pm 1.02$ | $1.65 \pm 0.14$ | $1.62 \pm 0.08$ |
| **Blogcatalog** | ACLT | $87.78 \pm 3.14$ | $88.25 \pm 4.84$ | $58.38 \pm 35.27$ | $81.27 \pm 5.43$ | $94.92 \pm 3.48$ | $63.98 \pm 8.80$ | $78.50 \pm 6.35$ | $78.11 \pm 6.82$ |
| | AILT | $58.52 \pm 6.32$ | $59.95 \pm 7.53$ | $37.31 \pm 16.42$ | $53.17 \pm 4.94$ | $46.59 \pm 10.66$ | $55.37 \pm 6.46$ | $65.54 \pm 5.15$ | $64.26 \pm 4.67$ |
| | AUCS | $71.37 \pm 3.05$ | $71.96 \pm 2.63$ | $44.65 \pm 25.60$ | $65.42 \pm 3.40$ | $71.06 \pm 4.20$ | $59.45 \pm 8.76$ | $68.73 \pm 4.84$ | $68.63 \pm 6.14$ |
| | AUU | $70.64 \pm 2.99$ | $71.23 \pm 2.50$ | $43.95 \pm 25.74$ | $64.57 \pm 3.45$ | $70.21 \pm 4.29$ | $58.64 \pm 8.74$ | $68.13 \pm 4.73$ | $68.10 \pm 5.96$ |
| | AUIS | $72.32 \pm 3.60$ | $72.92 \pm 3.15$ | $45.86 \pm 24.99$ | $66.18 \pm 3.41$ | $71.63 \pm 4.87$ | $60.15 \pm 8.94$ | $69.36 \pm 5.60$ | $69.75 \pm 5.52$ |
| | Time | $0.94 \pm 0.26$ | $1.05 \pm 0.35$ | $3.87 \pm 4.26$ | $3.32 \pm 1.03$ | $3.86 \pm 1.66$ | $5.95 \pm 0.23$ | $3.47 \pm 0.47$ | $2.80 \pm 0.45$ |
| **Flickr** | ACLT | $93.22 \pm 3.98$ | $89.85 \pm 6.21$ | $77.80 \pm 22.21$ | $76.89 \pm 1.79$ | $81.19 \pm 8.35$ | $35.80 \pm 6.44$ | $31.01 \pm 4.21$ | $30.41 \pm 7.36$ |
| | AILT | $20.27 \pm 5.10$ | $25.32 \pm 9.39$ | $21.54 \pm 7.61$ | $24.37 \pm 3.79$ | $22.35 \pm 5.49$ | $22.27 \pm 6.43$ | $25.70 \pm 4.85$ | $25.02 \pm 5.29$ |
| | AILMT | $68.88 \pm 10.70$ | $57.16 \pm 16.09$ | $58.87 \pm 18.91$ | $55.92 \pm 5.74$ | $58.79 \pm 11.60$ | $17.57 \pm 4.30$ | $15.95 \pm 1.97$ | $16.15 \pm 2.16$ |
| | AUCS | $51.31 \pm 2.98$ | $50.55 \pm 2.72$ | $42.35 \pm 10.53$ | $45.96 \pm 3.58$ | $45.69 \pm 4.07$ | $25.86 \pm 4.90$ | $27.27 \pm 3.34$ | $26.77 \pm 4.94$ |
| | AUU | $50.72 \pm 2.80$ | $50.09 \pm 2.60$ | $41.95 \pm 10.40$ | $45.45 \pm 3.43$ | $45.28 \pm 3.81$ | $25.30 \pm 5.00$ | $27.02 \pm 3.22$ | $26.63 \pm 4.73$ |
| | AUIS | $50.92 \pm 3.67$ | $49.94 \pm 2.87$ | $42.63 \pm 10.01$ | $45.23 \pm 3.69$ | $45.09 \pm 3.99$ | $24.81 \pm 4.58$ | $27.67 \pm 2.46$ | $27.09 \pm 4.25$ |
| | Time | $1.83 \pm 0.72$ | $1.40 \pm 0.67$ | $6.48 \pm 2.78$ | $4.21 \pm 1.01$ | $3.76 \pm 2.29$ | $1.13 \pm 1.84$ | $3.05 \pm 0.74$ | $2.79 \pm 0.65$ |

Table A4: Test accuracy of GLN methods (10 Runs). N/A indicates time or memory exceeded.

| Dataset | Noise type | GCN | NRGNN | RTGNN | CP | CLNode | PIGNN | DGNN | RNCGLN | UnionNET | CGNN | CR-GNN |
|---|---|---|---|---|---|---|---|---|---|---|---|---|
| Cora | Clean data | 80.66 ± 0.54 | 79.16 ± 0.74 | 72.31 ± 1.94 | 80.49 ± 1.02 | 80.90 ± 0.73 | 77.46 ± 1.28 | 72.41 ± 2.77 | 78.72 ± 1.02 | 81.61 ± 0.73 | 80.18 ± 1.28 | 81.23 ± 1.09 |
| | 10% pair | 76.44 ± 2.48 | 77.72 ± 1.03 | 70.28 ± 3.75 | 76.93 ± 2.50 | 75.84 ± 2.77 | 75.00 ± 1.89 | 68.27 ± 3.11 | 77.10 ± 1.94 | 77.26 ± 2.59 | 76.09 ± 2.47 | 76.49 ± 2.36 |
| | 20% pair | 73.07 ± 2.46 | 75.38 ± 1.78 | 68.47 ± 3.91 | 72.70 ± 3.30 | 71.84 ± 1.46 | 70.66 ± 1.38 | 64.55 ± 3.13 | 75.32 ± 2.78 | 72.96 ± 2.95 | 71.27 ± 2.69 | 73.03 ± 2.28 |
| | 30% pair | 65.36 ± 5.54 | 69.73 ± 4.82 | 63.44 ± 5.93 | 66.12 ± 5.63 | 64.55 ± 5.04 | 65.04 ± 5.83 | 58.49 ± 4.41 | 68.48 ± 3.51 | 66.29 ± 5.23 | 63.27 ± 5.71 | 65.37 ± 5.48 |
| | 40% pair | 54.02 ± 3.48 | 59.03 ± 4.36 | 54.86 ± 3.81 | 56.43 ± 5.19 | 55.03 ± 4.09 | 55.77 ± 4.10 | 48.71 ± 7.23 | 59.10 ± 6.28 | 56.20 ± 4.38 | 54.64 ± 6.17 | 55.67 ± 3.49 |
| | 50% pair | 44.15 ± 8.52 | 43.42 ± 6.99 | 43.69 ± 7.46 | 43.36 ± 7.24 | 44.16 ± 5.78 | 41.81 ± 7.92 | 42.41 ± 6.80 | 46.31 ± 12.67 | 43.74 ± 8.51 | 43.37 ± 7.78 | 44.53 ± 6.74 |
| | 10% uniform | 78.58 ± 2.04 | 78.28 ± 1.42 | 70.24 ± 4.59 | 78.16 ± 2.56 | 77.17 ± 2.17 | 76.07 ± 1.66 | 69.35 ± 4.00 | 78.01 ± 1.12 | 79.04 ± 2.09 | 77.89 ± 2.10 | 77.60 ± 2.04 |
| | 20% uniform | 75.92 ± 1.49 | 76.67 ± 2.30 | 69.27 ± 3.24 | 76.65 ± 1.54 | 73.47 ± 1.89 | 74.05 ± 1.99 | 64.67 ± 4.01 | 76.90 ± 1.21 | 76.08 ± 1.74 | 75.07 ± 1.81 | 75.77 ± 3.38 |
| | 30% uniform | 71.06 ± 4.39 | 74.86 ± 2.82 | 66.33 ± 3.72 | 71.76 ± 5.85 | 68.24 ± 3.95 | 70.43 ± 4.19 | 56.15 ± 6.40 | 73.29 ± 5.17 | 72.83 ± 4.74 | 69.57 ± 3.47 | 72.03 ± 4.48 |
| | 40% uniform | 67.88 ± 3.73 | 73.98 ± 2.53 | 66.02 ± 4.00 | 69.19 ± 3.31 | 63.82 ± 3.45 | 66.07 ± 4.48 | 52.06 ± 4.64 | 74.08 ± 3.90 | 68.92 ± 3.24 | 64.38 ± 5.45 | 67.98 ± 4.62 |
| | 50% uniform | 54.42 ± 4.72 | 64.90 ± 5.32 | 57.67 ± 5.68 | 55.03 ± 7.13 | 51.14 ± 5.97 | 55.28 ± 7.40 | 41.48 ± 5.81 | 67.32 ± 6.56 | 55.60 ± 4.13 | 54.08 ± 5.48 | 56.09 ± 4.53 |
| | 10% random | 78.19 ± 1.98 | 77.94 ± 1.48 | 71.12 ± 2.70 | 78.55 ± 2.17 | 77.01 ± 1.64 | 75.83 ± 1.71 | 69.83 ± 2.43 | 77.90 ± 1.32 | 79.09 ± 1.52 | 77.82 ± 1.59 | 78.22 ± 1.70 |
| | 20% random | 74.27 ± 3.42 | 75.58 ± 1.42 | 67.60 ± 3.50 | 75.51 ± 2.54 | 73.71 ± 3.38 | 73.05 ± 3.19 | 66.06 ± 4.18 | 75.43 ± 3.51 | 74.71 ± 3.68 | 72.63 ± 4.84 | 75.12 ± 3.65 |
| | 30% random | 69.72 ± 3.49 | 74.48 ± 2.86 | 66.84 ± 5.09 | 71.09 ± 4.08 | 67.61 ± 3.57 | 69.03 ± 3.46 | 57.16 ± 3.48 | 74.35 ± 5.29 | 71.02 ± 4.09 | 69.01 ± 4.11 | 70.40 ± 4.92 |
| | 40% random | 62.59 ± 3.24 | 70.93 ± 1.97 | 60.83 ± 4.12 | 64.05 ± 3.02 | 62.92 ± 3.60 | 62.03 ± 2.44 | 51.65 ± 3.80 | 69.15 ± 5.45 | 63.45 ± 2.87 | 59.47 ± 3.10 | 62.30 ± 3.87 |
| | 50% random | 55.59 ± 6.92 | 64.02 ± 6.68 | 55.24 ± 6.77 | 55.45 ± 4.93 | 50.56 ± 6.44 | 55.47 ± 6.12 | 40.27 ± 13.10 | 63.45 ± 8.03 | 56.51 ± 7.37 | 53.11 ± 6.73 | 54.12 ± 7.13 |
| Citeseer | Clean data | 69.01 ± 0.72 | 69.26 ± 1.49 | 61.61 ± 2.39 | 68.75 ± 1.34 | 68.38 ± 0.99 | 67.63 ± 1.55 | 64.18 ± 1.37 | 66.08 ± 3.72 | 71.97 ± 0.65 | 66.19 ± 1.51 | 67.40 ± 1.84 |
| | 10% pair | 65.08 ± 1.90 | 67.86 ± 1.29 | 58.00 ± 1.96 | 64.86 ± 1.83 | 63.98 ± 1.58 | 64.52 ± 3.35 | 56.37 ± 6.90 | 62.22 ± 4.37 | 68.36 ± 1.35 | 61.50 ± 2.16 | 63.73 ± 2.41 |
| | 20% pair | 58.22 ± 3.29 | 61.67 ± 4.73 | 52.07 ± 4.79 | 57.33 ± 3.00 | 58.43 ± 4.27 | 61.28 ± 4.03 | 51.53 ± 3.40 | 61.10 ± 7.20 | 61.50 ± 4.99 | 54.81 ± 4.04 | 55.58 ± 2.49 |
| | 30% pair | 53.66 ± 5.03 | 59.28 ± 6.21 | 48.39 ± 5.15 | 54.15 ± 5.44 | 53.19 ± 3.19 | 55.97 ± 4.79 | 46.76 ± 6.57 | 54.16 ± 4.56 | 58.15 ± 6.09 | 51.38 ± 4.19 | 52.45 ± 5.07 |
| | 40% pair | 43.47 ± 4.89 | 45.27 ± 6.66 | 37.36 ± 4.27 | 43.43 ± 5.31 | 42.51 ± 6.03 | 46.97 ± 5.87 | 40.69 ± 6.46 | 46.06 ± 8.13 | 44.72 ± 5.42 | 41.58 ± 4.27 | 43.55 ± 5.04 |
| | 50% pair | 35.48 ± 5.20 | 36.90 ± 5.35 | 33.67 ± 4.58 | 35.41 ± 5.92 | 34.41 ± 3.86 | 34.63 ± 5.49 | 34.48 ± 5.23 | 38.74 ± 10.06 | 37.18 ± 6.65 | 33.02 ± 5.12 | 35.90 ± 3.29 |
| | 10% uniform | 65.48 ± 2.38 | 67.52 ± 1.36 | 58.27 ± 3.26 | 65.65 ± 1.58 | 64.03 ± 2.64 | 66.61 ± 1.32 | 58.95 ± 2.15 | 64.76 ± 3.91 | 68.97 ± 1.79 | 61.48 ± 3.31 | 64.01 ± 2.41 |
| | 20% uniform | 61.40 ± 3.00 | 66.11 ± 1.75 | 55.98 ± 4.39 | 61.43 ± 2.96 | 59.62 ± 3.17 | 64.09 ± 1.69 | 52.24 ± 3.61 | 65.32 ± 4.39 | 66.50 ± 3.64 | 57.80 ± 3.30 | 59.86 ± 3.52 |
| | 30% uniform | 55.05 ± 3.11 | 61.87 ± 3.44 | 51.38 ± 4.11 | 56.72 ± 3.42 | 55.48 ± 2.92 | 60.71 ± 2.98 | 48.99 ± 2.57 | 58.56 ± 7.25 | 58.87 ± 3.25 | 52.31 ± 3.22 | 55.87 ± 3.38 |
| | 40% uniform | 48.89 ± 4.56 | 60.22 ± 3.64 | 51.14 ± 4.39 | 51.55 ± 5.27 | 49.08 ± 5.48 | 54.65 ± 4.29 | 40.53 ± 6.10 | 61.23 ± 8.80 | 55.78 ± 5.18 | 45.33 ± 4.28 | 47.63 ± 3.94 |
| | 50% uniform | 43.51 ± 5.26 | 54.57 ± 4.99 | 43.19 ± 3.19 | 44.93 ± 6.30 | 43.82 ± 6.02 | 50.09 ± 6.06 | 35.22 ± 6.90 | 55.43 ± 6.34 | 46.18 ± 5.54 | 40.72 ± 6.12 | 42.03 ± 5.10 |
| | 10% random | 66.01 ± 2.17 | 68.32 ± 2.40 | 59.59 ± 3.25 | 65.61 ± 2.62 | 64.86 ± 2.89 | 65.96 ± 1.56 | 58.35 ± 3.34 | 70.49 ± 1.05 | 69.37 ± 2.32 | 63.12 ± 2.22 | 64.68 ± 2.08 |
| | 20% random | 61.11 ± 4.66 | 67.02 ± 2.84 | 57.50 ± 3.69 | 61.44 ± 4.25 | 60.06 ± 3.06 | 63.39 ± 2.38 | 51.72 ± 6.41 | 68.99 ± 1.29 | 66.96 ± 4.28 | 56.33 ± 5.24 | 59.20 ± 5.01 |
| | 30% random | 56.44 ± 4.94 | 63.33 ± 2.19 | 52.57 ± 5.02 | 57.71 ± 4.44 | 55.24 ± 3.41 | 59.43 ± 4.42 | 49.39 ± 3.29 | 66.12 ± 3.72 | 60.85 ± 4.54 | 51.09 ± 4.89 | 53.92 ± 4.99 |
| | 40% random | 47.80 ± 5.64 | 59.98 ± 4.11 | 47.70 ± 3.11 | 48.55 ± 5.95 | 47.01 ± 6.10 | 54.88 ± 4.87 | 41.63 ± 5.58 | 62.23 ± 8.47 | 53.42 ± 6.23 | 44.54 ± 4.87 | 48.62 ± 6.48 |
| | 50% random | 41.76 ± 6.67 | 52.43 ± 8.79 | 43.05 ± 5.49 | 42.39 ± 7.26 | 41.90 ± 4.25 | 49.81 ± 5.27 | 36.49 ± 4.11 | 53.72 ± 8.85 | 45.76 ± 7.98 | 37.18 ± 5.37 | 41.36 ± 5.21 |
| Pubmed | Clean data | 78.68 ± 0.49 | 75.93 ± 1.16 | 75.73 ± 2.98 | 76.91 ± 1.04 | 78.31 ± 0.69 | 77.23 ± 0.49 | 76.14 ± 0.93 | N/A | 78.94 ± 0.37 | 70.55 ± 18.48 | 77.73 ± 0.59 |
| | 10% pair | 74.49 ± 3.06 | 68.92 ± 6.19 | 73.47 ± 3.41 | 73.21 ± 2.41 | 73.43 ± 3.99 | 74.51 ± 2.19 | 71.36 ± 3.22 | N/A | 74.62 ± 3.05 | 66.60 ± 17.41 | 74.42 ± 2.51 |
| | 20% pair | 70.61 ± 6.79 | 64.52 ± 6.90 | 68.42 ± 8.95 | 70.26 ± 5.21 | 69.59 ± 7.12 | 72.13 ± 5.03 | 67.23 ± 3.97 | N/A | 71.39 ± 6.57 | 62.96 ± 17.13 | 70.53 ± 4.88 |
| | 30% pair | 62.91 ± 5.49 | 57.22 ± 5.93 | 59.04 ± 6.65 | 62.67 ± 4.62 | 62.25 ± 5.23 | 66.75 ± 3.88 | 58.36 ± 4.87 | N/A | 62.77 ± 4.97 | 56.01 ± 14.88 | 62.91 ± 5.47 |
| | 40% pair | 55.67 ± 9.59 | 53.03 ± 7.33 | 52.44 ± 10.05 | 55.52 ± 8.86 | 55.67 ± 9.31 | 59.81 ± 8.60 | 54.71 ± 6.29 | N/A | 56.30 ± 8.13 | 51.31 ± 13.87 | 58.92 ± 6.61 |
| | 50% pair | 42.99 ± 9.12 | 39.40 ± 10.32 | 45.24 ± 8.91 | 45.67 ± 8.32 | 42.24 ± 8.27 | 42.70 ± 9.55 | 43.51 ± 6.93 | N/A | 42.98 ± 8.58 | 41.81 ± 11.41 | 43.89 ± 6.26 |
| | 10% uniform | 74.61 ± 2.04 | 69.54 ± 5.72 | 71.45 ± 4.12 | 73.67 ± 1.74 | 73.28 ± 2.13 | 74.55 ± 1.29 | 71.75 ± 1.34 | N/A | 75.56 ± 1.65 | 64.92 ± 18.30 | 74.58 ± 2.10 |
| | 20% uniform | 70.26 ± 3.66 | 68.67 ± 6.74 | 68.55 ± 4.38 | 70.73 ± 3.02 | 67.67 ± 3.80 | 71.80 ± 2.38 | 65.19 ± 4.38 | N/A | 70.24 ± 3.87 | 59.76 ± 17.22 | 70.50 ± 3.65 |
| | 30% uniform | 66.53 ± 6.23 | 62.37 ± 6.79 | 68.30 ± 4.31 | 68.11 ± 4.27 | 64.56 ± 6.50 | 69.02 ± 3.34 | 63.29 ± 7.74 | N/A | 65.93 ± 6.56 | 57.73 ± 16.75 | 66.42 ± 6.35 |
| | 40% uniform | 57.86 ± 4.98 | 57.10 ± 4.79 | 62.22 ± 5.34 | 59.84 ± 4.94 | 57.02 ± 7.49 | 62.72 ± 5.45 | 58.03 ± 5.13 | N/A | 57.45 ± 4.49 | 47.69 ± 12.59 | 58.28 ± 6.60 |
| | 50% uniform | 52.73 ± 6.42 | 53.40 ± 7.20 | 53.47 ± 9.12 | 53.84 ± 8.70 | 49.56 ± 8.32 | 55.16 ± 7.33 | 53.93 ± 5.55 | N/A | 52.04 ± 7.06 | 44.69 ± 11.83 | 53.73 ± 6.08 |
| | 10% random | 73.79 ± 2.37 | 65.99 ± 9.76 | 70.62 ± 5.67 | 72.70 ± 2.19 | 72.84 ± 3.37 | 73.93 ± 1.74 | 71.14 ± 2.55 | N/A | 74.06 ± 2.79 | 63.09 ± 18.92 | 73.45 ± 1.79 |
| | 20% random | 72.49 ± 1.69 | 68.43 ± 8.09 | 70.77 ± 2.13 | 71.86 ± 2.69 | 70.53 ± 2.49 | 71.93 ± 1.68 | 67.68 ± 3.26 | N/A | 73.12 ± 1.68 | 63.19 ± 17.14 | 72.12 ± 1.48 |
| | 30% random | 66.52 ± 2.29 | 62.38 ± 6.99 | 66.72 ± 7.14 | 68.17 ± 2.64 | 66.20 ± 2.74 | 70.50 ± 2.26 | 63.42 ± 5.39 | N/A | 65.60 ± 2.35 | 59.76 ± 15.18 | 69.48 ± 3.39 |
| | 40% random | 56.98 ± 8.35 | 51.32 ± 6.43 | 57.27 ± 14.75 | 59.18 ± 8.14 | 55.27 ± 15.68 | 64.81 ± 7.37 | 54.89 ± 11.11 | N/A | 56.36 ± 9.10 | 48.42 ± 15.98 | 61.51 ± 7.83 |
| | 50% random | 46.24 ± 9.08 | 42.60 ± 10.42 | 45.96 ± 10.76 | 47.35 ± 11.26 | 44.11 ± 14.08 | 48.59 ± 14.18 | 45.64 ± 8.83 | N/A | 43.20 ± 11.41 | 43.27 ± 10.73 | 49.01 ± 8.70 |
| A-Computers | Clean data | 84.73 ± 0.82 | 74.40 ± 2.22 | 66.68 ± 3.33 | 83.86 ± 0.86 | 84.28 ± 0.85 | 81.50 ± 1.06 | 60.23 ± 5.71 | 73.65 ± 1.64 | 37.71 ± 1.97 | 47.48 ± 25.39 | 50.49 ± 40.36 |
| | 10% pair | 83.01 ± 1.46 | 73.88 ± 3.77 | 70.83 ± 2.94 | 81.70 ± 1.19 | 82.20 ± 1.37 | 82.38 ± 1.55 | 56.80 ± 7.62 | 72.59 ± 2.12 | 37.90 ± 2.14 | 43.40 ± 16.71 | 55.85 ± 36.29 |
| | 20% pair | 77.62 ± 4.47 | 70.10 ± 5.98 | 64.89 ± 7.35 | 76.92 ± 4.64 | 77.77 ± 4.59 | 77.77 ± 6.35 | 55.38 ± 5.16 | 67.93 ± 3.20 | 38.05 ± 2.26 | 41.01 ± 19.36 | 48.03 ± 31.61 |
| | 30% pair | 70.95 ± 4.21 | 67.00 ± 2.65 | 61.40 ± 10.02 | 70.49 ± 4.59 | 73.96 ± 4.61 | 72.22 ± 5.80 | 48.11 ± 7.24 | 63.62 ± 3.93 | 37.33 ± 2.18 | 40.55 ± 21.80 | 39.18 ± 31.03 |
| | 40% pair | 60.92 ± 10.03 | 56.43 ± 14.29 | 50.20 ± 16.45 | 61.08 ± 10.15 | 60.01 ± 12.62 | 60.01 ± 11.62 | 40.71 ± 8.98 | 48.07 ± 14.57 | 31.39 ± 12.72 | 37.31 ± 18.55 | 36.62 ± 24.52 |
| | 50% pair | 39.23 ± 9.60 | 38.35 ± 10.68 | 36.66 ± 13.49 | 38.34 ± 6.33 | 38.69 ± 11.58 | 38.84 ± 8.31 | 26.84 ± 13.37 | 33.65 ± 12.41 | 22.61 ± 17.74 | 28.54 ± 13.79 | 26.44 ± 18.60 |
| | 10% uniform | 83.06 ± 1.50 | 73.77 ± 2.40 | 56.93 ± 23.29 | 81.30 ± 2.06 | 83.74 ± 1.29 | 83.28 ± 1.29 | 57.40 ± 7.00 | 71.62 ± 1.77 | 37.53 ± 1.81 | 45.72 ± 24.90 | 51.85 ± 33.73 |
| | 20% uniform | 79.79 ± 2.68 | 70.32 ± 4.95 | 62.58 ± 20.17 | 79.34 ± 3.35 | 81.55 ± 2.48 | 81.45 ± 2.44 | 52.25 ± 7.38 | 67.59 ± 5.16 | 37.92 ± 2.76 | 43.56 ± 20.07 | 48.38 ± 31.49 |
| | 30% uniform | 77.26 ± 3.34 | 70.47 ± 3.45 | 69.04 ± 4.96 | 77.74 ± 3.13 | 80.31 ± 2.41 | 79.04 ± 2.28 | 50.53 ± 4.26 | 62.22 ± 3.97 | 37.34 ± 2.94 | 46.07 ± 22.63 | 48.29 ± 31.43 |
| | 40% uniform | 73.78 ± 3.27 | 65.82 ± 7.04 | 60.52 ± 18.90 | 73.81 ± 3.08 | 76.85 ± 3.45 | 76.23 ± 3.48 | 48.98 ± 6.32 | 62.25 ± 5.59 | 30.89 ± 10.88 | 46.03 ± 21.39 | 46.49 ± 30.39 |
| | 50% uniform | 67.94 ± 5.05 | 61.69 ± 8.57 | 62.48 ± 4.20 | 69.01 ± 3.90 | 75.79 ± 2.01 | 72.22 ± 2.85 | 42.59 ± 12.96 | 54.84 ± 8.30 | 31.20 ± 11.16 | 43.40 ± 16.86 | 38.63 ± 25.90 |
| | 10% random | 82.90 ± 0.84 | 74.43 ± 2.85 | 68.79 ± 4.22 | 82.52 ± 1.81 | 83.48 ± 1.26 | 81.75 ± 1.77 | 55.92 ± 8.31 | 68.97 ± 2.82 | 30.62 ± 12.21 | 49.72 ± 25.80 | 56.34 ± 36.10 |
| | 20% random | 80.19 ± 3.79 | 71.52 ± 4.48 | 72.04 ± 3.23 | 78.78 ± 2.00 | 81.14 ± 2.15 | 80.28 ± 2.29 | 54.64 ± 5.05 | 67.49 ± 3.26 | 32.99 ± 10.65 | 53.33 ± 22.68 | 53.45 ± 34.11 |
| | 30% random | 75.79 ± 4.22 | 68.32 ± 7.22 | 70.07 ± 7.28 | 74.75 ± 5.27 | 75.79 ± 8.24 | 75.17 ± 5.59 | 50.73 ± 6.15 | 62.74 ± 7.95 | 30.76 ± 12.31 | 52.05 ± 20.16 | 49.17 ± 31.79 |
| | 40% random | 72.87 ± 3.86 | 63.84 ± 9.49 | 66.39 ± 6.75 | 72.35 ± 2.63 | 74.03 ± 7.41 | 75.14 ± 6.00 | 46.33 ± 5.40 | 56.25 ± 8.75 | 30.17 ± 12.85 | 47.62 ± 20.01 | 42.84 ± 28.15 |
| | 50% random | 64.73 ± 7.97 | 54.25 ± 7.00 | 53.52 ± 20.06 | 65.02 ± 8.34 | 65.64 ± 8.76 | 69.58 ± 5.98 | 45.59 ± 6.60 | 51.87 ± 11.60 | 31.11 ± 11.67 | 34.05 ± 22.38 | 39.78 ± 25.67 |
| A-Ratings | 0% clean | 39.61 ± 0.15 | 37.99 ± 0.34 | 37.45 ± 0.69 | 40.11 ± 0.16 | 38.58 ± 1.00 | 39.22 ± 0.06 | 39.57 ± 0.44 | 35.76 ± 0.28 | 36.71 ± 0.03 | 35.31 ± 4.24 | 33.08 ± 4.99 |
| | 10% pair | 39.27 ± 0.54 | 37.34 ± 0.56 | 37.65 ± 0.63 | 39.65 ± 0.30 | 38.14 ± 1.04 | 38.82 ± 0.31 | 37.38 ± 0.61 | 34.68 ± 0.93 | 36.63 ± 0.66 | 35.32 ± 4.23 | 36.64 ± 0.36 |
| | 20% pair | 39.07 ± 0.45 | 37.89 ± 0.65 | 37.36 ± 0.78 | 38.82 ± 1.05 | 38.33 ± 0.94 | 38.72 ± 0.46 | 36.33 ± 1.37 | 33.74 ± 1.04 | 36.50 ± 1.03 | 37.03 ± 0.36 | 36.01 ± 1.60 |
| | 30% pair | 38.41 ± 0.67 | 37.36 ± 0.48 | 36.53 ± 0.78 | 38.59 ± 0.71 | 37.83 ± 0.52 | 37.88 ± 0.57 | 33.79 ± 2.36 | 31.34 ± 1.75 | 35.78 ± 1.82 | 36.11 ± 2.53 | 35.67 ± 3.19 |
| | 40% pair | 36.73 ± 1.42 | 36.65 ± 1.36 | 35.12 ± 1.37 | 36.82 ± 1.54 | 36.26 ± 3.54 | 36.86 ± 1.51 | 32.96 ± 1.09 | 30.55 ± 1.45 | 32.91 ± 4.25 | 33.36 ± 4.20 | 35.37 ± 3.88 |
| | 50% pair | 34.41 ± 2.09 | 32.54 ± 3.69 | 33.56 ± 1.27 | 34.41 ± 2.07 | 31.36 ± 4.14 | 34.00 ± 2.94 | 31.72 ± 2.38 | 29.27 ± 1.71 | 27.52 ± 1.20 | 27.00 ± 7.63 | 28.70 ± 3.59 |
| | 10% uniform | 39.01 ± 0.59 | 37.32 ± 0.30 | 37.43 ± 0.58 | 38.99 ± 0.93 | 38.21 ± 0.52 | 38.90 ± 0.28 | 36.30 ± 1.86 | 33.98 ± 0.78 | 36.84 ± 0.08 | 35.42 ± 4.27 | 36.76 ± 0.10 |
| | 20% uniform | 38.53 ± 0.53 | 37.04 ± 0.30 | 37.18 ± 0.72 | 38.76 ± 0.36 | 37.84 ± 0.75 | 38.46 ± 0.37 | 35.62 ± 0.56 | 32.29 ± 0.80 | 36.82 ± 0.09 | 35.41 ± 4.27 | 36.81 ± 0.00 |
| | 30% uniform | 37.73 ± 0.91 | 36.91 ± 0.35 | 36.74 ± 0.35 | 37.59 ± 0.98 | 37.30 ± 0.58 | 37.55 ± 0.78 | 34.34 ± 1.13 | 31.00 ± 0.61 | 36.84 ± 0.07 | 35.37 ± 4.26 | 36.80 ± 0.02 |
| | 40% uniform | 36.90 ± 1.35 | 36.41 ± 0.62 | 36.12 ± 1.23 | 37.06 ± 1.03 | 36.72 ± 0.81 | 36.98 ± 0.72 | 32.89 ± 1.64 | 28.40 ± 1.32 | 36.52 ± 0.74 | 34.44 ± 5.07 | 36.81 ± 0.00 |
| | 50% uniform | 36.75 ± 0.84 | 36.82 ± 0.21 | 36.11 ± 0.91 | 36.82 ± 0.73 | 36.79 ± 0.52 | 36.87 ± 0.59 | 30.54 ± 1.26 | 26.15 ± 0.95 | 36.61 ± 0.62 | 33.40 ± 5.50 | 36.81 ± 0.00 |
| | 10% random | 38.85 ± 0.54 | 37.89 ± 0.45 | 37.61 ± 0.47 | 39.19 ± 0.48 | 37.86 ± 1.13 | 38.27 ± 0.26 | 37.32 ± 0.82 | 33.68 ± 0.44 | 36.54 ± 0.08 | 35.16 ± 4.13 | 36.51 ± 0.00 |
| | 20% random | 38.31 ± 0.29 | 37.23 ± 0.48 | 37.02 ± 0.57 | 38.58 ± 0.75 | 38.26 ± 0.66 | 38.08 ± 0.51 | 35.08 ± 0.70 | 31.92 ± 0.89 | 36.65 ± 0.24 | 35.51 ± 4.28 | 36.76 ± 0.00 |
| | 30% random | 37.69 ± 0.86 | 35.31 ± 2.67 | 36.50 ± 1.53 | 37.64 ± 1.05 | 37.03 ± 0.71 | 36.76 ± 1.48 | 34.45 ± 0.68 | 31.01 ± 1.26 | 35.75 ± 3.14 | 34.20 ± 5.02 | 34.57 ± 4.55 |
| | 40% random | 36.18 ± 2.43 | 34.94 ± 2.87 | 35.18 ± 2.96 | 35.81 ± 3.15 | 35.79 ± 3.39 | 35.94 ± 3.33 | 31.77 ± 1.91 | 28.50 ± 1.64 | 35.02 ± 3.38 | 33.02 ± 5.29 | 34.76 ± 3.52 |
| | 50% random | 32.61 ± 4.52 | 31.83 ± 5.21 | 31.40 ± 5.35 | 33.07 ± 4.87 | 32.40 ± 4.31 | 32.22 ± 4.63 | 29.46 ± 2.94 | 26.41 ± 1.74 | 31.55 ± 5.38 | 29.34 ± 5.86 | 31.39 ± 5.75 |

Table A5: Test accuracy of GLN methods (10 Runs). N/A indicates time or memory exceeded.

| Dataset | Noise type | GCN | NRGNN | RTGNN | CP | CLNode | PIGNN | DGNN | RNCGLN | UnionNET | CGNN | CR-GNN |
|---|---|---|---|---|---|---|---|---|---|---|---|---|
| A-Photos | Clean data | 91.82 ± 0.69 | 82.53 ± 1.37 | 82.31 ± 1.31 | 91.28 ± 0.72 | 90.84 ± 0.69 | 88.84 ± 0.55 | 76.60 ± 9.02 | 80.11 ± 4.75 | 33.16 ± 4.03 | 57.29 ± 27.33 | 62.09 ± 39.92 |
| | 10% pair | 89.83 ± 1.42 | 80.91 ± 1.68 | 82.15 ± 1.53 | 88.47 ± 2.03 | 89.18 ± 1.53 | 86.87 ± 2.12 | 70.58 ± 5.88 | 78.70 ± 4.12 | 33.34 ± 4.41 | 58.28 ± 24.31 | 60.35 ± 38.82 |
| | 20% pair | 85.74 ± 2.86 | 78.04 ± 2.29 | 82.31 ± 2.61 | 85.39 ± 2.47 | 84.76 ± 2.56 | 82.86 ± 3.73 | 61.95 ± 3.77 | 71.39 ± 6.39 | 33.16 ± 4.03 | 50.90 ± 30.22 | 43.59 ± 35.13 |
| | 30% pair | 79.26 ± 4.79 | 71.92 ± 6.84 | 70.33 ± 8.11 | 77.45 ± 5.26 | 79.12 ± 4.90 | 79.43 ± 4.63 | 51.09 ± 13.97 | 64.79 ± 9.92 | 30.89 ± 7.39 | 45.70 ± 25.46 | 37.53 ± 35.25 |
| | 40% pair | 64.90 ± 5.56 | 56.86 ± 10.90 | 61.67 ± 8.26 | 62.77 ± 6.27 | 63.67 ± 5.85 | 64.04 ± 5.48 | 47.22 ± 5.86 | 50.06 ± 9.59 | 22.80 ± 8.77 | 37.52 ± 16.62 | 40.27 ± 25.85 |
| | 50% pair | 44.96 ± 15.14 | 37.20 ± 13.45 | 41.39 ± 16.90 | 41.61 ± 14.76 | 42.52 ± 12.84 | 43.37 ± 11.65 | 32.95 ± 7.72 | 34.80 ± 9.47 | 21.10 ± 9.65 | 25.19 ± 16.12 | 20.85 ± 21.01 |
| | 10% uniform | 89.42 ± 1.61 | 81.42 ± 3.06 | 82.32 ± 1.68 | 89.53 ± 1.28 | 89.56 ± 1.88 | 87.85 ± 1.34 | 66.22 ± 17.13 | 75.31 ± 3.78 | 33.66 ± 3.87 | 55.16 ± 28.60 | 28.67 ± 39.10 |
| | 20% uniform | 88.02 ± 1.99 | 81.18 ± 3.14 | 83.00 ± 2.41 | 87.20 ± 2.35 | 87.76 ± 1.48 | 87.70 ± 2.42 | 62.00 ± 5.65 | 77.28 ± 5.43 | 33.61 ± 3.59 | 55.92 ± 26.11 | 33.93 ± 38.56 |
| | 30% uniform | 84.86 ± 3.27 | 78.30 ± 3.31 | 82.08 ± 6.02 | 83.52 ± 3.00 | 85.46 ± 2.70 | 84.90 ± 4.59 | 54.28 ± 8.37 | 72.80 ± 6.60 | 32.29 ± 4.47 | 47.26 ± 25.51 | 35.24 ± 32.66 |
| | 40% uniform | 79.33 ± 5.96 | 69.73 ± 10.22 | 77.42 ± 5.38 | 78.80 ± 5.91 | 81.17 ± 6.59 | 79.26 ± 7.16 | 49.35 ± 4.40 | 64.81 ± 4.72 | 27.91 ± 8.75 | 43.28 ± 23.00 | 34.55 ± 28.13 |
| | 50% uniform | 74.39 ± 5.08 | 64.94 ± 7.34 | 74.03 ± 10.33 | 72.11 ± 7.05 | 74.58 ± 4.91 | 76.51 ± 3.84 | 42.48 ± 5.84 | 59.78 ± 8.76 | 25.74 ± 9.98 | 38.23 ± 21.83 | 22.50 ± 25.64 |
| | 10% random | 88.05 ± 1.46 | 76.41 ± 3.22 | 83.83 ± 1.10 | 87.50 ± 1.68 | 87.42 ± 1.56 | 87.92 ± 0.83 | 63.42 ± 13.10 | 82.06 ± 2.76 | 28.18 ± 6.24 | 50.11 ± 29.55 | 43.39 ± 41.37 |
| | 20% random | 86.79 ± 1.58 | 77.31 ± 2.56 | 85.09 ± 2.25 | 85.05 ± 2.29 | 86.49 ± 1.61 | 86.79 ± 1.47 | 55.21 ± 15.74 | 76.70 ± 4.16 | 29.66 ± 6.36 | 53.08 ± 29.02 | 32.42 ± 37.39 |
| | 30% random | 82.24 ± 4.39 | 72.38 ± 8.61 | 83.68 ± 1.94 | 82.02 ± 3.14 | 82.57 ± 3.90 | 83.75 ± 2.97 | 56.66 ± 5.02 | 74.37 ± 8.86 | 27.53 ± 6.83 | 49.43 ± 27.36 | 31.77 ± 35.66 |
| | 40% random | 76.94 ± 5.44 | 70.85 ± 9.39 | 78.66 ± 8.15 | 76.52 ± 5.42 | 78.61 ± 4.14 | 81.20 ± 2.93 | 48.07 ± 9.46 | 71.55 ± 7.76 | 28.50 ± 8.17 | 48.97 ± 28.32 | 25.67 ± 28.29 |
| | 50% random | 69.85 ± 7.07 | 62.36 ± 8.28 | 72.54 ± 8.95 | 70.68 ± 4.97 | 71.03 ± 7.34 | 74.68 ± 5.07 | 42.65 ± 8.63 | 61.20 ± 5.19 | 27.25 ± 6.40 | 38.75 ± 25.24 | 24.68 ± 22.94 |
| DBLP | Clean data | 77.03 ± 0.35 | 81.35 ± 0.45 | 68.13 ± 2.16 | 78.19 ± 1.26 | 78.85 ± 0.33 | 78.37 ± 2.02 | 76.14 ± 1.03 | 73.00 ± 1.52 | 77.13 ± 0.24 | 67.23 ± 13.37 | 77.20 ± 1.00 |
| | 10% pair | 74.04 ± 1.58 | 79.58 ± 1.14 | 66.08 ± 1.99 | 75.85 ± 1.92 | 75.35 ± 1.65 | 76.90 ± 2.37 | 72.53 ± 2.17 | 70.17 ± 1.23 | 74.38 ± 1.53 | 63.26 ± 11.22 | 74.73 ± 2.17 |
| | 20% pair | 70.11 ± 1.40 | 74.90 ± 2.76 | 63.72 ± 3.82 | 71.35 ± 3.02 | 69.78 ± 1.87 | 74.85 ± 2.03 | 68.68 ± 2.75 | 67.46 ± 2.57 | 70.47 ± 1.56 | 63.05 ± 5.47 | 70.92 ± 2.01 |
| | 30% pair | 62.56 ± 2.39 | 69.82 ± 3.20 | 59.26 ± 6.03 | 65.80 ± 2.68 | 62.59 ± 1.50 | 69.31 ± 2.55 | 61.99 ± 2.79 | 64.35 ± 2.40 | 62.92 ± 2.86 | 58.06 ± 5.73 | 63.79 ± 3.39 |
| | 40% pair | 52.16 ± 7.86 | 61.01 ± 8.12 | 50.27 ± 7.46 | 54.76 ± 6.94 | 51.96 ± 7.01 | 59.45 ± 8.84 | 57.41 ± 4.63 | 51.76 ± 5.31 | 52.82 ± 7.67 | 49.49 ± 6.41 | 55.89 ± 10.76 |
| | 50% pair | 39.99 ± 7.94 | 47.50 ± 5.29 | 36.53 ± 8.64 | 40.44 ± 7.24 | 43.67 ± 7.20 | 46.66 ± 11.83 | 45.39 ± 8.74 | 42.75 ± 6.77 | 40.71 ± 8.01 | 41.54 ± 6.57 | 44.86 ± 9.65 |
| | 10% uniform | 75.24 ± 1.04 | 79.38 ± 1.68 | 66.81 ± 3.05 | 76.67 ± 1.62 | 75.40 ± 1.65 | 76.29 ± 2.31 | 72.79 ± 2.26 | 70.65 ± 2.65 | 75.57 ± 1.07 | 67.01 ± 6.27 | 75.88 ± 1.54 |
| | 20% uniform | 72.37 ± 3.11 | 77.38 ± 2.75 | 67.14 ± 2.95 | 72.98 ± 3.93 | 70.09 ± 3.66 | 74.21 ± 4.32 | 66.36 ± 3.45 | 69.06 ± 3.50 | 72.77 ± 3.11 | 60.02 ± 11.62 | 72.96 ± 2.57 |
| | 30% uniform | 69.66 ± 2.72 | 76.21 ± 1.79 | 66.89 ± 6.43 | 70.49 ± 3.71 | 65.31 ± 2.41 | 71.98 ± 2.19 | 63.47 ± 3.00 | 63.65 ± 3.44 | 70.33 ± 2.82 | 63.26 ± 5.73 | 70.23 ± 2.03 |
| | 40% uniform | 64.53 ± 5.58 | 70.36 ± 5.56 | 65.35 ± 5.54 | 65.74 ± 4.48 | 59.78 ± 4.32 | 67.83 ± 3.06 | 56.94 ± 6.30 | 61.03 ± 4.17 | 65.17 ± 5.64 | 56.51 ± 6.56 | 63.93 ± 5.99 |
| | 50% uniform | 57.05 ± 7.88 | 65.23 ± 6.91 | 60.56 ± 6.79 | 58.89 ± 6.91 | 52.34 ± 6.30 | 58.92 ± 7.79 | 53.47 ± 6.33 | 54.17 ± 5.30 | 57.92 ± 7.84 | 51.86 ± 7.91 | 57.42 ± 6.78 |
| | 10% random | 75.40 ± 0.88 | 80.01 ± 0.58 | 66.52 ± 2.01 | 75.33 ± 1.53 | 73.87 ± 1.39 | 77.08 ± 1.94 | 71.22 ± 2.14 | 72.38 ± 2.12 | 75.32 ± 1.01 | 65.57 ± 7.25 | 70.04 ± 8.89 |
| | 20% random | 72.50 ± 2.27 | 77.87 ± 2.03 | 64.52 ± 2.73 | 72.80 ± 1.99 | 70.71 ± 1.67 | 75.45 ± 2.44 | 68.85 ± 1.79 | 70.45 ± 1.66 | 72.72 ± 2.45 | 63.90 ± 7.54 | 71.74 ± 2.00 |
| | 30% random | 66.60 ± 3.99 | 73.83 ± 3.74 | 65.59 ± 3.50 | 68.30 ± 3.63 | 62.45 ± 3.15 | 70.76 ± 3.68 | 61.45 ± 1.60 | 65.54 ± 3.69 | 66.97 ± 4.03 | 58.70 ± 9.03 | 66.94 ± 3.11 |
| | 40% random | 62.76 ± 4.23 | 69.41 ± 3.04 | 62.10 ± 5.19 | 64.28 ± 2.67 | 56.62 ± 3.03 | 67.23 ± 4.10 | 58.28 ± 3.95 | 61.25 ± 5.42 | 63.30 ± 4.24 | 54.87 ± 9.01 | 64.35 ± 2.42 |
| | 50% random | 54.26 ± 6.94 | 65.10 ± 4.86 | 57.80 ± 8.88 | 56.75 ± 6.97 | 51.48 ± 7.06 | 61.17 ± 6.78 | 50.59 ± 5.01 | 54.78 ± 7.42 | 54.77 ± 7.22 | 51.81 ± 8.24 | 54.65 ± 6.99 |
| Blogcatalog | Clean data | 76.52 ± 0.58 | 78.46 ± 0.85 | 75.49 ± 0.37 | 75.70 ± 0.74 | 75.72 ± 0.92 | 63.13 ± 9.60 | 59.45 ± 12.70 | 56.59 ± 0.67 | 75.30 ± 0.94 | 23.26 ± 6.65 | 71.57 ± 1.07 |
| | 10% pair | 72.81 ± 1.53 | 76.73 ± 1.42 | 76.03 ± 0.71 | 71.82 ± 1.51 | 72.08 ± 1.09 | 61.10 ± 9.84 | 54.20 ± 11.46 | 54.71 ± 1.35 | 71.29 ± 1.03 | 26.77 ± 5.01 | 67.36 ± 2.20 |
| | 20% pair | 67.09 ± 2.88 | 73.76 ± 1.77 | 75.12 ± 2.37 | 65.53 ± 2.62 | 66.82 ± 2.26 | 55.00 ± 9.66 | 44.15 ± 11.46 | 62.53 ± 14.47 | 65.74 ± 3.29 | 26.11 ± 8.32 | 61.84 ± 3.59 |
| | 30% pair | 60.69 ± 1.76 | 69.55 ± 2.13 | 69.94 ± 3.25 | 59.52 ± 2.96 | 59.41 ± 2.50 | 53.04 ± 6.92 | 41.08 ± 12.47 | 47.48 ± 2.61 | 57.49 ± 3.23 | 25.86 ± 4.42 | 52.43 ± 5.01 |
| | 40% pair | 46.75 ± 4.37 | 53.59 ± 4.95 | 52.21 ± 7.91 | 46.48 ± 3.92 | 47.39 ± 6.44 | 40.62 ± 8.24 | 40.40 ± 6.53 | 39.02 ± 0.96 | 47.21 ± 3.24 | 20.54 ± 4.34 | 44.06 ± 3.28 |
| | 50% pair | 35.36 ± 5.71 | 36.95 ± 8.57 | 35.47 ± 6.85 | 35.92 ± 5.90 | 34.57 ± 7.27 | 33.90 ± 5.81 | 27.53 ± 6.93 | 31.90 ± 9.03 | 35.59 ± 4.51 | 20.53 ± 3.13 | 33.41 ± 5.06 |
| | 10% uniform | 74.40 ± 1.03 | 77.33 ± 2.01 | 76.41 ± 0.77 | 73.17 ± 1.40 | 74.02 ± 1.09 | 58.20 ± 8.23 | 53.92 ± 13.38 | 58.23 ± 9.32 | 73.38 ± 0.71 | 23.90 ± 6.39 | 70.37 ± 1.17 |
| | 20% uniform | 71.30 ± 1.23 | 76.21 ± 1.94 | 77.68 ± 0.93 | 70.54 ± 1.25 | 70.28 ± 1.32 | 59.68 ± 7.42 | 41.84 ± 10.65 | 52.78 ± 0.93 | 69.36 ± 1.37 | 31.22 ± 9.44 | 66.30 ± 3.31 |
| | 30% uniform | 69.36 ± 2.99 | 75.37 ± 1.88 | 76.34 ± 1.13 | 67.79 ± 2.01 | 68.18 ± 2.72 | 56.30 ± 9.27 | 37.26 ± 16.11 | 58.43 ± 15.32 | 67.36 ± 3.18 | 26.45 ± 7.98 | 63.32 ± 2.64 |
| | 40% uniform | 64.73 ± 2.36 | 73.18 ± 3.45 | 73.20 ± 2.17 | 63.04 ± 1.57 | 63.00 ± 3.75 | 49.31 ± 8.45 | 30.95 ± 12.83 | 68.15 ± 15.31 | 62.37 ± 2.65 | 18.72 ± 2.72 | 61.27 ± 4.73 |
| | 50% uniform | 60.08 ± 3.58 | 71.72 ± 3.53 | 69.56 ± 2.34 | 60.18 ± 2.63 | 57.85 ± 2.73 | 49.30 ± 9.73 | 32.47 ± 11.46 | 63.63 ± 13.45 | 59.56 ± 3.27 | 21.91 ± 5.18 | 52.90 ± 4.96 |
| | 10% random | 72.63 ± 1.36 | 75.98 ± 1.96 | 74.73 ± 1.08 | 71.72 ± 1.12 | 72.11 ± 1.76 | 59.90 ± 11.53 | 55.78 ± 12.89 | 52.63 ± 1.02 | 24.37 ± 3.36 | 33.94 ± 7.71 | 69.11 ± 1.81 |
| | 20% random | 70.70 ± 1.28 | 74.47 ± 2.51 | 76.07 ± 1.29 | 69.45 ± 2.13 | 70.47 ± 1.52 | 58.83 ± 7.93 | 46.83 ± 14.20 | 58.87 ± 14.45 | 22.75 ± 4.21 | 27.60 ± 8.19 | 67.86 ± 2.05 |
| | 30% random | 65.81 ± 2.07 | 72.24 ± 1.82 | 72.67 ± 3.25 | 64.86 ± 3.17 | 64.88 ± 2.94 | 53.74 ± 5.12 | 42.24 ± 13.18 | 54.80 ± 13.27 | 21.90 ± 2.88 | 21.63 ± 7.64 | 61.84 ± 4.50 |
| | 40% random | 61.75 ± 4.23 | 69.90 ± 4.74 | 69.63 ± 4.19 | 60.25 ± 4.69 | 61.20 ± 4.64 | 49.55 ± 11.35 | 32.52 ± 9.61 | 41.72 ± 5.49 | 20.01 ± 3.39 | 20.21 ± 5.05 | 55.99 ± 5.15 |
| | 50% random | 57.61 ± 3.48 | 66.08 ± 3.92 | 65.12 ± 3.75 | 56.03 ± 3.56 | 55.67 ± 3.99 | 44.43 ± 12.37 | 40.12 ± 6.37 | 61.65 ± 13.14 | 20.12 ± 3.92 | 17.78 ± 1.70 | 52.04 ± 3.54 |
| Flickr | 0% clean | 56.75 ± 0.44 | 46.04 ± 1.10 | 41.93 ± 3.77 | 52.92 ± 0.87 | 54.85 ± 1.08 | 55.41 ± 1.04 | 17.68 ± 5.87 | 31.81 ± 0.37 | 30.88 ± 0.22 | 13.53 ± 2.40 | 48.97 ± 1.37 |
| | 10% pair | 54.43 ± 1.22 | 46.77 ± 4.45 | 37.44 ± 4.04 | 49.53 ± 0.78 | 52.60 ± 1.59 | 54.79 ± 1.22 | 19.10 ± 5.04 | 27.12 ± 0.73 | 21.78 ± 2.87 | 11.34 ± 1.55 | 25.50 ± 18.51 |
| | 20% pair | 51.55 ± 1.59 | 45.01 ± 4.49 | 37.35 ± 3.93 | 46.50 ± 1.30 | 49.98 ± 2.10 | 51.20 ± 1.75 | 19.33 ± 3.00 | 25.51 ± 0.96 | 23.06 ± 3.76 | 12.47 ± 1.88 | 31.07 ± 17.16 |
| | 30% pair | 45.68 ± 3.02 | 38.79 ± 4.60 | 36.50 ± 5.36 | 40.74 ± 2.88 | 45.06 ± 2.77 | 44.61 ± 3.11 | 15.64 ± 4.02 | 22.95 ± 0.82 | 20.75 ± 3.80 | 11.33 ± 1.50 | 29.38 ± 12.67 |
| | 40% pair | 38.82 ± 1.82 | 31.31 ± 3.55 | 34.79 ± 3.85 | 35.05 ± 2.35 | 37.89 ± 2.52 | 39.35 ± 2.95 | 19.33 ± 4.71 | 20.94 ± 0.97 | 16.60 ± 3.27 | 10.85 ± 1.28 | 26.57 ± 10.67 |
| | 50% pair | 28.91 ± 2.87 | 28.41 ± 4.70 | 29.81 ± 3.90 | 28.42 ± 1.43 | 28.96 ± 2.93 | 28.71 ± 2.38 | 13.62 ± 3.24 | 18.36 ± 0.89 | 16.97 ± 2.96 | 11.42 ± 1.63 | 25.38 ± 5.58 |
| | 10% uniform | 55.44 ± 0.82 | 48.35 ± 4.38 | 39.50 ± 3.16 | 49.66 ± 1.03 | 53.54 ± 1.04 | 55.20 ± 1.43 | 16.85 ± 6.45 | 27.45 ± 1.03 | 21.84 ± 2.01 | 12.59 ± 2.70 | 40.03 ± 15.28 |
| | 20% uniform | 53.06 ± 1.88 | 45.50 ± 5.08 | 42.85 ± 2.06 | 46.72 ± 1.29 | 51.38 ± 1.43 | 52.41 ± 2.48 | 17.81 ± 4.04 | 25.30 ± 0.75 | 21.22 ± 3.69 | 11.76 ± 2.24 | 30.66 ± 16.81 |
| | 30% uniform | 49.77 ± 0.96 | 42.76 ± 5.18 | 43.72 ± 3.28 | 43.13 ± 1.41 | 48.06 ± 1.50 | 49.38 ± 1.40 | 19.24 ± 5.02 | 23.99 ± 1.51 | 21.19 ± 3.89 | 11.71 ± 1.72 | 31.09 ± 13.98 |
| | 40% uniform | 47.29 ± 2.44 | 40.46 ± 5.09 | 40.63 ± 4.48 | 40.74 ± 2.19 | 45.56 ± 2.43 | 47.35 ± 1.48 | 17.53 ± 4.37 | 22.22 ± 1.18 | 19.85 ± 3.98 | 12.00 ± 2.14 | 29.95 ± 13.03 |
| | 50% uniform | 42.95 ± 1.99 | 36.47 ± 5.39 | 40.70 ± 3.35 | 37.58 ± 2.04 | 40.80 ± 2.14 | 43.15 ± 1.82 | 15.09 ± 3.36 | 19.51 ± 0.75 | 17.57 ± 3.30 | 10.85 ± 1.37 | 29.24 ± 9.87 |
| | 10% random | 55.17 ± 1.09 | 46.89 ± 2.42 | 39.02 ± 5.28 | 49.93 ± 1.25 | 53.92 ± 0.90 | 54.08 ± 1.43 | 19.25 ± 4.82 | 27.13 ± 0.91 | 23.27 ± 3.11 | 11.48 ± 1.66 | 39.96 ± 15.25 |
| | 20% random | 53.72 ± 0.98 | 47.13 ± 4.11 | 41.10 ± 2.94 | 46.46 ± 2.01 | 51.19 ± 1.45 | 52.14 ± 1.03 | 17.34 ± 4.98 | 27.60 ± 1.03 | 27.58 ± 3.78 | 11.97 ± 1.48 | 38.67 ± 10.05 |
| | 30% random | 49.57 ± 1.31 | 44.52 ± 6.94 | 40.85 ± 5.70 | 42.54 ± 1.52 | 47.45 ± 1.37 | 49.06 ± 2.28 | 15.80 ± 3.41 | 24.38 ± 1.18 | 22.41 ± 4.58 | 11.73 ± 1.15 | 30.19 ± 13.35 |
| | 40% random | 47.16 ± 1.51 | 36.38 ± 5.53 | 41.73 ± 3.50 | 39.96 ± 2.65 | 46.19 ± 1.91 | 46.67 ± 2.31 | 17.19 ± 4.82 | 21.74 ± 2.01 | 17.30 ± 4.10 | 11.26 ± 1.02 | 34.01 ± 12.29 |
| | 50% random | 40.66 ± 2.78 | 31.50 ± 7.18 | 37.42 ± 4.48 | 34.18 ± 3.25 | 38.76 ± 3.69 | 40.81 ± 2.90 | 13.88 ± 4.46 | 19.26 ± 1.73 | 14.87 ± 3.03 | 11.95 ± 1.25 | 23.60 ± 10.92 |
| Roman Empire | 0% clean | 36.97 ± 0.28 | 48.89 ± 0.44 | 48.94 ± 0.32 | 36.71 ± 0.41 | 35.74 ± 0.24 | 33.23 ± 0.48 | 24.04 ± 0.48 | 53.16 ± 4.25 | 16.00 ± 1.18 | 20.88 ± 0.81 | 28.78 ± 0.68 |
| | 10% pair | 34.89 ± 0.65 | 49.80 ± 0.30 | 49.45 ± 0.62 | 33.71 ± 0.78 | 34.12 ± 0.76 | 33.52 ± 0.71 | 22.67 ± 0.53 | 52.58 ± 5.47 | 15.50 ± 1.26 | 20.69 ± 0.86 | 27.04 ± 1.15 |
| | 20% pair | 32.91 ± 0.59 | 48.66 ± 0.80 | 47.25 ± 0.98 | 31.49 ± 1.06 | 31.83 ± 0.97 | 32.15 ± 0.93 | 22.00 ± 0.64 | 48.34 ± 6.98 | 15.44 ± 1.25 | 19.61 ± 0.90 | 26.19 ± 1.59 |
| | 30% pair | 30.31 ± 0.99 | 45.26 ± 1.32 | 44.39 ± 1.10 | 29.65 ± 0.74 | 28.23 ± 4.86 | 28.38 ± 1.32 | 21.37 ± 0.49 | 41.78 ± 6.19 | 14.53 ± 1.16 | 18.07 ± 1.05 | 24.01 ± 1.72 |
| | 40% pair | 26.36 ± 1.23 | 39.31 ± 2.08 | 37.22 ± 2.59 | 25.23 ± 1.12 | 25.78 ± 1.30 | 25.77 ± 1.04 | 19.24 ± 0.60 | 39.56 ± 8.16 | 14.26 ± 0.43 | 17.48 ± 1.11 | 22.39 ± 1.23 |
| | 50% pair | 21.70 ± 1.32 | 29.53 ± 1.97 | 29.24 ± 1.88 | 21.62 ± 1.04 | 21.73 ± 1.28 | 21.57 ± 1.87 | 17.00 ± 0.75 | 28.30 ± 4.38 | 14.38 ± 0.92 | 16.45 ± 1.27 | 20.00 ± 1.90 |
| | 10% uniform | 34.97 ± 0.37 | 49.06 ± 0.68 | 49.78 ± 0.48 | 33.91 ± 0.38 | 34.32 ± 0.59 | 33.49 ± 0.62 | 22.80 ± 0.51 | 52.74 ± 3.82 | 16.45 ± 1.30 | 20.33 ± 0.93 | 26.29 ± 1.59 |
| | 20% uniform | 33.73 ± 0.37 | 47.99 ± 0.76 | 49.15 ± 0.82 | 32.34 ± 0.73 | 33.05 ± 0.94 | 32.87 ± 0.80 | 22.07 ± 0.53 | 53.16 ± 2.91 | 15.55 ± 1.39 | 19.37 ± 0.26 | 25.22 ± 1.17 |
| | 30% uniform | 32.58 ± 0.80 | 47.07 ± 0.47 | 48.14 ± 1.07 | 31.36 ± 0.71 | 32.34 ± 0.82 | 31.37 ± 0.86 | 21.39 ± 0.67 | 51.79 ± 2.52 | 15.78 ± 1.26 | 18.96 ± 0.81 | 23.70 ± 1.33 |
| | 40% uniform | 30.33 ± 0.90 | 45.63 ± 0.67 | 46.74 ± 1.14 | 28.84 ± 0.87 | 30.24 ± 0.96 | 29.62 ± 1.19 | 19.88 ± 1.04 | 49.72 ± 2.67 | 15.02 ± 0.95 | 17.19 ± 1.85 | 17.76 ± 2.65 |
| | 50% uniform | 27.56 ± 1.22 | 43.91 ± 0.92 | 45.05 ± 1.05 | 26.72 ± 1.19 | 27.73 ± 1.15 | 27.26 ± 1.13 | 18.29 ± 0.92 | 45.58 ± 2.34 | 15.24 ± 1.31 | 16.29 ± 1.68 | 16.43 ± 1.70 |
| | 10% random | 35.59 ± 0.42 | 48.25 ± 0.58 | 49.24 ± 0.50 | 35.27 ± 0.55 | 34.54 ± 0.48 | 33.31 ± 0.62 | 23.59 ± 0.58 | 52.91 ± 3.51 | 15.10 ± 1.06 | 21.16 ± 1.19 | 27.76 ± 1.20 |
| | 20% random | 33.94 ± 0.55 | 47.75 ± 0.56 | 48.62 ± 0.58 | 32.92 ± 0.63 | 33.45 ± 1.20 | 32.17 ± 0.82 | 23.29 ± 0.38 | 50.75 ± 3.67 | 16.21 ± 1.10 | 22.55 ± 1.50 | 25.84 ± 1.85 |
| | 30% random | 31.09 ± 0.77 | 45.63 ± 0.67 | 46.48 ± 0.65 | 30.57 ± 0.93 | 29.37 ± 4.38 | 30.92 ± 0.87 | 20.95 ± 0.82 | 50.49 ± 2.13 | 14.53 ± 1.09 | 17.26 ± 1.61 | 23.81 ± 1.51 |
| | 40% random | 30.63 ± 0.58 | 45.66 ± 0.79 | 46.26 ± 0.92 | 29.71 ± 0.41 | 30.34 ± 0.85 | 29.68 ± 1.07 | 21.49 ± 0.59 | 48.73 ± 3.86 | 14.50 ± 0.85 | 18.43 ± 2.51 | 21.21 ± 3.03 |
| | 50% random | 28.22 ± 0.99 | 43.43 ± 1.38 | 44.57 ± 1.17 | 27.49 ± 0.81 | 27.90 ± 0.89 | 28.10 ± 1.33 | 19.81 ± 1.05 | 47.47 ± 1.21 | 14.84 ± 0.74 | 16.55 ± 2.07 | 17.52 ± 1.28 |

Table A6: Additional Results For GLN under 30% Uniform noise (10 Runs), N/A indicates time or memory exceeded. ACLT denotes Accuracy of Correct Labeled Training nodes, AILT denotes Accuracy of Incorrect Labeled Training nodes, AUCS denotes Accuracy of Unlabeled Correct Supervised nodes, AUU denotes Accuracy of Unlabeled Unsupervised nodes, AUIS denotes Accuracy of Unlabeled Incorrect Supervised nodes

| Dataset | Records | GCN | NRGNN | RTGNN | CP | CLNode | PIGNN | DGNN | RNCGLN | UnionNET | CGNN | CR-GNN |
|---|---|---|---|---|---|---|---|---|---|---|---|---|
| Cora | ACLT | 98.53±0.93 | 84.53±4.07 | 82.29±3.33 | 98.77±1.49 | 94.14±3.44 | 98.43±1.55 | 81.06±10.80 | 99.00±1.18 | 98.60±1.18 | 97.16±2.79 | 96.39±1.92 |
| | AILT | 33.77±9.40 | 75.64±6.68 | 71.57±8.33 | 32.04±14.17 | 34.88±9.96 | 29.19±20.67 | 37.58±15.48 | 13.44±8.24 | 33.89±15.53 | 31.37±10.72 | 34.93±8.11 |
| | AILMT | 62.14±10.37 | 9.84±7.64 | 11.04±4.65 | 65.25±15.24 | 56.09±10.07 | 68.10±22.30 | 41.82±19.00 | 84.10±9.49 | 62.01±17.07 | 63.61±12.09 | 58.93±8.69 |
| | AUCS | 80.76±2.95 | 83.11±3.16 | 75.53±4.80 | 80.85±3.46 | 76.77±3.30 | 80.30±4.69 | 70.89±7.28 | 77.06±3.30 | 82.45±2.69 | 80.41±3.62 | 80.09±2.90 |
| | AUU | 71.99±3.44 | 81.33±2.25 | 73.44±4.95 | 72.32±4.45 | 67.11±5.11 | 70.28±8.69 | 63.60±6.94 | 75.36±3.33 | 73.46±3.70 | 70.38±4.04 | 71.75±4.73 |
| | AUIS | 51.55±6.53 | 78.81±5.94 | 69.50±8.06 | 51.46±12.99 | 43.86±7.48 | 49.25±16.24 | 45.72±10.64 | 72.92±4.66 | 52.59±9.00 | 46.81±10.04 | 51.88±6.64 |
| | Time | 0.26±0.47 | 9.39±2.21 | 12.03±1.33 | 16.26±0.85 | 5.46±0.18 | 3.16±4.52 | 3.18±0.71 | 17.34±3.47 | 15.90±13.85 | 1.08±0.25 | 0.28±0.04 |
| Citeseer | ACLT | 99.18±0.96 | 75.93±3.79 | 79.36±5.55 | 98.27±1.83 | 92.63±5.08 | 99.54±0.81 | 96.49±4.40 | 99.88±0.37 | 96.96±1.82 | 97.51±1.98 | 97.63±2.10 |
| | AILT | 1.39±2.65 | 43.78±8.23 | 35.36±7.72 | 1.73±3.01 | 8.88±6.91 | 1.17±2.84 | 6.83±6.77 | 3.39±4.21 | 5.37±3.32 | 3.09±2.21 | 2.31±1.77 |
| | AILMT | 97.77±4.20 | 30.20±7.94 | 41.27±11.34 | 97.43±4.78 | 81.70±13.29 | 98.54±3.73 | 91.28±8.18 | 96.61±4.21 | 90.69±5.27 | 94.65±2.53 | 96.06±3.42 |
| | AUCS | 74.70±3.49 | 72.69±3.35 | 68.45±4.52 | 74.16±2.86 | 74.23±3.31 | 75.02±3.53 | 69.51±3.47 | 72.77±4.05 | 74.60±4.58 | 75.26±3.72 | 74.88±3.88 |
| | AUU | 57.50±3.60 | 66.67±4.22 | 62.19±3.25 | 57.40±3.08 | 58.85±4.43 | 57.92±3.84 | 54.90±4.11 | 67.08±4.66 | 59.48±3.82 | 58.54±4.01 | 58.02±3.99 |
| | AUIS | 16.45±5.60 | 51.37±17.86 | 46.72±15.10 | 18.15±6.69 | 21.92±5.69 | 17.26±6.85 | 20.80±9.17 | 53.44±14.21 | 23.41±7.92 | 19.82±7.47 | 18.61±5.69 |
| | Time | 0.57±0.43 | 7.02±1.32 | 38.50±25.07 | 20.23±0.46 | 5.90±0.26 | 13.80±5.01 | 3.26±0.73 | 7.76±3.93 | 91.13±55.45 | 7.79±12.90 | 0.92±0.87 |
| pubmed | ACLT | 97.08±2.61 | 74.37±15.52 | 92.21±3.85 | 96.99±2.48 | 97.38±3.31 | 98.77±2.10 | 92.27±12.38 | N/A | 96.85±3.44 | 87.66±22.82 | 96.76±2.72 |
| | AILT | 30.01±15.38 | 61.79±14.74 | 57.26±9.83 | 33.22±17.78 | 19.79±19.87 | 18.18±17.78 | 20.04±16.47 | N/A | 17.85±15.35 | 16.77±15.34 | 25.29±14.37 |
| | AILMT | 66.93±16.47 | 25.11±12.47 | 35.58±8.01 | 64.81±18.64 | 78.16±19.66 | 80.85±17.99 | 76.56±19.22 | N/A | 79.02±20.17 | 77.93±24.69 | 71.15±15.53 |
| | AUCS | 69.49±7.41 | 59.98±13.05 | 68.50±11.06 | 71.10±8.60 | 72.16±5.99 | 73.42±9.21 | 69.09±7.99 | N/A | 70.95±6.55 | 62.57±14.01 | 72.93±7.06 |
| | AUU | 59.47±7.87 | 58.42±8.75 | 64.74±7.36 | 60.00±9.67 | 58.42±8.02 | 62.11±7.77 | 58.95±13.31 | N/A | 59.47±8.61 | 52.11±14.35 | 62.11±9.54 |
| | AUIS | 27.06±21.05 | 52.10±29.07 | 49.64±22.84 | 35.95±26.69 | 19.48±16.76 | 27.66±18.07 | 37.54±29.11 | N/A | 25.08±17.63 | 22.86±15.84 | 41.15±31.33 |
| | Time | 0.31±0.35 | 119.30±44.57 | 99.15±23.67 | 95.65±1.74 | 5.72±0.23 | 37.37±10.76 | 3.09±1.34 | N/A | 7.66±3.09 | 48.74±78.02 | 2.44±3.53 |
| A-Computers | ACLT | 90.81±3.10 | 61.82±5.95 | 73.58±7.57 | 88.51±4.38 | 92.01±2.61 | 93.43±2.98 | 37.44±10.38 | 84.67±10.92 | 13.17±4.37 | 44.71±32.57 | 53.11±30.63 |
| | AILT | 68.95±10.52 | 52.34±7.43 | 65.89±6.97 | 70.81±4.68 | 75.59±4.39 | 70.94±7.66 | 37.92±9.60 | 26.18±13.75 | 13.20±5.35 | 38.55±29.40 | 43.38±24.06 |
| | AILMT | 13.27±6.96 | 9.41±4.94 | 6.75±4.86 | 13.87±6.04 | 11.40±3.23 | 17.77±7.85 | 7.61±3.72 | 53.33±24.40 | 11.02±1.87 | 10.19±4.64 | 10.96±4.16 |
| | AUCS | 83.38±2.39 | 72.48±5.35 | 72.57±5.67 | 82.29±3.76 | 84.09±2.50 | 83.08±2.38 | 50.31±8.42 | 58.66±4.12 | 26.42±5.21 | 47.06±24.61 | 52.15±30.77 |
| | AUU | 81.58±3.70 | 71.15±5.00 | 71.37±5.33 | 80.65±4.00 | 82.77±2.99 | 81.69±3.08 | 49.28±7.37 | 57.63±3.75 | 26.26±5.46 | 46.04±24.70 | 51.12±30.06 |
| | AUIS | 78.02±7.47 | 69.03±6.67 | 67.42±6.79 | 77.52±4.44 | 80.19±5.43 | 79.78±5.93 | 46.11±7.94 | 54.96±6.97 | 21.99±7.48 | 42.20±26.69 | 48.21±29.15 |
| | Time | 1.28±0.41 | 120.63±28.31 | 304.07±56.41 | 82.97±2.11 | 7.82±0.27 | 117.10±27.10 | 6.49±1.42 | 120.78±64.12 | 130.79±115.06 | 53.97±68.57 | 5.71±3.92 |
| A-Photos | ACLT | 91.74±2.96 | 70.74±6.42 | 88.81±1.92 | 89.85±5.02 | 93.21±2.31 | 92.58±2.35 | 61.43±16.39 | 90.61±7.73 | 17.57±3.35 | 45.89±31.92 | 46.76±36.50 |
| | AILT | 80.78±5.59 | 64.16±8.28 | 87.14±4.78 | 78.46±6.17 | 84.82±5.38 | 81.95±7.74 | 49.46±10.26 | 44.18±15.75 | 17.33±6.44 | 38.54±24.55 | 40.67±29.28 |
| | AILMT | 11.01±5.49 | 8.46±3.45 | 3.64±2.57 | 11.86±6.11 | 7.90±4.54 | 11.69±7.82 | 16.23±7.15 | 43.90±20.70 | 9.88±4.15 | 12.58±6.89 | 11.40±3.17 |
| | AUCS | 92.21±2.44 | 85.62±2.96 | 89.98±2.10 | 90.74±2.98 | 93.08±1.97 | 92.04±4.62 | 61.05±15.59 | 75.71±6.59 | 21.70±7.80 | 47.20±30.80 | 44.70±38.10 |
| | AUU | 89.76±2.84 | 84.29±2.79 | 89.19±2.13 | 88.85±3.20 | 91.15±2.33 | 89.68±5.00 | 57.84±13.94 | 74.40±6.73 | 22.05±7.18 | 46.53±29.54 | 42.32±36.11 |
| | AUIS | 87.01±4.72 | 82.46±5.86 | 88.84±4.76 | 86.34±3.99 | 88.71±3.40 | 86.29±6.30 | 52.75±9.63 | 71.08±7.89 | 18.81±6.56 | 43.12±28.41 | 41.11±30.97 |
| | Time | 0.87±0.29 | 69.88±18.58 | 171.43±15.79 | 42.46±2.46 | 6.35±0.21 | 25.60±10.09 | 4.28±1.26 | 46.69±19.13 | 84.42±63.51 | 19.21±30.75 | 1.92±1.94 |
| DBLP | ACLT | 98.34±1.71 | 77.39±3.28 | 81.42±3.61 | 94.75±5.81 | 91.51±4.99 | 95.77±3.37 | 84.85±8.72 | 99.91±0.27 | 97.68±1.87 | 89.25±13.11 | 97.83±1.29 |
| | AILT | 21.13±8.02 | 65.65±7.34 | 64.47±11.16 | 25.14±12.36 | 18.56±7.63 | 32.41±9.67 | 29.66±20.08 | 0.35±0.74 | 23.42±9.24 | 25.90±17.84 | 20.36±6.45 |
| | AILMT | 75.33±9.66 | 19.50±6.18 | 21.84±10.51 | 69.94±15.57 | 71.48±9.17 | 61.76±11.53 | 57.28±25.14 | 99.65±0.74 | 73.04±11.00 | 65.68±24.20 | 74.00±6.77 |
| | AUCS | 83.15±1.43 | 82.14±1.29 | 77.00±3.16 | 82.77±1.69 | 81.08±1.49 | 83.16±2.32 | 77.32±3.08 | 71.17±4.42 | 83.31±1.58 | 79.18±4.77 | 83.03±1.01 |
| | AUU | 74.10±2.64 | 79.51±2.22 | 75.81±3.95 | 74.42±2.84 | 66.41±4.95 | 76.30±2.74 | 65.52±4.56 | 67.94±5.00 | 75.13±2.52 | 70.24±4.15 | 73.31±3.48 |
| | AUIS | 55.65±9.47 | 74.46±5.69 | 73.98±8.53 | 57.55±9.84 | 37.50±6.55 | 62.98±6.53 | 42.95±12.85 | 61.48±6.60 | 58.56±9.00 | 53.80±14.58 | 54.13±8.60 |
| | Time | 0.13±0.04 | 40.36±3.96 | 79.03±11.68 | 94.58±0.76 | 5.25±0.43 | 9.18±2.04 | 3.36±0.42 | 382.95±99.05 | 20.22±4.44 | 9.73±6.18 | 0.76±0.13 |
| Blogcatalog | ACLT | 87.77±4.37 | 86.65±4.06 | 87.76±1.94 | 89.31±4.93 | 89.04±3.32 | 74.12±12.13 | 32.02±15.23 | 100.00±0.00 | 86.94±5.68 | 23.93±7.31 | 79.57±4.99 |
| | AILT | 55.16±11.35 | 80.93±6.00 | 81.74±4.27 | 51.32±10.19 | 55.96±7.26 | 51.53±9.62 | 30.42±12.37 | 0.00±0.00 | 58.00±9.93 | 21.26±6.69 | 58.42±5.52 |
| | AILMT | 23.31±15.15 | 5.27±2.94 | 5.31±2.81 | 27.56±12.78 | 24.04±9.24 | 23.57±7.21 | 14.46±4.97 | 22.48±11.43 | 18.68±5.00 | 17.04±5.83 |  |
| | AUCS | 70.78±3.83 | 76.74±1.78 | 78.57±0.97 | 69.66±2.40 | 69.89±2.97 | 57.70±9.10 | 29.69±13.23 | 58.45±15.30 | 69.67±3.21 | 22.06±7.07 | 66.30±2.76 |
| | AUU | 70.07±3.86 | 76.16±1.85 | 77.83±0.95 | 68.88±2.21 | 69.11±2.92 | 57.10±9.16 | 29.35±12.93 | 58.45±15.33 | 68.75±3.24 | 21.53±6.59 | 65.70±2.77 |
| | AUIS | 71.71±4.40 | 76.95±2.52 | 79.49±1.49 | 69.84±2.53 | 70.41±3.21 | 57.68±8.90 | 29.74±13.14 | 57.51±15.99 | 70.37±3.76 | 21.65±7.70 | 66.71±2.53 |
| | Time | 0.99±0.47 | 60.40±12.65 | 115.91±27.74 | 32.03±0.81 | 7.42±0.80 | 20.30±4.93 | 5.71±2.75 | 7.75±5.92 | 230.91±79.49 | 6.11±7.66 | 3.40±1.19 |
| Flickr | ACLT | 95.45±1.89 | 68.78±7.09 | 66.96±5.37 | 92.14±8.13 | 91.20±4.01 | 91.54±3.13 | 16.61±3.97 | 100.00±0.00 | 30.89±5.99 | 13.19±4.12 | 51.54±35.03 |
| | AILT | 18.91±5.75 | 54.56±5.12 | 55.47±6.18 | 11.09±8.43 | 22.33±6.63 | 24.37±5.78 | 16.01±6.84 | 0.00±0.00 | 19.05±7.03 | 12.25±3.89 | 16.05±5.82 |
| | AILMT | 69.51±6.85 | 10.91±2.84 | 8.02±2.08 | 75.50±20.23 | 61.10±12.00 | 59.36±7.47 | 10.69±3.00 | 100.00±0.00 | 17.50±3.14 | 12.71±4.29 | 34.00±25.64 |
| | AUCS | 50.84±1.32 | 43.94±5.70 | 42.07±5.04 | 44.45±1.33 | 49.32±2.36 | 50.02±2.44 | 18.03±4.40 | 19.22±1.85 | 19.74±3.91 | 11.84±2.42 | 29.45±13.64 |
| | AUU | 50.72±1.10 | 43.39±5.72 | 42.28±5.04 | 44.20±1.30 | 49.12±2.23 | 50.09±2.11 | 17.84±3.91 | 20.46±1.64 | 20.11±3.87 | 11.71±2.31 | 28.91±13.83 |
| | AUIS | 50.17±0.86 | 42.86±5.57 | 40.50±5.28 | 43.13±1.90 | 48.25±2.47 | 49.39±2.66 | 18.13±3.64 | 18.19±1.81 | 18.48±3.69 | 11.26±2.66 | 26.79±12.93 |
| | Time | 1.74±0.50 | 70.22±20.06 | 78.78±41.05 | 41.99±0.79 | 9.00±0.85 | 38.00±4.57 | 7.67±1.19 | 2.27±0.64 | 187.40±202.26 | 6.89±6.41 | 1.98±1.34 |
| A-Ratings | ACLT | 46.76±4.04 | 41.32±2.96 | 44.54±1.60 | 48.01±6.13 | 42.83±1.63 | 40.48±1.01 | 64.51±14.29 | 87.64±1.34 | 37.40±0.50 | 36.19±4.32 | 37.46±0.43 |
| | AILT | 34.53±2.29 | 36.17±0.89 | 35.23±2.40 | 35.13±2.45 | 35.49±1.04 | 37.00±0.88 | 23.78±8.55 | 12.54±2.19 | 36.95±1.06 | 35.24±4.24 | 36.79±1.05 |
| | AILMT | 21.13±3.33 | 18.29±2.85 | 19.79±2.09 | 21.45±3.94 | 19.15±2.24 | 18.25±2.24 | 44.55±19.71 | 67.97±5.40 | 16.63±2.51 | 17.17±2.79 | 16.69±2.47 |
| | AUCS | 39.19±0.70 | 37.45±0.98 | 38.28±0.65 | 39.05±0.85 | 38.32±1.17 | 38.23±0.53 | 36.68±1.17 | 32.26±1.22 | 37.91±0.49 | 36.47±4.70 | 37.93±0.41 |
| | AUU | 38.47±0.59 | 37.03±0.64 | 37.69±0.47 | 38.46±0.65 | 37.78±0.84 | 37.75±0.35 | 34.51±1.35 | 31.74±0.97 | 37.44±0.18 | 35.99±4.65 | 37.47±0.00 |
| | AUIS | 38.02±1.00 | 37.10±0.62 | 37.24±1.00 | 38.07±0.98 | 37.72±1.27 | 37.89±0.98 | 30.19±3.46 | 31.11±1.28 | 37.92±1.12 | 36.41±5.05 | 37.89±1.22 |
| | Time | 0.38±0.70 | 119.00±79.45 | 369.63±87.64 | 80.26±0.36 | 4.37±0.27 | 130.84±93.01 | 2.16±0.59 | 16.09±12.09 | 285.42±140.46 | 4.48±4.53 | 0.36±0.25 |
| Roman Empire | ACLT | 54.23±5.13 | 55.75±2.31 | 60.58±1.12 | 51.12±4.83 | 55.34±3.09 | 39.46±1.19 | 44.71±9.48 | 67.67±6.59 | 15.82±2.06 | 22.20±1.75 | 27.42±2.94 |
| | AILT | 27.20±1.93 | 47.50±3.24 | 50.12±2.92 | 27.62±2.09 | 27.91±1.95 | 29.56±2.52 | 18.71±1.96 | 46.65±2.53 | 15.01±1.90 | 19.48±2.91 | 24.45±2.87 |
| | AILMT | 13.80±3.46 | 4.88±1.41 | 5.78±1.69 | 12.98±3.37 | 13.89±2.40 | 6.48±1.64 | 16.41±3.90 | 15.54±4.88 | 5.70±0.97 | 5.05±1.01 | 5.62±1.37 |
| | AUCS | 27.01±1.34 | 41.09±1.93 | 43.82±1.43 | 27.05±1.31 | 26.88±0.61 | 28.77±1.68 | 18.58±1.36 | 48.63±1.15 | 14.98±1.18 | 18.52±1.09 | 22.75±1.65 |
| | AUU | 28.49±1.27 | 41.63±1.68 | 44.37±1.16 | 28.19±0.98 | 28.34±0.78 | 29.59±1.27 | 19.91±1.04 | 48.70±0.86 | 14.81±0.94 | 18.87±1.00 | 22.99±1.65 |
| | AUIS | 31.38±1.58 | 42.21±2.28 | 44.81±2.11 | 30.59±1.38 | 31.18±1.63 | 31.15±1.37 | 22.65±1.06 | 48.33±0.99 | 14.30±1.29 | 19.25±1.09 | 23.29±2.05 |
| | Time | 0.53±0.65 | 151.39±19.51 | 340.92±68.92 | 73.48±0.59 | 6.54±0.30 | 233.25±27.18 | 1.77±0.08 | 159.07±69.94 | 921.33±832.73 | 188.38±93.65 | 9.82±0.83 |

# B  Additional experiment result figures

## B.1  Transition patterns

In this work, we primarily consider two types of label noise: pair noise and uniform noise, as defined in Section 2. Additionally, our code implementation supports random noise. Unlike the fixed transition patterns of the first two types, the label transition pattern for random noise is generated randomly. Below are examples of their label transition probabilities.

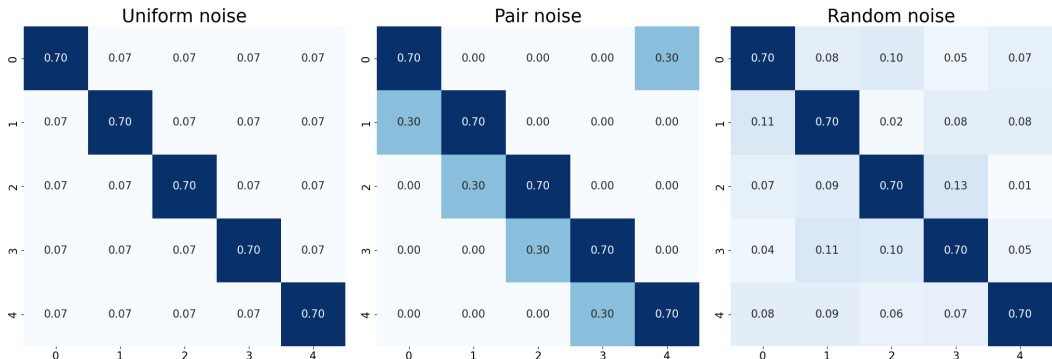

Figure A1: Label transition probability under 30% uniform noise, pair noise and random noise, respectively.

## B.2  Test accuracy of different methods under different noise rate

In Section 4, we investigated the performance of different LLN and GLN methods, here are additional experimental results.

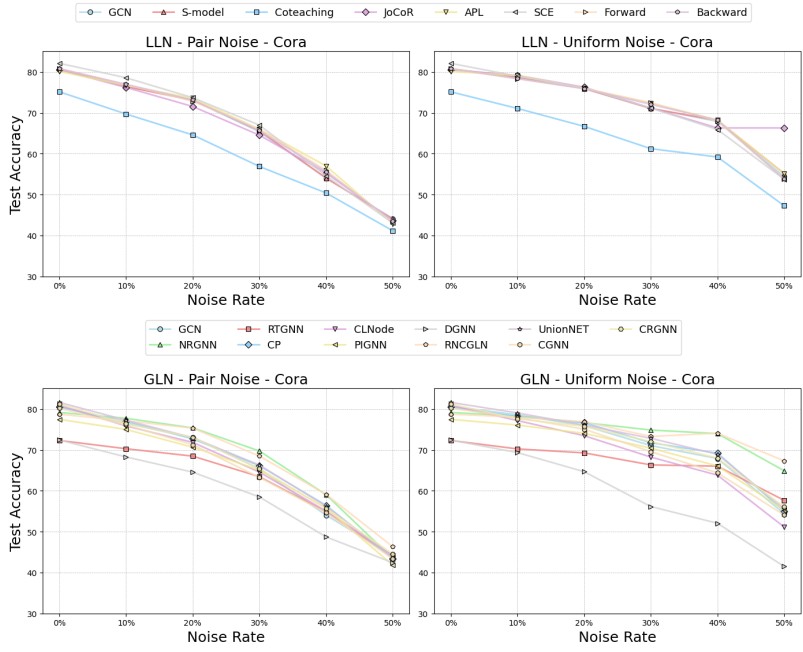

Figure A2: Test accuracy of LLN and GLN methods on Cora dataset under different rate of pair and uniform noise, respectively (10 Runs).

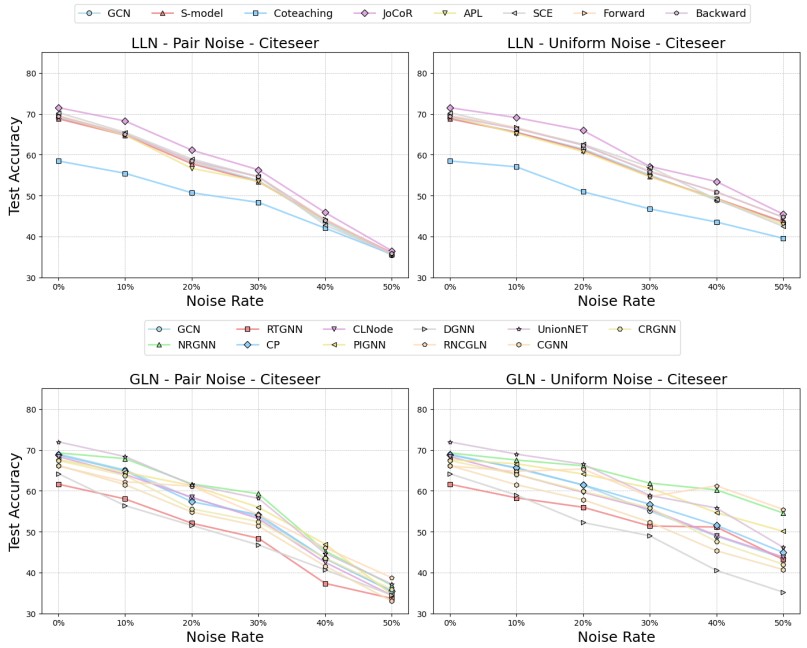

Figure A3: Test accuracy of LLN and GLN methods on Citeseer dataset under different rate of pair and uniform noise, respectively (10 Runs).

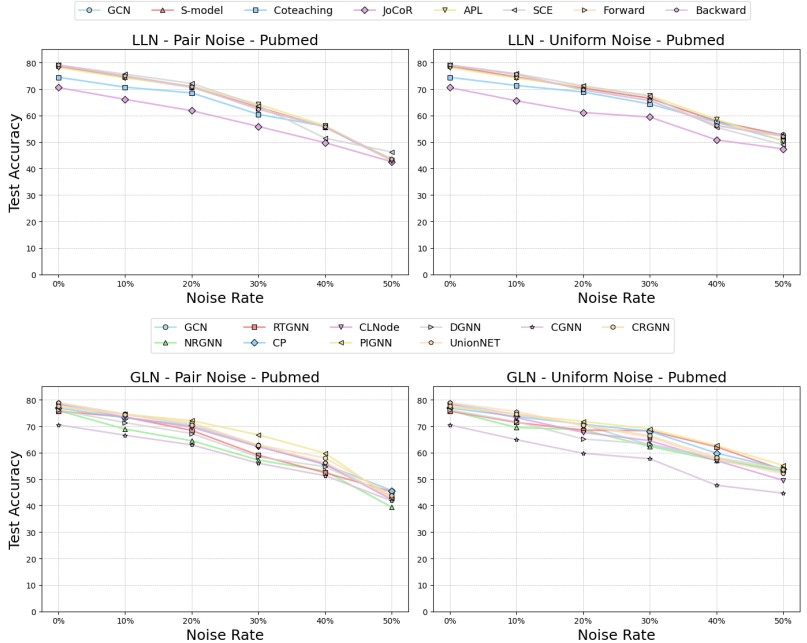

Figure A4: Test accuracy of LLN and GLN methods on Pubmed dataset under different rate of pair and uniform noise, respectively (10 Runs).

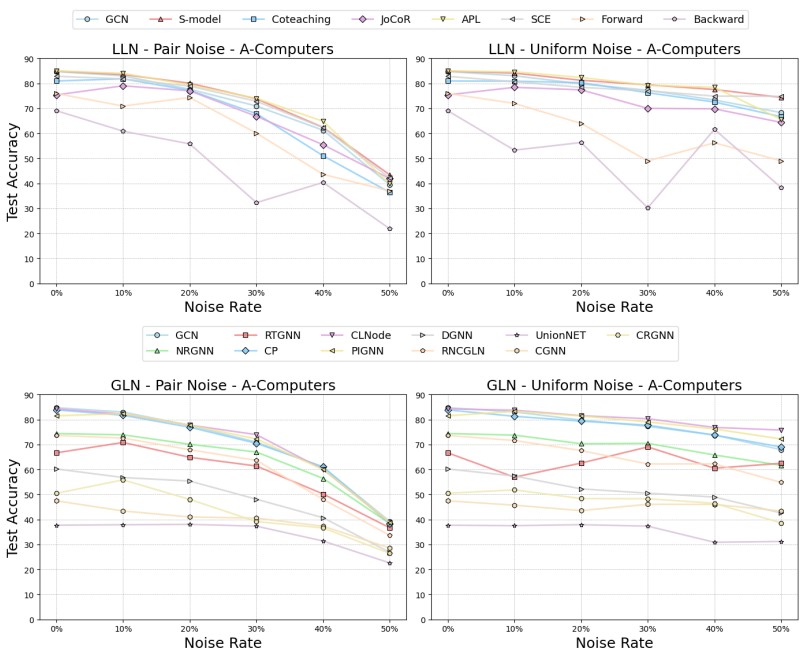

Figure A5: Test accuracy of LLN and GLN methods on Amazon-Computers dataset under different rate of pair and uniform noise, respectively (10 Runs).

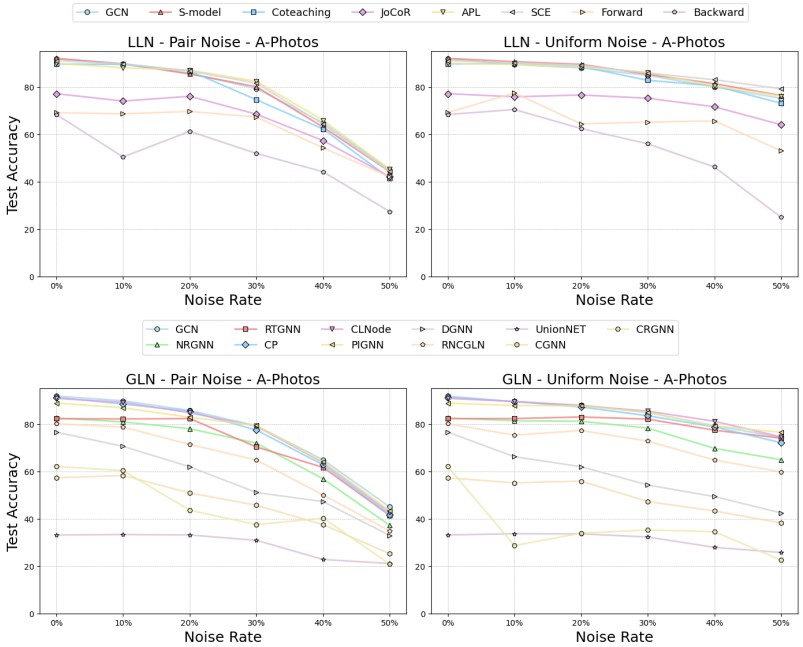

Figure A6: Test accuracy of LLN and GLN methods on Amazon-Photos dataset under different rate of pair and uniform noise, respectively (10 Runs).

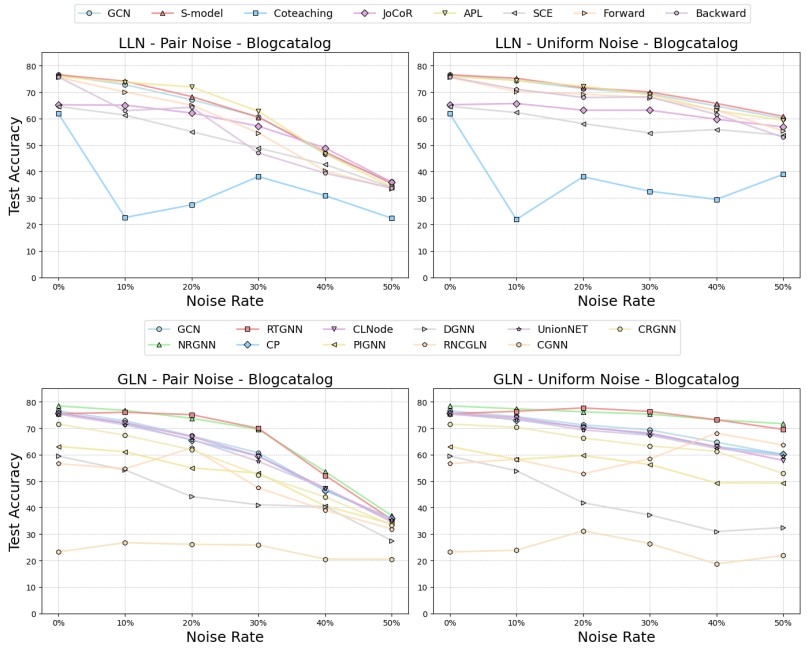

Figure A7: Test accuracy of LLN and GLN methods on Blogcatalog dataset under different rate of pair and uniform noise, respectively (10 Runs).

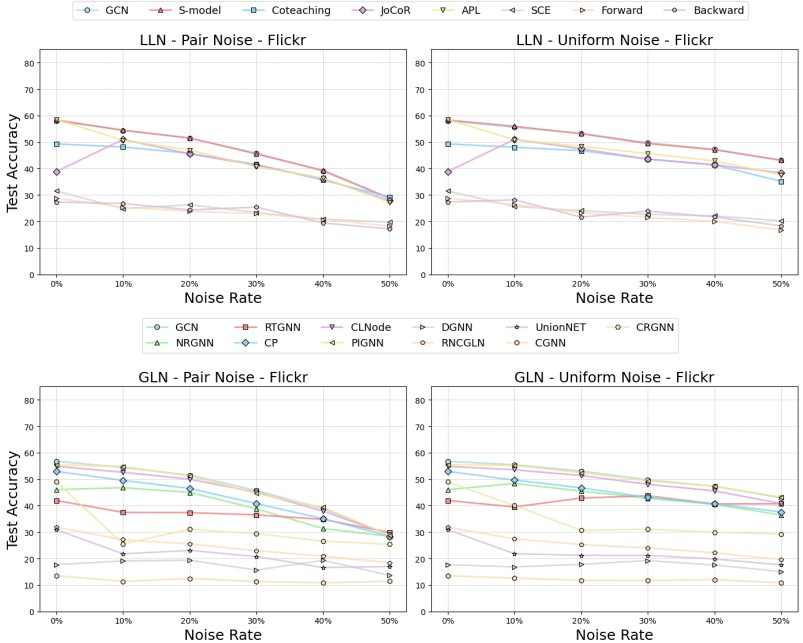

Figure A8: Test accuracy of LLN and GLN methods on Flickr dataset under different rate of pair and uniform noise, respectively (10 Runs).

## B.3    Additional results of time efficiency

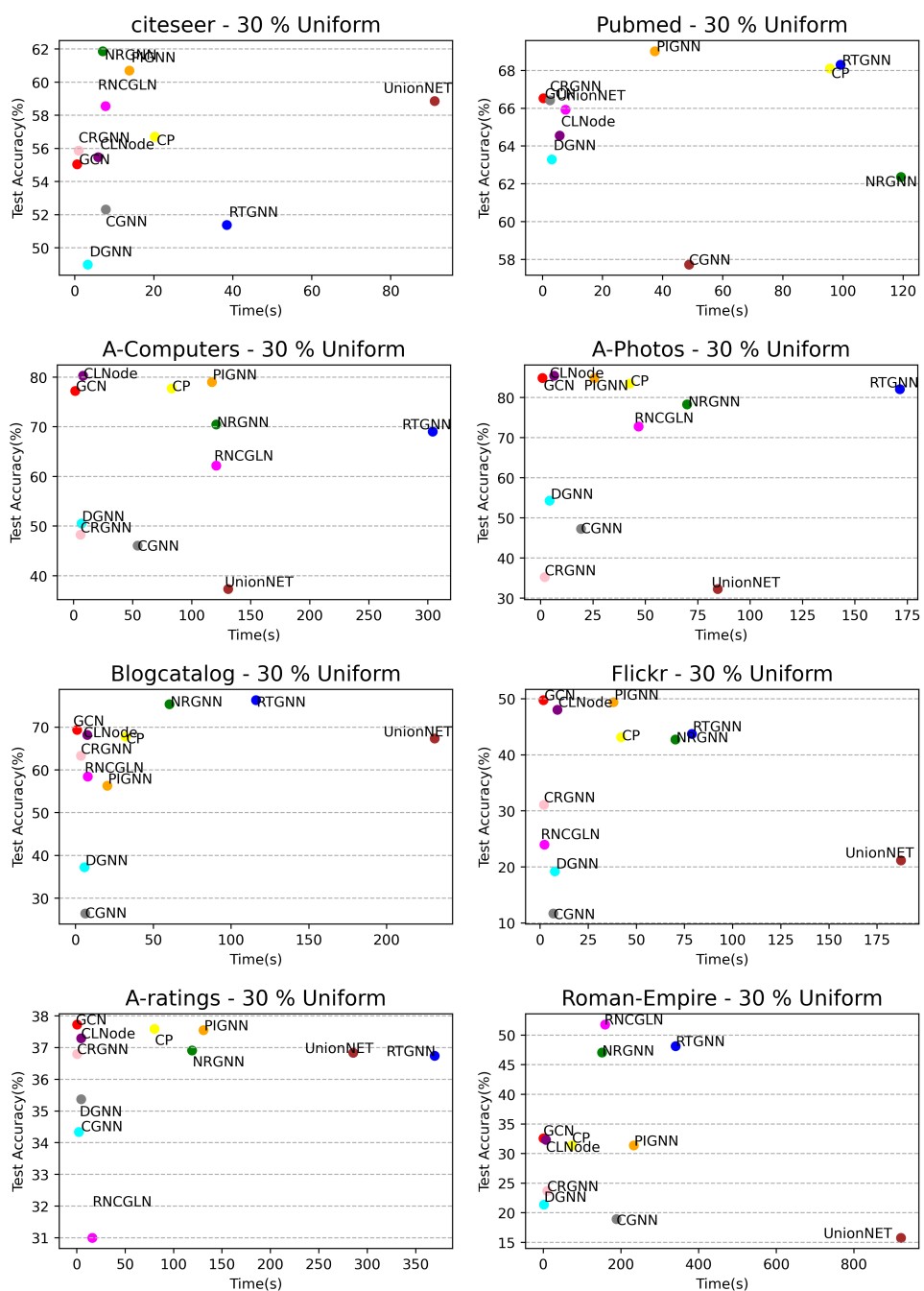

Figure A9: Time consumption and Test accuracy of different GLN methods on different datasets under 30% uniform noise (10 Runs).

# C    Additional details of the Benchmark

## C.1    Datasets

Table A7: Overview of the datasets used in this study.

| Dataset | # Nodes | # Edges | # Feat. | # Classes | # Homophily | Avg. # degree |
|---|---|---|---|---|---|---|
| Cora | 2,708 | 5,278 | 1,433 | 7 | 0.81 | 3.90 |
| Citeseer | 3,327 | 4,552 | 3,703 | 6 | 0.74 | 2.74 |
| Pubmed | 19,717 | 44,324 | 500 | 3 | 0.80 | 4.50 |
| Amazon-Computers | 13,752 | 491,722 | 767 | 10 | 0.78 | 35.76 |
| Amazon-Photos | 7,650 | 238,162 | 745 | 8 | 0.83 | 31.13 |
| DBLP | 17,716 | 105,734 | 1,639 | 4 | 0.83 | 5.97 |
| BlogCatalog | 5,196 | 343,486 | 8,189 | 6 | 0.40 | 66.11 |
| Flickr | 7575 | 239738 | 12047 | 9 | 0.24 | 63.30 |
| Amazon-Ratings | 24,492 | 93,050 | 300 | 5 | 0.38 | 7.60 |
| Roman-Empire | 22,662 | 32,927 | 300 | 18 | 0.05 | 2.90 |

**Cora, Citeseer and Pubmed** [19] are citation networks that most commonly used in previous graph learning under label noise studies  [1, 31, 24, 11, 18]. Each node represents a paper and each edge represents citation relationship between papers. Node features are 0/1-valued word vector indicating the absence/presence of the corresponding word from the dictionary. The label of each node is its category of research topic.

**Amazon-Computers and Amazon-Photo** [20] are co-purchase graphs extracted from Amazon, where each node represents a product, edges represent the co-purchased relationships between products. Features are bag-of-words vectors extracted from product reviews, labels of each node is its corresponding product category. These datasets were frequently used in robust graph learning under label noise studies [24, 12].

**Amazon-Ratings** [17] is derived from the Amazon product co-purchasing network metadata from SNAP Datasets. In this dataset, nodes represent products, while edges indicate products frequently bought together. Node features are calculated as the average of FastText embeddings for words in the product descriptions. The product ratings are categorized into five distinct classes as labels.

**Roman-Empire** [17] is derived from the Roman Empire article on English Wikipedia. The text was obtained from the English Wikipedia dump dated 2022.03.01. Each node in the dataset's graph represents a (non-unique) word in the text. Node features are calculated by FastText embeddings. Two words are connected if they follow each other in the text or are linked in the dependency tree of a sentence. Nodes are labeled based on their syntactic roles and identified using spaCy. The 17 most frequent roles are considered unique classes, while others are grouped into the 18th class.

**DBLP** [15] is an author collaboration network in computer science, each node represents a document and edges represent their citation links. Features are word vectors and labels are category of research topic. This dataset was used in study [1].

**Blogcatalog** [26] is a social network formed by an online community, each node is a blogger and edges represent their relationships. The features of each node are derived from the keywords present in their blog descriptions, and the labels are selected from a collection of established categories that reflect the bloggers' interests. This dataset was used in study [18].

**Flickr** [26] is a platform where users can share videos and images. User can follow each others thus form a social network. The feature of each node are generated from the user-specified tags, and labels represent the groups they have joined.

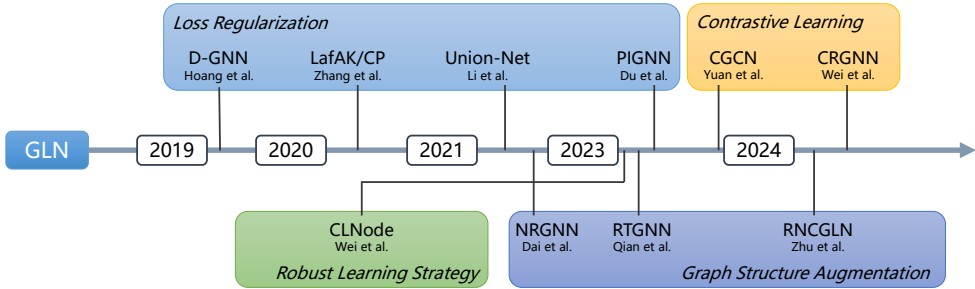

Figure A10: Timeline of GLN research. Existing GLN methods can be categorized into Loss regularization, Robust training strategy, Graph structure augmentation and Contrastive learning.

## C.2 Algorithms

### C.2.1 Graph Neural Networks with Label Noise

**NRGNN** [1] believe that since the labels on the graph are sparse, the falsely labeled nodes may affect the unlabeled nodes in its neighborhood, which make it difficult to receive supervision from correctly labeled nodes. To address these issues, NRGNN first connects nodes with similar features to create a refined graph. Based on this refined graph, precise pseudo-labels are generated, allowing unlabeled samples to receive more supervision from correctly labeled samples, thereby reducing the impact of noisy labels.

**RTGNN** [18] followed the work of NRGNN. The authors point out that although NRGNN emphasizes providing additional supervision for unlabeled nodes through link prediction, it does not distinguish between incorrectly labeled and correctly labeled nodes. Instead, it merely connects nodes with similar features indiscriminately, which may lead to the spread of influence from incorrectly labeled nodes. To solve this problem, the authors propose RTGNN, which, building based on NRGNN, uses the small-loss criterion from Co-teaching [4] to further distinguish between trustworthy and untrustworthy nodes, and corrects the labels of some untrustworthy nodes, mitigates some of the shortcomings of NRGNN.

**CP** [31] studied the impact of adversarial label-flipping attacks on the generalization ability of Graph Convolutional Networks (GCNs). To counteract label-flipping attacks, the authors proposed a defense framework named CP, which uses community labels as high-level signals to guide the node classification task. The CP framework includes a constraint with community information to prevent overfitting to the flipped noisy labels. The use of community labels is motivated by their similarity to the output of GCNs.

**D-GNN** [14] obtains a label noise robust Graph Neural Network by adopting backward loss correction [16] on GIN [25] backbone, which estimates the unbiased loss on clean labels.

**RNCGLN** [32] aim to simultaneously mitigate graph and label noise issues. To achieve this, it first use graph contrastive loss to conduct local graph learning, and adopt multi-head self-attention mechanism to learn node representation from a global perspective. Then utilize pseudo graphs and pseudo labels to deal with graph noise and label noise, respectively.

**CLNode** [24] adopt a curriculum learning strategy to mitigate the impact of label noise. To be specific, it first utilize a multi-perspective difficulty measurer to accurately measure the quality of training nodes. Then employ a training scheduler that selects appropriate training nodes to train GNN in each epoch based on the measured qualities. The authors demonstrated this method enhances the robustness of backbone GNN to label noise.

**PIGNN** [2] enhances the GNN's resistance to label noise by introducing additional pair-wise labels. The motivation is pair-wise labels are more robust than node-wise labels. In authors' definition, a pair interaction label is 1 if the nodes have the same label, and 0 otherwise, and they designed a PI label estimation method based on the similarity of node embeddings. During training, the estimated PI label serves as the confidence level for the node classifier's predictions, thereby constraining the training process of the node classifier. This method performs well on homophilic graphs but poorly on heterophilic graphs.

**Union-NET** [11] tries to limit the gradient passing process of mislabeled samples through neighborhood labeling, like a kind of neighborhood voting with node representation similarity weighting. A GNNs first generates node representations and predicted labels. Context nodes are then aggregated using random walks, and an attention mechanism calculates class probability distributions. This guides a reweighting scheme to minimize the impact of noisy labels. Labels are corrected by aligning them with the most consistent context labels, and a KL-divergence loss maintains alignment with the prior distribution. The training involves pre-training the GNN and updating model parameters with a combined loss function, ensuring robust training and effective label correction.

**CGNN** [30] addresses label noise in GNNs by combining neighborhood-based label correction and contrastive learning. It utilizes message passing neural networks to update node representations, integrating graph contrastive learning for consistent representations across augmented graph views. Finally, CGNN employs an MLP for prediction distributions and iteratively corrects noisy labels by comparing them with their neighbors and choosing the most labels.

**CR-GNN** [10] introduces contrastive learning to enhance GNNs robustness in the face of sparse and noisy node labels. Through techniques like feature masking and edge dropping, CR-GNN preserves node semantics while generating augmented views. Contrastive loss captures local structural information and mitigates noisy label effects, while dynamic cross-entropy loss addresses overfitting and adversarial vulnerabilities. Also, cross-space consistency ensures semantic alignment between embeddings.

### C.2.2 Learning with Label Noise methods

**S-model** [3] adds a noise adaptation layer that models the transition pattern of noisy labels on true labels. In the training procedure, this layer is parameterized by bias terms and allows the network to learn both the classifier and noise model simultaneously. In the test procedure, the noise adaptation layer is removed, which enables the network to predict true labels more effectively.

**Co-teaching** [4] works by simultaneously training two deep neural networks (DNNs), each of which selects a certain number of small-loss samples from them and passes these samples to the other for further training. It assume that mislabeling typically leads to larger losses and thus is less likely to be selected, and then each network selects the samples that perform best on its own with lower loss. This peer-to-peer training mechanism helps to reduce the effect of noisy labels, as both networks focus on more reliable data.

**JoCoR** [23] utilizes consistency maximization to deal with the noisy labels. Instead of using hard sampling, two different classifiers are made to converge in their predictions through explicit regularization. Specifically, the two classifiers are trained by a joint loss function to minimize the differences between them. During the training process, these two classifiers update their parameters at the same time and are jointly trained by means of a pseudo-twin network. The loss function consists of a supervised learning loss and a contrast loss, where the contrast loss is used to maximize the agreement between the two classifiers.

**SCE** [22] enhances the robustness of a model in the presence of noisy labels by combining a noise tolerance term with the standard cross entropy (CCE) loss. Inspired by the Kullback-Leibler scattering symmetry, SCE incorporates the reverse cross entropy loss, a noise tolerance term, and combines it with the standard CCE loss to improve the model's ability to tolerate noisy data. This approach not only retains the advantages of the CCE loss, but also significantly improves the generalization performance in noisy environments through symmetry processing and noise tolerance.

**Forward correction** [16] corrects the sample loss by linearly combining the softmax output of the DNN before applying the loss function. During the forward propagation process, the estimated label transfer probability is multiplied with the softmax output to obtain the corrected loss value. In this way, the softmax output of each sample is first combined with the corresponding transfer probability, and then the loss function is applied, which improves the robustness of the model in noisy labeling environments.

**Backward correction** [16] adjusts the loss for each sample by multiplying the estimated label transfer probability with the output of the specified DNN. The learning of the label transfer probability is decoupled from the learning of the model, and the label transfer matrix is first approximated using the softmax output of the DNN in the uncorrected loss case. Then, when retraining the DNN, the original loss is corrected based on the estimated matrix. The correction loss is computed by linearly combining

the loss values for each observable label, where the coefficients are the transfer probabilities from each observable label to the target label.

# D Package

We have developed an open-source software package NoisyGL, which provides a comprehensive and unbiased platform for evaluating GLN algorithms and advancing future research. The code structure of NoisyGL is well-designed to ensure fair experimental setups for different algorithms, easy reproducibility of experimental results, and support for flexible assembly of models for experiments. NoisyGL includes the following key modules. The Config module consists of the files that define the necessary hyperparameters and settings. The Dataset module is used to load datasets, and the label Contaminator modifies the raw data to create a contaminated graph. The Base-predictor serves as the base class for various reproduced LLN/GLN predictors, and the LLN/GLN Predictor evaluates the contaminated graph to predict performance.

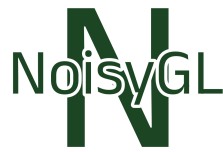

As shown in the Figure A11, the code structure is well-organized to ensure fair experimental settings across algorithms, easy reproduction of experimental results, and convenient trials on flexibly assembled models. Given a specific dataset and config file, a solver will return the learned structure and the task performance. For more details and updated features, please refer to our GitHub repository.

**General Experimental Settings.** We endeavor to follow the original implementations of the various GLN methods in their associated papers or source code. To this end, we integrate the different options into a standardized framework as shown in Figure. In this way, we can ensure consistency and comparability of experiments, allowing the performance of different GLN methods to be fairly evaluated on the same platform. We run most experiments on NVIDIA Geforce RTX 3090 GPU with 24 GB memory, the out-of-memory error during the training is reported as N/A in Appendix A. For the two large datasets, Amazon Ratings and Roman Empire, we run these experiments on NVIDIA A100 with 80GB memory.

**Hyperparameter.** We performed manual hyperparameter tuning to ensure an unbiased evaluation of these GLN methods. The hyperparameter search space for all methods is shown in Table A8. For details on the meaning of these hyperparameters, please refer to their original papers. Through exhaustive tuning and setup, we strive for the best performance of each method under different configurations, thus ensuring the accuracy and fairness of the evaluation.

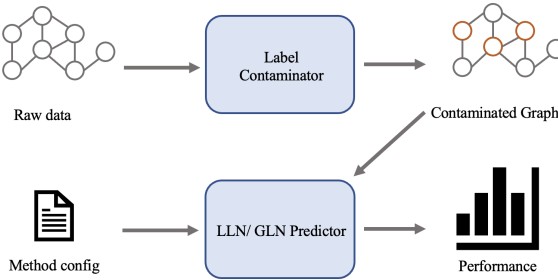

Figure A11: The structure of NoisyGL. The Raw data is processed by the Label Contaminator to introduce label noise, resulting in a Contaminated Graph. This contaminated graph, along with the Method config, is then input to the LLN/GLN Predictor, which evaluates performance metrics based on the specified method configuration.

Table A8: Hyper-parameter search space of all implemented GLN methods.

| Algorithm | Hyper-parameter | Search Space |
|---|---|---|
| General Settings | learning rate | 1e-1, 5e-2, 1e-2, 5e-3, 1e-3, 5e-4, 1e-4, 5e-5 |
| | weight decay | 5e-2, 5e-3, 5e-4, 5e-5 |
| | layer number | 2, 3, 4, 5 |
| | hidden size | 16, 32, 64, 128 |
| NRGNN [1] | $\alpha$ | 0.01, 0.02, 0.03 |
| | $\beta$ | 0.01, 0.1, 1, 10 |
| RTGNN [18] | $\tau$ | 0, 0.05, 0.1, 0.2 |
| | $\lambda$ | 0.05, 0.1, 0.2 |
| | $\alpha$ | 0.03, 0.1, 0.3, 1 |
| LAFAK/CP [31] | $\lambda$ | 0.1, 0.2, 0.3 |
| CLNode [24] | $\lambda_0$ | 0.25, 0.5, 0.75 |
| | $T$ | 50, 100, 150 |
| PIGNN [2] | N/A | N/A |
| DGNN [14] | N/A | N/A |
| RNCGLN [32] | $\alpha$ | $10^{-3}, 10^{-2}, ..., 10^3$ |
| UnionNET [11] | $\alpha$ | 0.1, 0.5, 1.0 |
| | $\beta$ | 0.1, 0.5, 1.0 |
| CGNN [30] | $\gamma$ | 0.6, 0.7, 0.8, 0.9, 0.95 |
| | $\omega$ | 0.6, 0.7, 0.8, 0.9, 0.95 |
| CRGNN [10] | $\alpha$ | 0.1, 0.2, 0.3, ... , 1 |
| | $\beta$ | 0.1, 0.2, 0.3, ... , 1 |

# E   Reproducibility

All of NoisyGL's experimental results are highly reproducible. We provide more detailed information on the following aspects to ensure the reproducibility of the experiments.

**Accessibility.** You can access all datasets, algorithm implementations, and experimental configurations in our open source project `https://github.com/eaglelab-zju/NoisyGL` without a personal request.

**Dataset.** The datasets used are publicly available. The Cora, Citeseer, and Pubmed datasets are accessible online and are used under the Creative Commons 4.0 license. The BlogCatalog and Flickr datasets were originally published by [26] and further processed in subsequent studies. To the best of our knowledge, these datasets do not have a specific license. The DBLP dataset can be found in [15] and is released under the MIT license. All of these datasets are licensed by the authors for academic research and do not contain any personally identifiable information or offensive content.

**Documentation and uses.** We've dedicated ourselves to providing users with comprehensive documentation, guaranteeing a smooth experience with our library. Our code includes ample comments to enhance readability. Furthermore, we furnish all essential files to replicate experimental outcomes, which also serve as illustrative guides on library utilization. Running the code is straightforward; users need only execute the '.py' files with specified arguments like data, method, and GPU.

**License.** We use an MIT license for our open-sourced project.

**Code maintenance.** We are dedicated to maintaining our code through continuous updates, actively engaging with user feedback, and addressing any issues promptly. Additionally, we are eager to receive contributions from the community to improve our library and benchmark algorithms. However, we will uphold rigorous version control measures to uphold reproducibility standards during maintenance procedures.

