# OpenReview forum: "NoisyGL: A Comprehensive Benchmark for Graph Neural Networks under Label Noise"
_NeurIPS.cc/2024/Datasets_and_Benchmarks_Track — NeurIPS 2024 Track Datasets and Benchmarks Poster_

### Official Review · Reviewer_Vxe6 · 2024-07-20
**NoisyGL: A Comprehensive Benchmark for Graph Neural Networks under Label Noise**

**Rating:** 6
**Confidence:** 4
**Correctness:** Yes
**Clarity:** Yes

**Review:**

Pros:
1. NoisyGL provides a standardized framework for evaluating GNNs under label noise. 2. The benchmark includes a wide range of GLN and LLN methods tested across multiple datasets, ensuring robustness and generalizability of the findings. 3. The benchmark and its associated library are open-source, promoting transparency, reproducibility, and further advancements in the field.

Cons:
1. What are the computational requirements for running the experiments included in the benchmark? 2. What measures have been taken to ensure that the benchmark framework and the included models are scalable to larger and more complex graphs? 3. Is the NoisyGL library compatible with other popular graph analysis and machine learning tools, and are there plans to improve integration with these tools in the future?

**Strengths:**

See Pros above.

**Additional Feedback:**

NA

**Documentation:**

NA

**Limitations:**

Yes

**Opportunities For Improvement:**

NA

**Relation To Prior Work:**

Yes

**Summary And Contributions:**

The paper introduces NoisyGL, a comprehensive benchmark designed to evaluate GNNs under label noise conditions. NoisyGL includes 17 representative methods (10 GLN and 7 LLN) and evaluates their performance across eight commonly used datasets with consistent experimental settings. The study also explores the impact of graph structure and noise types on GNN performance and introduces the NoisyGL library to facilitate future research.

---

> ### Author Rebuttal · Authors · 2024-08-17
>
> Thanks for your valuable feedbacks. Below are our replies.
>
> **Q1. What are the computational requirements for running the experiments included in the benchmark?**
>
> A1: Thank you for your feedback. We run these experiments on **NVIDIA Geforce RTX 3090 GPU with 24 GB memory**, the out-of-memory error during the training is reported as N/A in Appendix A.
>
> **Q2. What measures have been taken to ensure that the benchmark framework and the included models are scalable to larger and more complex graphs?**
>
> A2: Thank you for your feedback. In most cases (except for RNCGLN), we store the graph’s adjacency matrix as a **sparse matrix**. This approach ensures that most algorithms can efficiently handle large graphs. Additionally, we are planning to introduce a **mini-batching** strategy in the data preprocessing module to further reduce the computational requirements for large and complex networks.
>
> **Q3. Is the NoisyGL library compatible with other popular graph analysis and machine learning tools, and are there plans to improve integration with these tools in the future?**
>
> A3: Thank you for your suggestion. Our library is primarily built upon **PyTorch Geometric (PyG)**. In recent commits, we’ve introduced a hyperparameter optimization tool (located in [hpyerparam_opt.py](https://github.com/eaglelab-zju/NoisyGL/blob/main/hyperparam_opt.py)) using Microsoft's **Neural Network Intelligence (NNI)**. With this tool, users can easily optimize and update the hyperparameter configuration for each method, following the instructions we provided in the [README](https://github.com/eaglelab-zju/NoisyGL/blob/main/README.md).

---

### Official Review · Reviewer_gQY9 · 2024-07-24
**The paper provides a well-organized and comprehensive benchmark for evaluating GLN and LLN methods with rigorous experimental design, though it lacks diversity in datasets and innovation in the benchmark itself.**

**Rating:** 6
**Confidence:** 3
**Correctness:** Yes.
**Clarity:** The paper is well written.

**Review:**

**Pros**:
  - The paper is well-organized and easy to read.
  - The paper presents a well-structured benchmark with rigorous experimental design and evaluation.
  - Comprehensive analysis of both GLN and LLN methods provides a thorough understanding of the state-of-the-art.


**Cons**:
  - The paper could include more diverse datasets, especially heterogeneous graphs, to evaluate the methods more broadly.
  - While the insights of the benchmark is novel, the evaluated methods, metrics and datasets are existing ones, and there is no innovation in terms of benchmark itself.

Overall, while the paper offers a valuable and well-organized benchmark with exhaustive experimental design and analysis, it could benefit from the inclusion of more diverse datasets and innovative approaches to further enhance its impact.

**Strengths:**

See above.

**Additional Feedback:**

See above.

**Documentation:**

Yes.

**Ethics:**

N/A.

**Limitations:**

N/A.

**Opportunities For Improvement:**

See cons above.

**Relation To Prior Work:**

The related work is discussed in detail.

**Summary And Contributions:**

This work introduces NoisyGL, a comprehensive benchmark designed to evaluate the robustness and performance of GNNs when subjected to label noise.

Contributions:

1. Detailed experiments on eight node classification datasets reveal the strengths and weaknesses of various methods.

2. This work offers some key insights through experimental evaluation:

- LLN methods generally do not improve GNN robustness to label noise.
- GLN methods can alleviate label noise, but their effectiveness is scenario-dependent.
- Pair noise is the most detrimental type of label noise.
- The spread of label noise can be amplified by the graph structure, especially in sparse graphs, but graph structure augmentation can mitigate this spread.

---

> ### Author Rebuttal · Authors · 2024-08-17
>
> Thanks for your valuable feedbacks. Below are our replies.
>
> **Q1. The paper could include more diverse datasets, especially heterogeneous graphs, to evaluate the methods more broadly.**
>
> A1. Thank you for your suggestion. In the latest version of [NoisyGL](https://github.com/eaglelab-zju/NoisyGL), we’ve integrated two heterogeneous graph datasets. Below are their statistics.
>
> |Dataset	|  \# Nodes |  \# Edges|   \# Feat. |   \# Classes |  \# Homophily | Avg. \# degree|
> | - | - | - | - | - | - | - |
> | Amazon-ratings	| 24,492	| 93,050	| 300	| 5	 | 0.38	| 7.60|
> |Roman-empire	| 22,662	| 32,927	| 300	| 18	| 0.05	| 2.91 |
>
> The corresponding data preprocessing configurations are now available in [NoisyGL/config/_dataset](https://github.com/eaglelab-zju/NoisyGL/tree/main/config/_dataset), We utilized the same data split as the blogcatalog and flickr. Here is a portion of the experimental results.
>
> |       Dataset          |     Noise type            | gcn          | smodel       | jocor        | apl          | nrgnn        | cp           | clnode       | dgnn         |
> |:----------------|-----------------|:-------------|:-------------|:-------------|:-------------|:-------------|:-------------|:-------------|:-------------|
> | amazon-ratings | 0 % clean | 40.61 ± 0.21 | 40.66 ± 0.23 | 40.44 ± 0.34 | 38.89 ± 1.58 | 37.27 ± 0.19 | 40.06 ± 0.33 | 37.92 ± 0.26 | 38.93 ± 0.27 |
> | amazon-ratings | 30 % uniform | 37.07 ± 0.35 | 36.83 ± 0.82 | 36.85 ± 0.68 | 37.42 ± 0.45 | 36.21 ± 1.43 | 37.32 ± 0.66 | 37.04 ± 1.19 | 33.66 ± 0.72 |
> | roman-empire |   0 % clean | 36.78 ± 0.21 | 36.77 ± 0.22 | 35.60 ± 0.41 | 32.53 ± 1.28 | 49.81 ± 0.28 | 36.94 ± 0.38 | 36.07 ± 0.33 | 24.39 ± 0.36 |
> | roman-empire | 30 % uniform | 32.39 ± 0.62 | 32.74 ± 0.69 | 32.59 ± 0.59 | 28.06 ± 0.72 | 47.06 ± 0.90 | 31.20 ± 0.59 | 32.39 ± 0.52 | 21.75 ± 0.51 |
>
> None of the eight methods mentioned above achieved satisfactory performance, highlighting the need to develop GLN methods specifically for heterogeneous graphs.
>
> **Q2. While the insights of the benchmark is novel, the evaluated methods, metrics and datasets are existing ones, and there is no innovation in terms of benchmark itself.**
>
> A2. Thank you for your feedback. While designing the benchmark, we primarily focused on widely accepted experimental settings and common label noise types.
> However, during the early stages of our research, we explored a novel type of label noise that better reflects real-world scenarios. Here is the definition.
>
> **Random noise**, assumes that the true label has a probability of $\epsilon$ of being flipped to another class, but the flipping probabilities are not constant. Formally, we have $\sum_{j \neq i}  p(\widetilde{y}=j |y^* = i ) = \epsilon$.
>
> To provide context, let’s revisit the definitions of pair noise and uniform noise from section 2 of our paper. Random noise lies somewhere between uniform noise and pair noise. Specifically:
>
> **Uniform noise** flips labels to any other classes with a fixed probability.
>
> **Pair noise** only allows labels to flip to the corresponding pair class.
>
> **Random noise** flips labels to all classes with varying probabilities.
>
> Based on the definitions provided above, we can observe that Uniform noise and pair noise represent the two extremes of random noise. In practice, we randomly generate label flipping probabilities at the beginning of each experiment using diverse random seeds. Consequently, each experiment exhibits a different label transition pattern. The advantage of random noise lies in its ability to test model stability under various label noise patterns.
>
> Initially, we considered naming this new label noise pattern “**asymmetric noise**.” However, before submitting the paper, we noticed that some prior work had already used “asymmetric noise” to describe pair noise. To avoid ambiguity, we removed the relevant experiment results from the main text. Nevertheless, we retained the implementation and experimental results of random noise. Below is a portion of the experimental results.
>
>  Dataset | Noise type | GCN | NRGNN | RTGNN | CP | CLNode | PIGNN | DGNN | RNCGLN | UnionNET | CGNN | CR-GNN |
> | - | - | - | - | - | - | - | - | - | - | - | - | -|
> | Cora | Clean data | $ 80.66 \pm 0.54 $  | $ 79.16 \pm 0.74 $ | $ 72.31 \pm 1.94 $ | $ 80.49 \pm 1.02 $ | $ 80.90 \pm 0.73 $ | $ 77.46 \pm 1.28 $ | $ 72.41 \pm 2.77 $ | $ 78.72 \pm 1.02 $ | $ 81.61 \pm 0.73 $ | $ 80.18 \pm 1.28 $ | $ 81.23 \pm 1.09 $  |
> | Cora |  $ 50 \\% $ pair | $ 44.15 \pm 8.52 $ | $ 43.42 \pm 6.99 $ | $ 43.69 \pm 7.46 $ | $ 43.36 \pm 7.24 $ | $ 44.16 \pm 5.78 $ | $ 41.81 \pm 7.92 $ | $ 42.41 \pm 6.80 $ | $ 46.31 \pm 12.67 $ | $ 43.74 \pm 8.51 $ | $ 43.37 \pm 7.78 $ | $ 44.53 \pm 6.74 $
> | Cora |  $ 50 \\% $ uniform | $ 54.42 \pm 4.72 $ | $ 64.90 \pm 5.32 $ | $ 57.67 \pm 5.68 $ | $ 55.03 \pm 7.13 $ | $ 51.14 \pm 5.97 $ | $ 55.28 \pm 7.40 $ | $ 41.48 \pm 5.81 $ | $ 67.32 \pm 6.56 $ | $ 55.60 \pm 4.13 $ | $ 54.08 \pm 5.48 $ | $ 56.09 \pm 4.53 $
> | Cora |  $ 50 \\% $ random | $ 55.59 \pm 6.92 $ | $ 64.02 \pm 6.68 $ | $ 55.24 \pm 6.77 $ | $ 55.45 \pm 4.93 $ | $ 50.56 \pm 6.44 $ | $ 55.47 \pm 6.12 $ | $ 40.27 \pm 13.10 $ | $ 63.45 \pm 8.03 $ | $ 56.51 \pm 7.37 $ | $ 53.11 \pm 6.73 $ | $ 54.12 \pm 7.13 $
> --------
>
> From the data above, it is evident that random noise is more destructive than uniform noise but less harmful than pair noise in most cases. For methods with a **larger standard deviation** of test accuracy under random noise (Like DGNN, CLNode, and CR-GNN), we believe that they are **sensitive to variations in the label noise pattern**. You can find the relevant code implementation in the NoisyGL repository (see [NoisyGL/utils/labelnoise.py](https://github.com/eaglelab-zju/NoisyGL/blob/main/utils/labelnoise.py) "random noise") and experimental results from the paper appendix (see Appendix.A results of "asymmetric noise").

---

### Official Review · Reviewer_d2CP · 2024-07-25

**Rating:** 7
**Confidence:** 4

**Review:**

I generally like this work quite a lot. I think the community is in need of more comprehensive benchmarks such as this one, and that the insights provided here (and the new methods tested under this framework) can immensely benefit the field, I only have minor concerns that I will point out below, but overall I find the paper well-written, sensible, well-posed, and the figures and accompanying discussion clear and easy to follow.

I therefore support this work, and I may consider increasing my score after addressing my lesser comments.

**Strengths:**

- The GNN community has (imho) a reproducibility problem in some aspects, and it pretty much needs benchmarks such as this one, unifying and comprehensively testing the proposed methods.
- The paper is well-written and (from a quick glimpse) the code seems simple enough to easily add new methods in the future.
- All claims are generally well-discussed, insights are provided when necessary, and the work reads in general quite well.
- I find the research questions well-posed and satisfactorily answered.
- While limited in scope, the paper is conscious of it and correctly self-contained, properly describing its limitations and future work.

**Additional Feedback:**

- RQ6 in line 291 is RQ7.
- Add a citation in the claim you do in line 135.
- In line 122, a multinomial with n=1 is a categorical distribution.

**Clarity:**

The paper is clearly written. Only minor points:
- The description of pair noise was not clear to me until I read the second explanation much later in the text.
- For figure 1, I would consider drawing two horizontal lines to denote the accuracy of GCN.
- Tables 1 and 2 should also include the performance of the baseline models.

**Correctness:**

I did not find any concerns in this regard, everything is well docummented and detailed.

**Documentation:**

While the code looks rather clean, the benchmark itself seems to lack a documentation (beyond the README file).

**Ethics:**

I don't see any concerns.

**Limitations:**

I think the limitations are well discussed by the authors.

**Opportunities For Improvement:**

- I would have liked to see a section in the work with emphasis on the software-side of the benchmark, e.g., what "standardized backbones" were used, or how is the code prepared to easily allow users to add new methods and benchmarks.
- Similarly, I miss one experiment where other architectures (besides GNNs) are tested to different levels of noise. For example, [L-CAT](https://arxiv.org/abs/2211.11853) (section 6.3.1) showed that their proposed architecture was more robust to feature- and edge- noise than GCNs and [GATs](https://arxiv.org/abs/1710.10903).
- I found some grammatical errors in the English text that were a bit distracting. I did not take note, as I did not consider them severe, but the authors could consider going through the paper once more to polish those rough edges.

**Relation To Prior Work:**

I am personally not aware of any other benchmark of this kind. Given the scope of the paper, I do not consider it essential to describe other non-benchmark-related prior works.

**Summary And Contributions:**

This work provides a comprehensive benchmark of Graph Neural Networks label noise, which was missing in the literature. Specifically, the authors have gathered eight commonly-used datasets with different properties and sizes, implemented ten methods tailored for GNNs under label noise, and seven general (non-GNN) methods for training under label noise. Then, the authors have run a large number of experiments under different types of label noise ratios and types, and presented all the experimental results in an easy-to-understand manner, providing novel insights on the different scenarios for which label noise can or not be effectively addressed to this day. Moreover, the code is publicly available.

---

> ### Author Rebuttal · Authors · 2024-08-17
>
> Thanks for your careful review and valuable feedbacks. Below are our replies.
>
> **Q1. I would have liked to see a section in the work with emphasis on the software-side of the benchmark, e.g., what "standardized backbones" were used, or how is the code prepared to easily allow users to add new methods and benchmarks.**
>
> A1. Thank you for your suggestion. In the next revision, we will include a software description section following Appendix D. Here’s an overview:
>
> Most GLN methods use GNNs as backbone. For instance, NRGNN, RTGNN, CP, CLNode, PIGNN, and UnionNET use GCN as backbone. However, their  backbone implementations are differ, leading to an unfair comparison. NoisyGL unifies these GNN backbone into a common Class with standardized API (located in predictor/module/GNNs) to eliminate the impact caused by backbone differences. Specificly, we implemented MLP,GCN and GIN, more backbones are comming soon.
>
> NoisyGL offers the following features:
>
> 1.  **A unified data loader module for diverse datasets.** Users can customize the configuration file of the dataset (located in config/_dataset) to modify data splitting and preprocessing strategies.
>
> 2.  **Generic noise injection schemes.** These schemes, widely used in previous studies, can comprehensively evaluate the robustness of each method.
>
> 3.  **Generic Base_predictor class.** NoisyGL provides a generic implementation template and API for different GLN predictors. Users can develop their customized methods by overriding specific methods.
>
> 4.  **Integrated hyperparameter optimization tool.** NoisyGL integrates Neural Network Intelligence (NNI) provided by Microsoft. Users can easily optimize and update hyperparameters for each method based on the instructions we provided in the README.
>
>
> The above features provide everyone with convenience and freedom when using our library. Users can modify the implementation details of specific methods, or add new modules to implement their novel methods within the framework we provide easily.
>
> Additionally, we have updated the README file with above descriptions.
>
> **Q2. Similarly, I miss one experiment where other architectures (besides GNNs) are tested to different levels of noise. For example, L-CAT (section 6.3.1) showed that their proposed architecture was more robust to feature and edge noise than GCNs and GATs.**
>
> A2. Thank you for your suggestion. We have implemented two non-GNN predictors, MLP and L-CAT, and added them to our library as you suggested. In the final version, we will cite the corresponding papers. So far, we have conducted a preliminary experiment for the newly introduced methods, and the following are some of the experimental results.
>
> |   |  Dataset  |  Noise type  |  L-CAT  |
> | --- | --- | --- | --- |
> |  0  |  cora  |  0 % clean  |  75.35 ± 0.29  |
> |  1  |  cora  |  20 % uniform  |  68.78 ±  2.14  |
> |  2  |  pubmed  |  0 % clean  |  74.99 ± 0.66  |
> |  3  |  pubmed  |  20 % uniform  |  66.73 ± 5.35  |
>
> The newly introduced methods are currently in the testing phase. We’re actively optimizing the code implementation and adjusting hyperparameters in the development branch to ensure fair comparisons. Once this process is complete, we’ll merge it into the main branch.
>
> **Q3. I found some grammatical errors in the English text that were a bit distracting. I did not take note, as I did not consider them severe, but the authors could consider going through the paper once more to polish those rough edges.**
>
> A3. Thank you for your reminder, we have carefully revised our paper and found several grammatical problems, and we will correct them in the final version.
>
> **Q4. The description of pair noise was not clear to me until I read the second explanation much later in the text.**
>
> A4. Thank you for your feedback. In the next revision, we will redescribe the definition of pair noise in a formal language in Section 2.
>
> **Q5. For figure 1, I would consider drawing two horizontal lines to denote the accuracy of GCN.**
>
> A5. Thank you for your suggestion. We will add two horizontal lines in Figure 1 in the next revision so that readers can easily compare the performance of different methods with the GCN baseline. To give the reader a clear overview, we also include this figure in README.
>
> **Q6. Tables 1 and 2 should also include the performance of the baseline models.**
>
> A6. Thank you for your suggestion. For Table 1, we have conducted several additional experiments.  And we will include these results in the final version. Now, you can check a portion of these results from the attachment.
> While adding these results won’t alter the conclusions drawn in findings 5 and 7, we can conclude that GLN and LLN methods can mitigate the overfitting effect of label noise in most circumstances. Among these methods, NRGNN exhibits the strongest resistance to overfitting the noisy labels.
> For table 2, we consider six methods to be enough to conclude, and more results can be found in Appendix A.
>
> **Q7. While the code looks rather clean, the benchmark itself seems to lack a documentation (beyond the README file).**
>
> A7. Thank you for your suggestion. We are currently working on a detailed API document. Once it’s complete, we will update the API reference link in the README.
>
> **Q8. RQ6 in line 291 is RQ7.**
>
> A8. Thank you for your reminder. Upon investigating RQ6, we have two findings. Finging 7 in line 291 is the second finging of RQ6.
>
> **Q9. Add a citation in the claim you do in line 135.**
>
> A9. Thank you for the reminder. In line 135, we will cite the paper NRGNN, where the author claimed "Though extensive approaches have been proposed for learning with noisy labels such as loss correction and sample selection, they are not directly applicable for learning GNNs with limited noisy labels." in section I.
>
> **Q10. In line 122, a multinomial with n=1 is a categorical distribution.**
>
> A10. Much appreciated for your correctness. We will update the description in the final version.

---

> > ### Comment · Reviewer_d2CP · 2024-08-31
> >
> > Dear authors, I truly apologize for the late response.
> >
> > In any case, I just wanted to acknowledge that I have read the rebuttal answer, and that I truly appreciate the detailed responses, additional experiments, and the effort put by you on the rebuttal.
> >
> > I think this work is in a really good shape. While I cannot update the score any more on OR (even though I am still within the discussion period), I want to reflect here that I would have *update my score to 8* if I were allowed to do it. I am sure the AC you take it into consideration anyway.

---

### Official Review · Reviewer_EW3m · 2024-08-01

**Rating:** 8
**Confidence:** 3
**Correctness:** Yes.
**Clarity:** Yes, the paper is well organized.

**Review:**

Pros:
1. The paper is well organized and the motivation is clear, each problem has the corresponding experimental design.
2. The authors point out the future work directions in three important dimensions.
3. This work unifies experimental settings and makes fair comparisons, it is of great reference significance for future works.

Cons:
1. The authors try to make the conclusions convincing via extensive experiments and explanations of experimental results. But lack of theoretical analysis of each problem.

**Strengths:**

Refer to Review.

**Additional Feedback:**

N/A

**Documentation:**

Yes, the documentation is sufficient.

**Ethics:**

No, there are no ethical concerns.

**Limitations:**

Yes, the authors discuss the future work and limitations.

**Opportunities For Improvement:**

The authors could analyze some formulas in previous GLN methods.

**Relation To Prior Work:**

Yes, it discusses how this work differs from previous contributions.

**Summary And Contributions:**

This paper provides the first comprehensive benchmark for graph neural networks under label noise, which unifies experimental settings and interfaces for fair comparisons and detailed analyses of GLN methods across various datasets. Moreover, it uncovers several important insights that were missed in previous research.

---

> ### Author Rebuttal · Authors · 2024-08-17
>
> Thanks for your valuable feedbacks. Below are our replies.
>
> **Q1. The authors try to make the conclusions convincing via extensive experiments and explanations of experimental results. But lack of theoretical analysis of each problem. The authors could analyze some formulas in previous GLN methods.**
>
> A1. Thank you for your suggestion. Here are theoretical analysis for some findings.
>
> **Theoretical analysis of finding 1 (RQ1).** When addressing graph label noise, we believe that it’s crucial to seek additional information beyond the contaminated labels, leveraging the graph structure. While most LLN methods simply preventing model from over-fitting, which is not sufficient.
> Furthermore, transition matrix-based LLN methods (e.g., Forward and Backward) often struggle to learn transition patterns due to the limited number of labeled nodes. As a result, they may suffer from distribution shifts, leading to unstable performance across different train-test splits.
>
> **Theoretical analysis of finding 2 (RQ2).** We observed that GLN methods have limitations in handling highly heterogeneous graphs. That's beacuse GLN methods were built upon homophily assumption.
>
> For example, CP assume that communities can guide node classification task as high-level signals. It employs an extra term in loss function which minimizing the cross-entropy between the GCN output and community labels. While this approach doesn't works on highly heterogeneous graphs where node labels within the same community exhibit diversity.
>
> $L\_{c}(\theta^{(L-1)},{\bf W}^{c};{\bf A},{\bf X})=-\sum\_{v}^{V\_{L}^{c}}\sum\_{i=1}^{K}{\bf Y}\_{v,i}^{c}l n{\bf Z}\_{v,i}^{c}$
>
> Union-Net incorporates a sample re-weighting process, akin to a voting mechanism. Specifically, neighborhood nodes with similar embeddings receive higher weights (as discussed in Sec 4.2). Notably, this approach relies on the homophily assumption.  Another similar example is [PIGNN](http://arxiv.org/abs/2106.07451).
>
> $p\_{r}(\hat{\bf{y}}|\hat{\bf{x}},S)= \sum\_{{\bf{x}}\_{i}\in S,{\bf{y}}\_i=\hat{\bf{y}}}\frac{\exp({\bf{h}}\_{{\bf{x}}\_{i}}^{T}{\bf{h}}\_{\hat{{\bf{x}}}})}{\sum\_{\bf{x}\_{i}\in S}^{{\bf{}}} \exp({\bf{h}}\_{{\bf{x}}\_{j}}^{T}{\bf{h}}\_{\hat{{\bf{x}}}})}{\bf{y}}\_{i}，$
>
> For NRGNN, it connects unlabeled nodes to labeled nodes based on embedding similarity.  Initially, it computes a new adjacency matrix in following way (described in Section 4),
>
> $Z=G C N({\bf A},{\bf X}), \quad \mathrm{S}\_{i j}=\sigma(\mathrm{z}\_{i}\mathrm{z}\_{j}^{T})$
>
> NRGNN connects unlabeled nodes with label nodes according to $S$ to get the altered adjacency matrix $S^L$. Then it calculate label prediction by putting $S^L$ and feature matrix into a GNN classifier.
>
> $\hat{Y}=f\_{{\cal G}}(\mathrm{S}^{L},X)$
>
> This graph structure augmentation process relies on the homophily assumption, improving feature homophily within the neighborhood. It shares similarities with RTGNN and RNCGLN. Other GLN methods, although not explicitly based on homophily assumptions, utilize homophily-driven Graph Neural Networks (GNNs) as their backbones, which restricts their effectiveness on heterogeneous graphs.
>
> **Theoretical analysis of finding 5 (RQ5).** We observed that pair noise is more likely to cause the model to overfit noisy labels. To explain this, Consider a toy model with a single linear layer:
>
> $\mathcal{L} = CrossEntropy(\tilde{y}, \hat{y}) =\tilde{y} \cdot log (Wx + b) \\\frac{ \partial \mathcal{L} }{ \partial w\_{ij} } = \tilde{y}\_i \cdot x\_j \frac{1}{w\_{ij} \cdot x\_j + b}$
>
> Uniform noise allows the true label to flip to any other class, and incorrect parameter updates due to mislabeled instances can be evenly distributed across all classes. This type of parameter update uniformly increases the classification probability of all categories and is less likely to result in misclassification. Consider$i$isn't the true label of instance $x$, we have
>
> $\mathbb{P}\_{uniform}(\frac{ \partial \mathcal{L} }{ \partial w\_{ij} }) =\mathbb{P}(\tilde{y}=i) \cdot \frac{1}{w\_{ij} \cdot x\_j + b}\cdot =  \frac{\epsilon}{c-1} \cdot \frac{1}{w\_{ij} \cdot x\_j + b}\cdot$
>
> However, for classifiers, pair noise can be even more misleading. All incorrect parameter updates concentrate within the paired class, increasing the likelihood of incorrect predictions. Once fully trained, the classifier tends to predict the paired class. Suppose $i$ is the pair class of instance $x$, we have
>
> $\mathbb{P}\_{pair}(\frac{ \partial \mathcal{L} }{ \partial w\_{ij} }) = \epsilon \cdot \frac{1}{w\_{ij} \cdot x\_j + b}\cdot$
>
> **Theoretical analysis of finding 6 (RQ6).** The spread effect of label noise can explained from two aspects. Firstly, the message passing mechanism often leads to similar representations within a node’s neighborhood. Secondly, graph data tend to have sparse labels. When a node is mislabeled, its neighboring unlabeled nodes with similar representations may also be misclassified. While  graph structure augmentation methods can make graph denser during the up-sampling process. As a result, unlabeled nodes receive more supervision from their neighborhood, reducing their reliance on a small number of incorrectly labeled samples.
>
> **Theoretical analysis of finding 7 (RQ6).** For sparse graphs, the prediction results of unlabeled nodes rely heavily on the annotated nodes in their neighborhood. If these nodes are mislabeled, it will lead to erroneous learning of the embedding for the unlabeled nodes in neighborhood. In contrast, for dense graphs, the neighborhood of unlabeled nodes contains many annotated nodes that can serve as references. As a result, the classifier model is more likely to find correct supervision from these annotated nodes.

---

> > ### Comment · Reviewer_EW3m · 2024-08-18
> >
> > Thanks for your response. I suggest that the above analysis be put into the final version, and I think this work is worth further exploring.

---

> > > ### Author Rebuttal · Authors · 2024-08-19
> > >
> > > Thank you again for your valuable reviews and reply! We will reorganize these analysis and incorporate it into the final version as you suggested.

---

### Decision · Program_Chairs · 2024-09-26

**Decision:**

Accept (Poster)

**Comment:**

This paper provides the first comprehensive benchmark for graph neural networks under label noise, which unifies experimental settings and interfaces for fair comparisons and detailed analyses of GLN methods across various datasets. Moreover, it uncovers several important insights that were missed in previous research.
All reviewers think it is a good paper and agree to accept it. So I suggest to accept it.